# Polygenic scoring accuracy varies across the genetic ancestry continuum

Yi Ding[1 ✉], Kangcheng Hou[1], Ziqi Xu[2], Aditya Pimplaskar[1], Ella Petter[2], Kristin Boulier[1], Florian Privé[3], Bjarni J. Vilhjálmsson[3,4,5], Loes M. Olde Loohuis[6,7] & Bogdan Pasaniuc[1,7,8,9,10 ✉]

Polygenic scores (PGSs) have limited portability across different groupings of individuals (for example, by genetic ancestries and/or social determinants of health), preventing their equitable use[1–3]. PGS portability has typically been assessed using a single aggregate population-level statistic (for example, $R^2$)[4], ignoring inter-individual variation within the population. Here, using a large and diverse Los Angeles biobank[5] (ATLAS, $n = 36{,}778$) along with the UK Biobank[6] (UKBB, $n = 487{,}409$), we show that PGS accuracy decreases individual-to-individual along the continuum of genetic ancestries[7] in all considered populations, even within traditionally labelled 'homogeneous' genetic ancestries. The decreasing trend is well captured by a continuous measure of genetic distance (GD) from the PGS training data: Pearson correlation of −0.95 between GD and PGS accuracy averaged across 84 traits. When applying PGS models trained on individuals labelled as white British in the UKBB to individuals with European ancestries in ATLAS, individuals in the furthest GD decile have 14% lower accuracy relative to the closest decile; notably, the closest GD decile of individuals with Hispanic Latino American ancestries show similar PGS performance to the furthest GD decile of individuals with European ancestries. GD is significantly correlated with PGS estimates themselves for 82 of 84 traits, further emphasizing the importance of incorporating the continuum of genetic ancestries in PGS interpretation. Our results highlight the need to move away from discrete genetic ancestry clusters towards the continuum of genetic ancestries when considering PGSs.

PGSs—estimates of an individual's genetic predisposition for complex traits and diseases (that is, genetic liability; also referred to as genetic value)—have garnered tremendous attention recently across a wide range of fields, from personalized genomic medicine[4,8–10] to disease risk prediction and prevention[11–14] to socio-genomics[3,15]. However, the variation in PGS performance across different genetic ancestries and/or socio-demographic features (for example, sex, age and social determinants of health)[2] poses a critical equity barrier that has prevented widespread adoption of PGSs. Similar portability issues have also been reported for non-genetic clinical models[16–18]. The interpretation and application of PGSs are further complicated by the conflation of genetic ancestries with social constructs such as nationality, race and/or ethnicity. Here we investigate PGS performance across genetically inferred ancestry (GIA), which describes the genetic similarity of an individual to a reference dataset (for example, 1000 Genomes[19]) as inferred by methods such as principal component analysis (PCA); GIAs do not represent the full genetic diversity of human populations.

Genetic prediction and its accuracy (or reliability) have been extensively studied in agricultural settings with a focus on breeding programmes[20–23]. At the population level, PGS accuracy can be expressed as a function of heritability, training sample size and the number of markers used in the predictor in single[24–26] or multi-population settings with or without effect size heterogeneity[27]. At the individual level, accuracy of genetic prediction from pedigree data[28–30] can be derived as a function of the inverse of the coefficient matrix of mixed-models equations, whereas accuracy of genetic prediction using whole-genome genetic data can be derived similarly, with the pedigree matrix replaced with the genomic relationships matrix[21–23,27,31,32] among training and testing individuals. Simulations guided by dairy breeding programmes showcase that genomic prediction accuracy varies with genetic relatedness of the testing individual to the training data[33,34] as well as across generations, owing to the decay of genetic relationships[35].

In humans, PGS performance evaluation has traditionally relied on population-level accuracy metrics (for example, $R^2$)[2,4]. PGS accuracy decays as the target populations become more dissimilar from the

[1]Bioinformatics Interdepartmental Program, UCLA, Los Angeles, CA, USA. [2]Department of Computer Science, UCLA, Los Angeles, CA, USA. [3]National Centre for Register-based Research, Aarhus University, Aarhus, Denmark. [4]Bioinformatics Research Centre, Aarhus University, Aarhus, Denmark. [5]Novo Nordisk Foundation Center for Genomic Mechanisms of Disease, Broad Institute, Cambridge, MA, USA. [6]Center for Neurobehavioral Genetics, Semel Institute for Neuroscience and Human Behavior, David Geffen School of Medicine at UCLA, Los Angeles, CA, USA. [7]Department of Human Genetics, David Geffen School of Medicine at UCLA, Los Angeles, CA, USA. [8]Department of Computational Medicine, David Geffen School of Medicine at UCLA, Los Angeles, CA, USA. [9]Department of Pathology and Laboratory Medicine, David Geffen School of Medicine at UCLA, Los Angeles, CA, USA. [10]Institute for Precision Health, UCLA, Los Angeles, CA, USA. ✉e-mail: yiding920@ucla.edu; pasaniuc@ucla.edu

training data using either relatedness[36,37] or continental or subcontinental ancestry groupings[1,38–40]; the decay may be explained by differences in linkage disequilibrium, minor allele frequencies and/or heterogeneity in genetic effects due to gene–gene and gene–environment interactions[41]. However, population-level metrics of accuracy provide only an aggregate (average) metric for all individuals in the population, thus implicitly assuming some level of homogeneity across individuals[2,4,42]. Homogeneous populations are an idealized concept that only roughly approximate human data; human diversity exists along a genetic ancestry continuum without clearly defined clusters and with various correlations between genetic and socio-environmental factors[7,42–46]. Grouping individuals into discrete GIA clusters obscures the impact of individual variation on PGS accuracy. This is evident among individuals with recently admixed genomes for which genetic ancestries vary individual-to-individual and locus-to-locus in the genome. For example, a single population-level PGS accuracy estimated across all African Americans overestimates PGS accuracy for African Americans with large proportions of African GIA[40]; likewise, coronary artery disease PGS performs poorly in Hispanic individuals with high proportions of African GIA[47]. The genetic ancestry continuum affects PGS accuracy even in traditionally labelled 'homogeneous' or 'non-admixed' populations. For example, PGS accuracy decays across a gradient of subcontinental ancestries within Europe as the target cohorts become more genetically dissimilar from the PGS training data[39,45]. Assessing PGS accuracy using population-level metrics is further complicated by technical issues in assigning individuals to discrete clusters of GIA. Different algorithms and/or reference panels may assign the same individual to different clusters[39,42,48], leading to different PGS accuracies. Moreover, many individuals are not assigned to any cluster owing to limited reference panels used for genetic ancestry inference[5,39], leaving such individuals outside PGS characterization. This poses equity concerns as it limits PGS applications only to individuals within well-defined GIAs.

Here we leverage classical theory[28–30] and methods that characterize PGS performance at the level of a single target individual[49] to evaluate the impact of the genetic ancestry continuum on PGS accuracy. We use simulations and real-data analyses to show that PGS accuracy decays continuously individual-to-individual across the genetic continuum as a function of GD from the PGS training data; GD is defined as a PCA projection of the target individual on the training data used to estimate the PGS weights. We leverage a large and diverse Los Angeles biobank at the University of California, Los Angeles[5] (ATLAS, $n = 36,778$) along with the UK Biobank[6] (UKBB, $n = 487,409$) to investigate the interplay between genetic ancestries and PGS for 84 complex traits and diseases. The accuracy of PGS models trained on individuals labelled as white British (WB; see Methods for naming convention used in this work) in the UKBB ($n = 371,018$) is negatively correlated with GD for all considered traits (average Pearson $R = -0.95$ across 84 traits), demonstrating pervasive individual variation in PGS accuracy. The negative correlation remains significant even when restricted to traditionally defined GIA clusters (ranging from $R = -0.43$ for East Asian GIA to $R = -0.85$ for the African American GIA in ATLAS). On average across the 84 traits, when rank-ordering individuals according to distance from training data, PGS accuracy decreases by 14% in the furthest versus closest decile in the European GIA. Notably, the furthest decile of individuals of European ancestries showed similar accuracy to the closest decile of Hispanic Latino individuals. Characterizing PGS accuracy across the continuum allows the inclusion of individuals unassigned to any GIA (6% of all ATLAS), thus allowing more individuals to be included in PGS applications. Finally, we explore the relationship between GD and PGS estimates themselves. Of 84 PGSs, 82 show significant correlation between GD and PGS with 30 showing opposite correlation (GD, trait) versus (GD, PGS); we exemplify the importance of incorporating GD in interpretation of PGSs using height and neutrophils in the ATLAS data. Our results demonstrate the need to incorporate the genetic ancestry continuum in assessing PGS performance and/or bias.

## Overview of the study

PGS accuracy has conventionally been assessed at the level of discrete GIA clusters using population-level metrics of accuracy. Individuals from diverse genetic backgrounds are routinely grouped into discrete GIA clusters using computational inference methods such as PCA[50] and/or admixture analysis[51] (Fig. 1a). Population-level metrics of PGS accuracy are then estimated for each GIA cluster and generalized to everyone in the cluster (Fig. 1b). This approach has three major limitations: the inter-individual variability within each cluster is ignored; the GIA cluster boundary is sensitive to algorithms and reference panels used for clustering; and a substantial proportion of individuals may not be assigned to any GIA owing to a lack of reference panels for genetic ancestry inference (for example, individuals of uncommon or admixed ancestries).

Here we evaluate PGS accuracy across the genetic ancestry continuum at the level of a single target individual. We model the phenotype of individual $i$ as $y_i = x_i^\top \beta + \epsilon_i$, in which $x_i$ is an $M \times 1$ vector of standardized genotypes for $M$ variants, $\beta$ is an $M \times 1$ vector of standardized causal effects, and $\epsilon_i$ is random noise. Under a random effects model, genetic liability $g_i = x_i^\top \beta$ and its PGS estimate $\hat{g}_i = E(x_i^\top \beta | D)$ are random variables for which the randomness comes from $\beta$ and training data $D$ ($D = (X_{\text{train}}, y_{\text{train}})$). We define the individual PGS accuracy as the correlation of an individual's genetic liability and PGS estimate with the following equation in consistence with classical theory[28,32,52]:

$$r_i^2(g_i, \hat{g}_i) = \frac{\text{cov}_{\beta,D}(g_i, \hat{g}_i)^2}{\text{var}_\beta(g_i)\text{var}_{\beta,D}(\hat{g}_i)} = 1 - \frac{E_D(\text{var}_{\beta|D}(x_i^\top \beta))}{\text{var}_\beta(x_i^\top \beta)} \tag{1}$$

Under an infinitesimal assumption for which all variants are causal and drawn from a normal distribution $N(0, \sigma_\beta^2)$, the analytical form of PGS accuracy can be derived as:

$$r_i^2(g_i, \hat{g}_i) = 1 - \frac{\sigma_e^2 \sum_{j=1}^J \frac{1}{\lambda_j} x_i^\top v_j v_j^\top x_i}{\sigma_\beta^2 x_i^\top x_i} = 1 - \frac{\sigma_e^2}{\sigma_\beta^2} \frac{\sum_{j=1}^J \frac{1}{\lambda_j} x_i^\top v_j v_j^\top x_i}{x_i^\top x_i}$$

in which $\sigma_\beta^2$ is per single nucleotide polymorphism (SNP) heritability; $\sigma_e^2$ is the variance of residual environmental noise; $v_j$ and $\lambda_j$ are the $j$th eigenvector and eigenvalues of training genotype data and $J$ is the total number of eigenvectors. $\sum_{j=1}^J \frac{1}{\lambda_j} x_i^\top v_j v_j^\top x_i$ is the squared Mahalanobis distance of the testing individual $i$ from the centre of the training genotype data on its principal component (PC) space; and $x_i^\top x_i$ is the sum of squared genotypes across all variants. Empirically, the ratio of the squared Mahalanobis distance to the sum of squared genotypes is highly correlated with the Euclidean distance of the individual from the training data on that PC space ($R = 1$, $P < 2.2 \times 10^{-16}$ in the UKBB). Given that this metric of accuracy is highly dependent on the GD from the training data, we term it the panel distance $r_i^2$. In practice, we use LDpred2 to estimate $E_D(\text{var}_{\beta|D}(x_i^\top \beta))$ (refs. 49,53) and approximate $\text{var}_\beta(x_i^\top \beta)$ as the heritability of the phenotype[30] (Methods). As a continuous GD, we use $d_i = \sqrt{\sum_{j=1}^J (x_i^\top v_j)^2}$ with $J$ set to 20 (Fig. 1c,d and Methods). We note two caveats of individual PGS accuracy: first, the genetic effects are assumed to be the same for all individuals regardless of their genetic ancestry background; second, the SNPs used for PGS training may not fully capture trait heritability. Therefore, the metric we proposed here is an upper bound of genetic prediction accuracy (Supplementary Note).

## PGS performance is calibrated in simulations

First, we evaluated calibration of the posterior variance of genetic liability $E_D(\text{var}_{\beta|D}(x_i^\top \beta))$ estimated by LDpred2 for individuals at various GDs from the UKBB WB training data by checking the calibration of the 90% credible intervals (Fig. 2a). We simulated 100 phenotypes at heritability $h_g^2 = 0.25$ and proportion of causal variants $p_{\text{causal}} = 1\%$

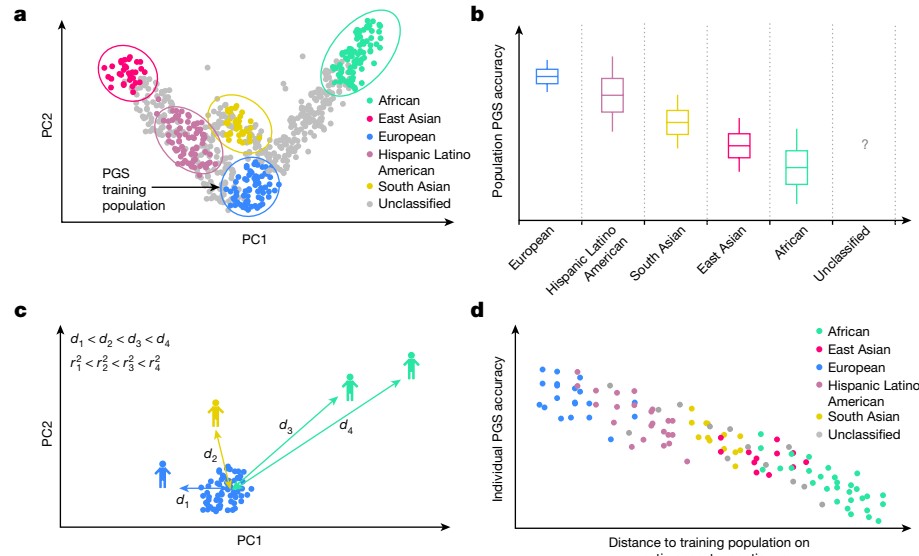

**Fig. 1 | Illustration of population-level versus individual-level PGS accuracy. a**, Discrete labelling of GIA with PCA-based clustering. Each dot represents an individual. The circles represent arbitrary boundaries imposed on the genetic ancestry continuum to divide individuals into different GIA clusters. The colour represents the GIA cluster label. The grey dots are individuals who are left unclassified. **b**, Schematic illustrating the variation of population-level PGS accuracy across clusters. The box plot represents the PGS accuracy (for example, $R^2$) measured at the population level. The question mark emphasizes that the PGS accuracy for unclassified individuals is unknown owing to the lack of a reference group. Grey dashed lines emphasize the categorical nature of GIA clustering. **c**, Continuous labelling of everyone's unique position on the genetic ancestry continuum with a PCA-based GD. The GD is defined as the Euclidean distance of an individual's genotype from the centre of the training data when projected on the PC space of training genotype data. Everyone has their own unique GD, $d_i$, and individual PGS accuracy, $r_i^2$. **d**, Individual-level PGS accuracy decays along the genetic ancestry continuum. Each dot represents an individual and its colour represents the assigned GIA label. Individuals labelled with the same ancestry spread out on the genetic ancestry continuum, and there are no clear boundaries between GIA clusters. This figure is illustrative and does not involve any real or simulated data.

for all individuals in the UKBB, assuming shared causal variants and homogeneous causal effect sizes for individuals from various genetic backgrounds (see Methods). Overall, the 90% credible intervals are approximately well calibrated (that is, the 90% credible interval overlaps with the true genetic liability across 90 of 100 replicates, for all individuals, regardless of their GD from the training population or GIA labels; Fig. 2a). For example, when individuals are binned into 10 deciles based on their GD from the training population, the average empirical coverage of the 90% credible intervals is 89.7% (s.d. 2.6%) for individuals from the closest decile (composed of 96.9% individuals labelled as WB, 3.1% labelled as PL under a discrete view of ancestries; see detailed naming convention in Methods) compared to the average empirical coverage of 82.4% (s.d. 4.6%) for individuals from the furthest decile (composed of 19.9% individuals labelled as CB and 80.1% labelled as NG).

Next, we investigated the impact of GD on individual-level PGS accuracy. As expected, the width of the credible interval increases linearly with GD, reflecting reduced predictive accuracy for the PGS (Fig. 2b). The average width of the 90% credible interval is 1.83 in the furthest decile of GD, a 1.8-fold increase over the average width in the closest decile of GD. In contrast to the credible interval width, the individual-level PGS accuracy $\widehat{r_i^2}$ decreases with GD from the training data (Fig. 2c); the average estimated accuracy of individuals in the closest decile GD is fourfold higher than that of individuals in the furthest decile. Even among the most homogeneous grouping of individuals traditionally labelled as WB, we observe a 5% relative decrease in accuracy for individuals at the furthest decile of GD as compared to those in the closest decile. Similar results are observed when using a population-level PGS metric of accuracy, albeit at the expense of binning individuals according to GD; we find a high degree of concordance between the average $\widehat{r_i^2}$ within the bin and the population-level $R^2$ estimated within the bin (Fig. 2d and Extended Data Fig. 1a). Similarly, we observe a high consistency between

average $\widehat{r_i^2}$ and squared correlation between PGS and simulated phenotypes ($R = 0.86$, $P < 10^{-10}$; Extended Data Fig. 1b). Taken together, our results show that the 90% credible intervals remain calibrated for individuals that are genetically distant from the training population at the expense of wider credible intervals, and $\widehat{r_i^2}$ captures the PGS accuracy decay across GD.

To demonstrate that the continuous accuracy decay is not specific to PGS models trained on European ancestries, we conducted further analyses using a non-European training dataset composed of individuals of NG and CB GIAs (we grouped the two GIAs to attain sufficient sample size for simulations). We simulated a high signal-to-noise trait by setting $h_g^2 = 0.8$ and proportion of causal variants $p_{causal} = 1\%$ and 0.1% with 56,539 SNPs on chromosome 10 alone. We trained PGS models on 5,000 individuals from the NG and CB GIA clusters and applied the models to the remaining testing individuals. The coverage of the 90% credible intervals was invariant to GD despite slight miscalibration. The 90% credible interval width increased and individual PGS accuracy decreased when the testing individual was further away from the training data. This trend is consistent with the observed decrease in empirical accuracy computed as squared correlation between PGS and genetic value as GD increases (Extended Data Figs. 2 and 3).

We further evaluated the impact of the number of PCs used for calculating GD on its ability to capture accuracy decay. We varied the number of PCs ($J$) from 1 to 20 and observed that the correlation between GD and individual accuracy ($-\text{cor}(d_i, r_i^2(g_i, \hat{g}_i))$) increases when more PCs are used for computing GD, but no further improvement is observed when $J > 15$ for any GIA clusters or the whole biobank (Extended Data Fig. 4). Therefore, we set $J = 20$ for simplicity. We also explored average squared genetic relationship from training data as an alternative metric of GD and found that it is a better prediction of accuracy decay within each GIA clusters (Extended Data Fig. 4). However, because this metric relies on individual-level training data that

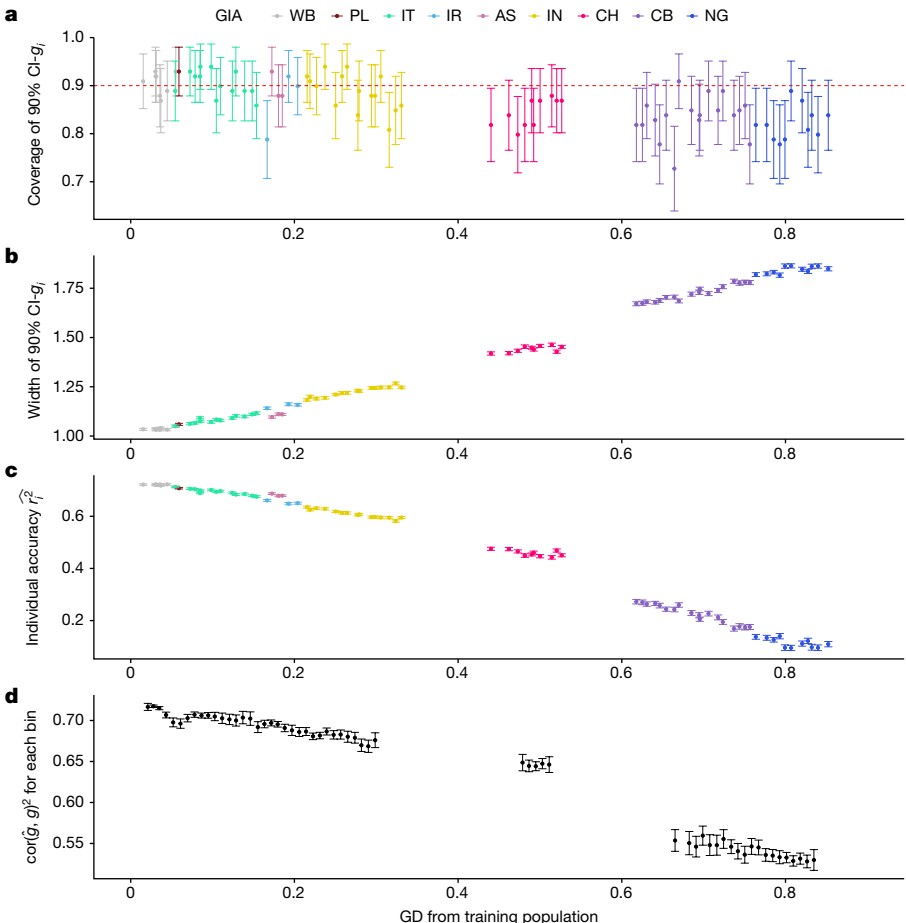

**Fig. 2 | PGS performance is calibrated across GD in simulations using UKBB data. a**, The 90% credible intervals of genetic liability (CI-$g_i$) are well calibrated for testing individuals at all GDs. The red dashed line represents the expected coverage of the 90% CI-$g_i$. Each dot represents a randomly selected UKBB testing individual. For each dot, the *x*-axis is its GD from the training data, the *y*-axis is the empirical coverage of the 90% CI-$g_i$ calculated as the proportion of simulation replicates for which the 90% CI-$g_i$ contain the individual's true genetic liability, and the error bars represent the mean ±1.96 standard error of the mean (s.e.m.) of the empirical coverage calculated from 100 simulations. **b**, The width of the 90% CI-$g_i$ increases with GD. For each dot, the *y*-axis is the average width of the 90% CI-$g_i$ across 100 simulation replicates, and the error bars represent ±1.96 s.e.m. **c**, Individual PGS accuracy decreases with GD. For each dot, the *y*-axis is the average individual-level PGS accuracy across 100 simulation replicates, and the error bars represent ±1.96 s.e.m. **d**, Population-level metrics of PGS accuracy recapitulates the decay in PGS accuracy across the genetic continuum. All UKBB testing individuals are divided into 100 equal-interval bins based on their GD. The *x*-axis is the average GD for the bin, and the *y*-axis is the squared correlation between genetic liability and PGS estimates for the individuals within the bin. The dot and error bars represent the mean and ±1.96 s.e.m from 100 simulations, respectively.

are usually not available, we choose to use PCA-based GD for convenience.

## PGS accuracy varies across the genetic continuum

Having validated our approach in simulations, we next turn to empirical data. For illustration purposes, we use height as an example, focusing on the ATLAS biobank as the target population with PGS trained on the 371,018 WB individuals from the UKBB (Methods); other traits show similar trends and are presented in the next sections. PGS accuracy at the individual level varies with GD across the entire biobank as well as within each GIA cluster (Fig. 3 and Extended Data Fig. 5). For example, GD strongly correlates with PGS accuracy of individuals in the GIA cluster labelled as Hispanic Latino American (HL, $R = -0.84$) and African American (AA, $R = -0.88$) in ATLAS. Notably, GD correlates with PGS accuracy even in non-admixed GIA clusters with correlations as $-0.66$, $-0.66$ and $-0.35$, for European American (EA), South Asian American (SAA) or East Asian American (EAA) GIA clusters, respectively. Similar qualitative results are also observed when applying PGS to test data from the UKBB; significant negative correlations are found between GD and individual PGS accuracy in all of the subcontinental GIA clusters

in the UKBB (Extended Data Fig. 5), with correlation coefficients ranging from $R = -0.031$ for the WB cluster to $R = -0.62$ for the CB cluster.

Next, we focused on the impact of GD on PGS accuracy across all ATLAS individuals regardless of GIA clustering ($R = -0.96$, $P < 10^{-10}$; Fig. 3b). Notably, we find a strong overlap of PGS accuracies across individuals from different GIA clusters demonstrating the limitation of using a single cluster-specific metric of accuracy. For example, when rank-ordering by GD, we find that the individuals from the closest GD decile in the HL cluster have similar estimated accuracy to the individuals from the furthest GD decile in EA cluster (average $\widehat{r_i^2}$ of 0.71 versus 0.71). This shows that GD enables identification of HL individuals with similar PGS performance to the EA cluster thus partly alleviating inequities due to limited access to accurate PGS. Most notably, GD can be used to evaluate PGS performance for individuals that cannot be easily clustered by current genetic inference methods (6% of ATLAS; Fig. 3b) partly owing to limitations of reference panels and algorithms for assigning ancestries. Among this traditionally overlooked group of individuals, we find the GD ranging from 0.02 to 0.64 and their corresponding estimated PGS accuracy $\widehat{r_i^2}$ ranging from 0.63 to 0.21. In addition to evaluating PGS accuracy with respect to the genetic liability, we also evaluated accuracy with respect to the

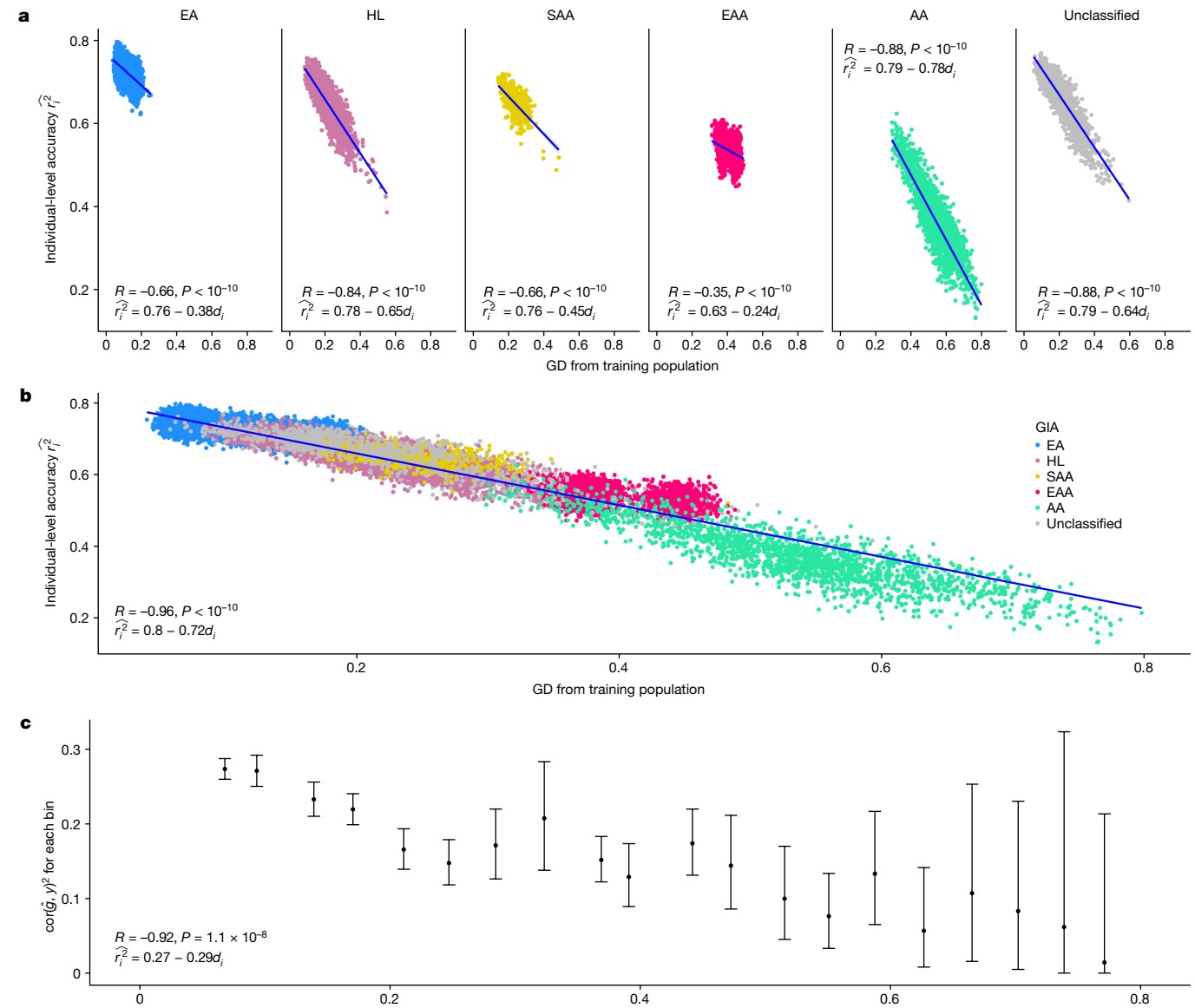

**Fig. 3 | The individual-level accuracy for height PGS decreases across the genetic ancestry continuum in ATLAS. a**, Individual PGS accuracy decreases within both homogeneous and admixed genetic GIA clusters. Each dot represents a testing individual from ATLAS. For each dot, the x-axis represents its distance from the training population on the genetic continuum; the y-axis represents its PGS accuracy. The colour represents the GIA cluster. **b**, Individual PGS accuracy decreases across the entire ATLAS. **c**, Population-level PGS accuracy decreases with the average GD in each GD bin. All ATLAS individuals are divided into 20 equal-interval GD bins. The x axis is the average

GD within the bin, and the y axis is the squared correlation between PGS and phenotype for individuals in the bin; the dot and error bar show the mean and 95% confidence interval from 1,000 bootstrap samples. R and P refer to the correlation between GD and PGS accuracy and its significance, respectively, from two-sided Pearson correlation tests without adjustment for multiple hypothesis testing. Any P value below $10^{-10}$ is shown as $P < 10^{-10}$. EA, European American; HL, Hispanic Latino American; SAA, South Asian American; EAA, East Asian American; AA, African American.

residual height after regressing out sex, age and PC1–10 on the ATLAS from the actual measured trait. Using equally spaced bins across the GD continuum, we find that correlation between PGS and the measured height tracks significantly with GD ($R = -0.92$, $P = 1.1 \times 10^{-8}$; Fig. 3c).

## PGS accuracy decay is pervasive

Having established the coupling of GD with PGS accuracy in simulations and for height, we next investigate whether this relationship is common across complex traits using PGSs for a broad set of 84 traits (Supplementary Table 1). We find consistent and pervasive correlations of GD with PGS accuracy across all considered traits in both ATLAS and the UKBB (Fig. 4). For example, the correlations between GD and individual

PGS accuracy range from −0.71 to −0.97 with an average of −0.95 across the 84 PGSs in ATLAS with similar results observed in the UKBB. Traits with sparser genetic architectures and fewer non-zero weights in the PGS have a lower correlation between GD and PGS accuracy; we reason that this is because GD represents genome-wide genetic variation patterns that may not reflect a limited number of causal SNPs well. For example, PGS for lipoprotein A (log_lipoA) has the lowest estimated polygenicity (0.02%) among the 84 traits and has the lowest correlation in ATLAS (−0.71) and the UKBB (−0.85). By contrast, we observe a high correlation between GD and PGS accuracy (>0.9) for all traits with an estimated polygenicity >0.1%. Next, we show that the fine-scale population structure accountable for the individual PGS accuracy variation is also prevalent within the traditionally defined genetic ancestry group.

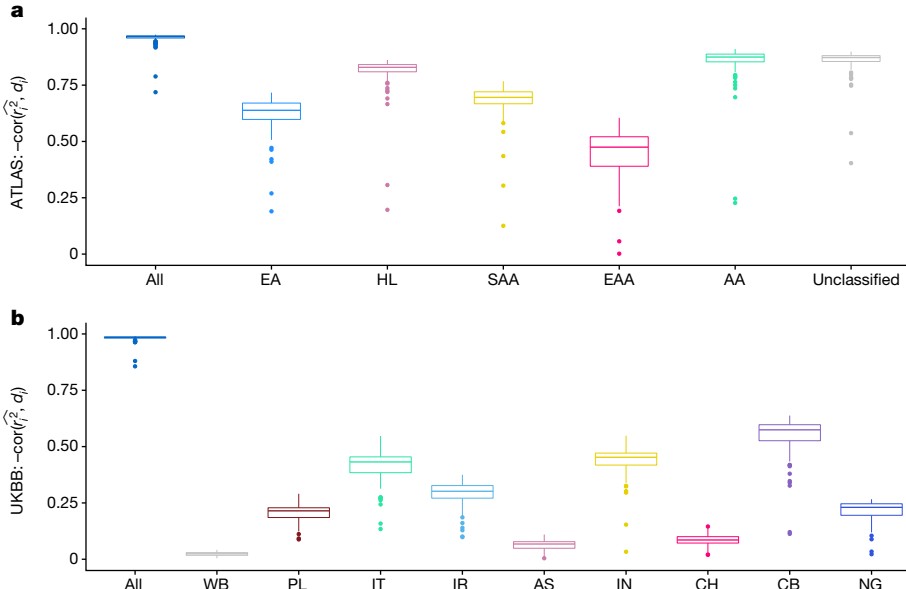

**Fig. 4 | The correlation between individual PGS accuracy and GD is pervasive across 84 traits across ATLAS and the UKBB. a**, The distribution of correlation between individual PGS accuracy and GD for 84 traits in ATLAS. **b**, The distribution of correlation between individual PGS accuracy and GD for 84 traits in the UKBB. Each box plot contains 84 points corresponding to the correlation between PGS accuracy and GD within the GIA group specified by the *x*-axis for each of the 84 traits. The box shows the first, second and third quartiles of the 84 correlations, and the whiskers extend to the minimum and maximum estimates located within 1.5 × IQR from the first and third quartiles, respectively. Numerical results are reported in Supplementary Tables 2 and 3.

For example, in ATLAS we find that 501 of 504 (84 traits across 6 GIA clusters) trait–ancestry pairs have a significant associations between GD and individual PGS accuracy after Bonferroni correction. In the UKBB, we find 572 of the 756 (84 traits across 9 subcontinental GIA clusters) trait–ancestry pairs have significant associations between GD and PGS accuracy after Bonferroni correction. We also find that a more stringent definition of homogeneous GIA clusters results in a lower correlation magnitude (Extended Data Fig. 6). Empirical analyses of PGS accuracy show a similar trend. When averaging across 84 traits, we find that the empirical accuracy decreases with increased GD across GIA clusters as reported by previous studies[39]. Further analyses based on GD bins show the decreasing trend at a finer scale (Extended Data Fig. 7).

## PGS varies across the genetic continuum

We have focused so far on investigating the relationship between GD ($d_i$) and PGS accuracy ($\widehat{r_i^2}$). Next, we evaluate the impact of GD on PGS estimates ($\hat{g}_i$) themselves. We find a significant correlation between GD and PGS estimates for 82 of 84 traits, with correlation coefficients ranging from $R = -0.52$ to $R = 0.74$ (Extended Data Fig. 8); this broad range of correlations is in stark contrast with the consistently observed negative correlation between GD and PGS accuracy. To better understand whether the coupling of PGS with GD is due to stratification or true signal, we compared the correlation of GD with PGS estimates ($\mathrm{cor}(d_i, \hat{g}_i)$) to the correlation of GD with measured phenotype values ($\mathrm{cor}(d_i, y_i)$). We find a wide range of couplings reflecting trait-specific signals; for 30 traits, GD correlates in opposite directions with PGS versus phenotype; for 40 traits, GD correlates in the same directions with PGS versus phenotype but differs in correlation magnitudes (Extended Data Fig. 8). For example, GD shows opposite and significantly different correlations for PGS versus trait for years of education (years_of_edu, $\mathrm{cor}(y_i, d_i) = 0.03$, $\mathrm{cor}(\hat{g}_i, d_i) = -0.18$). Other traits, such as hair colour, show a highly consistent impact of GD on PGS versus trait (darker_hair, $\mathrm{cor}(y_i, d_i) = 0.59$, $\mathrm{cor}(\hat{g}_i, d_i) = 0.74$), whereas for monocyte percentage, GD shows different magnitudes albeit with the same directions (monocyte_perc, $\mathrm{cor}(y_i, d_i) = -0.03$, $\mathrm{cor}(\hat{g}_i, d_i) = -0.52$).

Moreover, GD correlates with PGS and phenotype even within the same GIA cluster and the correlation patterns vary across clusters (Extended Data Fig. 9).

The correlation of GD with phenotype and PGS is also observed in ATLAS. For example, both height phenotype and height PGS vary along GD in ATLAS (Fig. 5); this holds true even when restricting analysis to the EA genetic ancestry cluster (Supplementary Fig. 1). This is consistent with genetic liability driving difference in phenotypes but could also be explained by residual population stratification. For neutrophil counts, phenotype and PGS vary in opposite directions with respect to GD across the ATLAS (Fig. 5), although the trend is similar for phenotype and PGS in the EA GIA clusters (Supplementary Fig. 1). This could be explained by genetic liability driving signal in Europeans with stratification for other groups. Neutrophil counts have been reported to vary greatly across ancestry groups with reduced counts in individuals of African ancestries[54]. In ATLAS, we observe a negative correlation (−0.04) between GD and neutrophil counts in agreement with the previous reports, whereas GD is positively correlated (0.08) with PGS estimates—genetically distant individuals traditionally labelled as African American having higher PGS than average. The opposite directions in phenotype–distance and PGS–distance correlations are partly attributed to the Duffy-null SNP rs2814778 on chromosome 1q23.2. This variant is strongly associated with neutrophil counts among individuals traditionally identified as African ancestry, but it is rare and excluded in our training data. This exemplifies the potential bias in PGS due to non-shared causal variants and emphasizes ancestral diversity in genetic studies. As PGS can vary across GD either as a reflection of true signal (that is, genetic liability varying with ancestry) or owing to biases in PGS estimation ranging from unaccounted residual population stratification to incomplete data (for example, partial ancestry-specific tagging of causal effects), our results emphasize the need to consider GD in PGS interpretation beyond adjusting for PGS $r_i^2$.

## Discussion

In this work, we have shown that PGS accuracy varies from individual to individual and proposed an approach to personalize PGS metrics of

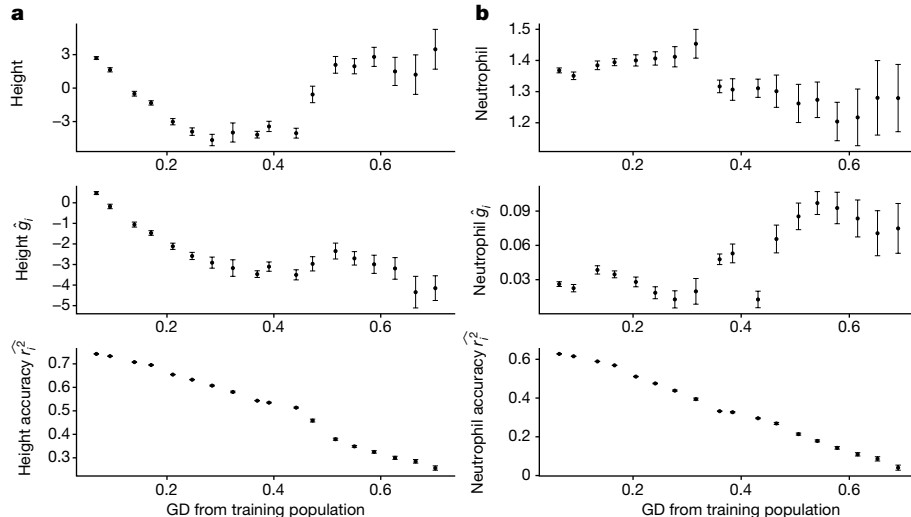

**Fig. 5 | Measured phenotype, PGS estimates and accuracy vary across ATLAS. a**, Variation of height phenotype, PGS estimates and accuracy across different GD bins in ATLAS. **b**, Variation of log neutrophil count phenotype, PGS estimates and accuracy across different GD bins in ATLAS. The 36,778 ATLAS individuals are divided into 20 equal-interval GD bins. Bins with fewer than 50 individuals are not shown owing to large s.e.m. All panels share the same layout: the $x$ axis is the average GD within the bin; the $y$ axis is the average phenotype (top), PGS (middle) and individual PGS accuracy (bottom); the error bars represent ±1.96 s.e.m.

performance. We used a PCA-based GD[39] from the centre of training data to describe an individual's unique location on the genetic ancestry continuum and showed that individual PGS accuracy tracks well with GD. The continuous decay of PGS performance as the target individual becomes further away from the training population is pervasive across traits and ancestries. We highlight the variability in PGS performance along the continuum of genetic ancestries, even within traditionally defined homogeneous populations. As the genetic ancestries are increasingly recognized as continuous rather than discrete[7,42–45], the individual-level PGS accuracy provides a powerful tool to study PGS performance across diverse individuals to enhance the utility of PGS. For example, by using individual-level PGS accuracy, we can identify individuals from Hispanic Latino GIA who have similar PGS accuracy to individuals of European GIA, thus partly alleviating inequities due to lack of access to accurate PGS.

Simulation and real-data analyses show that individual PGS accuracy is highly correlated with GD, in alignment with existing works showing that decreased similarity (measured by relatedness, linkage disequilibrium and/or minor allele frequency differences, fixation index ($F_{st}$) and so on)[41,55] between testing individuals and training data is a major contributor to PGS accuracy decay. However, practical factors that may affect transferability, such as genotype–environment interaction and population-specific causal variants, are not modelled in the calculation of individual PGS accuracy and this is left for future work.

Our results emphasize the importance of PGS training in diverse ancestries[56] as it can provide advantages for all individuals. Broadening PGS training beyond European ancestries can lead to improved accuracy in genetic effect estimation particularly for variants with higher frequencies in non-European data. It can also increase PGS portability by reducing the GD from target to training data. However, increased diversity may also bring challenges to statistical modelling; for example, differences in genetic effects may correlate with environment factors and could bias genetic risk prediction. To address these challenges, more sophisticated statistical methods are needed that can effectively leverage ancestrally diverse populations to train PGS[3] (for example, PRS-CSx[57], vilma[58] and CT-SLEB[59]). Concerted global effort and equitable collaborations are also crucial to increase the sample size of underrepresented individuals as part of an effort to reduce health disparities across ancestries[56,60].

We highlight the pervasive correlation between PGS estimates and GD of varying magnitude and sign as compared to the correlation between phenotype and GD. This provides a finer resolution of the mean shift of PGS estimates across genetic ancestry groupings[38]. The correlation between GD and PGS estimates can arise from bias and/or true biological difference, and more effort is needed to investigate the PGS bias in the context of genetic ancestry continuum.

We note several limitations and future directions of our work. First, our proposed individual PGS accuracy is an upper bound of true accuracy and should be interpreted only in terms of the additive heritability captured by SNPs included in the model. Missing heritability[61,62] and misspecification of the heritability model along with population-specific causal variants and effect sizes may further decrease real accuracy. For example, the prediction accuracy for neutrophil count is overestimated among African American individuals because the Duffy-null SNP rs2814778 (ref. 54) is not captured in the UKBB WB training data. Future work could investigate the impact of the population-specific components of genetic architecture on the calibration of PGS accuracy. Second, we approximate the variance of genetic liability in the denominator of equation (1) with heritability and set a fixed value for all individuals. Preliminary results show that replacing the denominator with a Monte Carlo estimation of genetic liability variance recapitulates the accuracy decay in estimated PGS accuracy, albeit the correlation is slightly reduced (Extended Data Fig. 10). Third, individual PGS accuracy evaluates how well the PGS estimates the genetic liability instead of phenotype. Quantifying the individual accuracy of PGS with respect to phenotype can be achieved by also modelling non-genetic factors for proper calibration. Fourth, limited by sample size, we combined GIA groups as a training set in simulation experiments to replicate PGS accuracy decay; this is not an optimal strategy for data analysis as the population structure in the training data may confound the true genetic effects and reduce prediction accuracy. We leave a more comprehensive investigation of non-European PGS training data for future work. Sixth, although we advocate for the use of continuous genetic ancestry, we trained our PGS models on a discrete GIA cluster of WB because current PGS methods rely on discrete genetic ancestry groupings. We leave the development of PGS training methods that are capable of modelling continuous ancestries as future work. Finally, we highlight that, just like PGS, the traditional clinical risk assessment

may suffer from limited portability across diverse populations[18]. For examples, the pooled cohort equation overestimates atherosclerotic cardiovascular disease risk among non-European populations[16]; and a traditional clinical breast cancer risk model developed in the European population in the USA overestimated the breast cancer risk among older Korean women[17]. Here we focus on genetic prediction potability owing to the wide interest and attention from both the research community and society. We emphasize that improving the portability of traditional clinical risk factor models in diverse populations is an essential component of health equity and requires thorough investigation.

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

## Methods

### Model

We model the phenotype of an individual with a standard linear model $y_i = x_i^\top \beta + \epsilon_i$, in which $x_i$ is an $M \times 1$ vector of standardized genotypes (centred and standardized with respect to the allele frequency in the training population for both training and testing individuals), $\beta$ is an $M \times 1$ vector of standardized genetic effects, and $\epsilon_i$ is random noise. Under a random effects model, $\beta$ is a vector of random variable sampled from a prior distribution $p(\beta)$ that differs under different genetic architecture assumptions[62] and PGS methods[53,63–65]. The PGS weights $\hat{\beta} = E_{\beta|D}(\beta)$ are estimated to be the posterior mean given the observed data $D$ ($D = (X_{\text{train}}, y_{\text{train}})$ with access to individual-level genotype, $X_{\text{train}}$, and phenotype, $y_{\text{train}}$; or $D = (\hat{\beta}_{\text{GWAS}}, \hat{R})$ with access to marginal association statistics $\hat{\beta}_{\text{GWAS}}$ and LD matrix $\hat{R}$, in which GWAS stands for genome-wide association study). The genetic liability ($g_i = x_i^\top \beta$) of an individual $i$ is estimated to be $\hat{g}_i = E_{\beta|D}(x_i^\top \beta)$, the uncertainty of which is estimated as the posterior variance of genetic liability $\text{var}(\hat{g}_i) = \text{var}_{\beta|D}(x_i^\top \beta)$ (ref. 49).

### Individual PGS accuracy

We define individual PGS accuracy as the squared correlation between an individual's genetic liability, $g_i$, and its PGS estimate, $\hat{g}_i$, following the general form in ref. 28:

$$r_i^2 = \frac{\text{cov}_{\beta,D}(g_i, \hat{g}_i)^2}{\text{var}_{\beta,D}(g_i)\text{var}_{\beta,D}(\hat{g}_i)} = \frac{\text{var}_D(x_i^\top \hat{\beta})^2}{\text{var}_\beta(x_i^\top \beta)\text{var}_D(x_i^\top \hat{\beta})}$$

Here we are interested in the PGS accuracy of a given individual; therefore, the genotype is treated as a fixed variable, and genetic effects are treated as a random variable. We note that a random effects model is essential; otherwise, $\text{cov}_{\beta,D}(g_i, \hat{g}_i)$ and $\text{var}_{\beta,D}(g_i)$ are 0. Under a random effects model, both the genetic liability and PGS estimate for individual $i$ are random variables. The randomness of $g_i = x_i^\top \beta$ comes from the randomness in $\beta$, and the randomness of $\hat{g}_i = x_i \hat{\beta}$ comes from the randomness of both $\beta$ and the training data $D$. Individual PGS accuracy measures the correlation between $g_i$ and $\hat{g}_i$, which can be computed with the following equation:

$$r_i^2 = 1 - \frac{E_D(\text{var}_{\beta|D}(x_i^\top \beta))}{\text{var}_\beta(x_i^\top \beta)} \tag{2}$$

in which $\text{var}_{\beta|D}(x_i^\top \beta)$ is the posterior variance of genetic liability given the training data, and $\text{var}_\beta(x_i^\top \beta)$ is the genetic variance. The equation is derived as follows.

First, we show that under the random effects model, $\text{cov}_{\beta,D}(x_i^\top \hat{\beta}, x_i^\top \beta) = \text{var}_D(x_i^\top \hat{\beta})$ (in which $\hat{\beta} = E_{\beta|D}(\beta)$) following equation 5.149 in ref. 66:

$$\begin{aligned}
\text{cov}_{\beta,D}(\hat{\beta}, \beta^\top) &= E_{\beta,D}(\hat{\beta}\beta^\top) - E_{\beta,D}(\hat{\beta})E_{\beta,D}(\beta^\top) \\
&= E_D(E_{\beta|D}(\hat{\beta}\beta^\top)) - E_{D,\beta}(\hat{\beta})E_D(E_{\beta|D}(\beta^\top)) \\
&= E_D(E_{\beta|D}(E_{\beta|D}(\beta)\beta^\top)) \\
&\quad - E_D(E_{\beta|D}(\beta))E_D(E_{\beta|D}(\beta^\top)) \\
&= E_D(E_{\beta|D}(\beta)E_{\beta|D}(\beta^\top)) \\
&\quad - E_D(E_{\beta|D}(\beta))E_D(E_{\beta|D}(\beta^\top)) \\
&= \text{var}_D(E_{\beta|D}(\beta)) \\
&= \text{var}_D(\hat{\beta})
\end{aligned}$$

Multiplying $x_i$ on both sides of the equation, we obtain:

$$x_i^T \text{cov}_{\beta,D}(\hat{\beta}, \beta)x_i = x_i^T \text{var}_D(\hat{\beta})x_i$$

$$\text{cov}_{\beta,D}(x_i^\top \hat{\beta}, x_i^\top \beta) = \text{var}_D(x_i^\top \hat{\beta}) \tag{3}$$

Equation (3) also implies the slope from regression of observed phenotypic values (or true genetic liability) on the estimated PGS equal to 1 (Supplementary Fig. 2), which offers an alternative way to assess the calibration of PGS as done in refs. 64,67.

$$\text{slope} = \frac{\text{cov}(x_i^\top \hat{\beta}, y_i)}{\text{var}(x_i^\top \hat{\beta})} = \frac{\text{cov}(x_i^\top \hat{\beta}, x_i^\top \beta + \epsilon_i)}{\text{var}(x_i^\top \hat{\beta})} = \frac{\text{var}(x_i^\top \hat{\beta})}{\text{var}(x_i^\top \hat{\beta})} = 1$$

Next, by applying the law of total variance, we show that:

$$\text{var}_{\beta,D}(g_i) = \text{var}_{\beta,D}(x_i^T \beta) = E_D(\text{var}_{\beta|D}(x_i^T \beta)) + \text{var}_D(E_{\beta|D}(x_i^T \beta))$$

$$\text{var}_D(x_i^\top \hat{\beta}) = \text{var}_{\beta,D}(x_i^\top \beta) - E_D(\text{var}_{\beta|D}(x_i^\top \beta)) \tag{4}$$

Third, we derive the correlation between $g_i$ and $\hat{g}_i$ as:

$$\begin{aligned}
r_i^2 &= \frac{\text{cov}_{\beta,D}(g_i, \hat{g}_i)^2}{\text{var}_{\beta,D}(g_i)\text{var}_{\beta,D}(\hat{g}_i)} \\
&= \frac{\text{var}_D(x_i^\top \hat{\beta})^2}{\text{var}_\beta(x_i^\top \beta)\text{var}_D(x_i^\top \hat{\beta})} \text{ by applying equation (3)} \\
&= \frac{\text{var}_D(x_i^\top \hat{\beta})}{\text{var}_\beta(x_i^\top \beta)} \\
&= \frac{\text{var}_{\beta,D}(x_i^\top \beta) - E_D(\text{var}_{\beta|D}(x_i^\top \beta))}{\text{var}_\beta(x_i^\top \beta)} \text{ by applying equation (4)} \\
&= 1 - \frac{E_D(\text{var}_{\beta|D}(x_i^\top \beta))}{\text{var}_\beta(x_i^\top \beta)}
\end{aligned}$$

The above equation is widely used in animal breeding theory to compute the reliability of estimated breeding value for each individual[30]. In this work, we use individual PGS uncertainty $\text{var}(\hat{g}_i) = \text{var}_{\beta|D}(x_i^\top \beta)$ as an unbiased estimator of $E_D(\text{var}_{\beta|D}(x_i^\top \beta))$. We also use estimated heritability to approximate $\text{var}_\beta(x_i^\top \beta)$ in simulations in which the phenotype has unit variance. In real-data analysis, as the phenotype does not necessarily have unit variance, we approximate $\text{var}_\beta(x_i^\top \beta)$ by scaling the estimated heritability with the residual phenotypic variance in the training population after regressing GWAS covariates including sex, age and precomputed UKBB PC1–16 (Data-Field 22009).

### Analytical form of individual PGS accuracy under infinitesimal assumption

Without loss of generality, we assume a prior distribution of genetic effects as follows:

$$p(\beta|\sigma_\beta^2) = MVN(0, \sigma_\beta^2 I_M)$$

where $M$ is the number of genetic variants. With access to individual genotype, $X_{\text{train}}$, and phenotype, $y_{\text{train}}$, data, the likelihood of the data is

$$p(y_{\text{train}} \mid X_{\text{train}}, \beta, \sigma_e^2) = MVN(X_{\text{train}}\beta, \sigma_e^2 I_N)$$

where $N$ is the training sample size. The posterior distribution of genetic effects given the data is proportional to the product of the prior and the likelihood:

$$\begin{aligned}
p(\beta|X_{\text{train}}, y_{\text{train}}, \sigma_\beta^2, \sigma_e^2) &\propto p(\beta|\sigma_\beta^2)p(y_{\text{train}}|X_{\text{train}}, \beta, \sigma_e^2) \\
&\propto MVN(0, \sigma_\beta^2 I_M)MVN(X_{\text{train}}\beta, \sigma_e^2 I_N) \\
&\propto MVN(\mu_\beta, \sigma_\beta)
\end{aligned}$$

in which $\mu_\beta = \left(\frac{\sigma_e^2}{\sigma_\beta^2}I_M + X_{\text{train}}^\top X_{\text{train}}\right)^{-1} X_{\text{train}}^\top y_{\text{train}}$ and $\Sigma_\beta = \sigma_e^2\left(\frac{\sigma_e^2}{\sigma_\beta^2}I_M + X_{\text{train}}^\top X_{\text{train}}\right)^{-1}$.
This form is equivalent to the solution of random effects in the best linear unbiased prediction with the pedigree matrix or genetic relationship matrix[29,32].

For a new target individual, the posterior variance of the genetic liability is:

$$\text{var}(x_i^\top \beta | x_i, X_{\text{train}}, y_{\text{train}}, \sigma_\beta^2, \sigma_e^2) = x_i^\top \Sigma_\beta x_i = \sigma_e^2 x_i^\top \left(\frac{\sigma_e^2}{\sigma_\beta^2}I_M + X_{\text{train}}^\top X_{\text{train}}\right)^{-1} x_i$$

After carrying out eigendecomposition on $X_{\text{train}}^\top X_{\text{train}} = \sum_{j=1}^J \lambda_j v_j v_j^\top$, we can rewrite

$$\left(\frac{\sigma_e^2}{\sigma_\beta^2}I_M + X_{\text{train}}^\top X_{\text{train}}\right)^{-1} = \left(\frac{\sigma_e^2}{\sigma_\beta^2}I_M + \sum_{j=1}^J \lambda_j v_j v_j^\top\right)^{-1} = \sum_{j=1}^J \left(\frac{\sigma_e^2}{\sigma_\beta^2} + \lambda_j\right)^{-1} v_j v_j^\top$$

in which $\lambda_j$ and $v_j$ correspond to the $j$th eigenvalue and unit-length eigenvector of the training genotype, $X_{\text{train}}$.

Thus, we can rewrite the posterior variance of genetic liability as

$$\text{var}(x_i^\top \beta | x_i, X_{\text{train}}, y_{\text{train}}, \sigma_\beta^2, \sigma_e^2) = \sigma_e^2 \sum_{j=1}^J \left(\frac{\sigma_e^2}{\sigma_\beta^2} + \lambda_j\right)^{-1} x_i^\top v_j v_j^\top x_i$$

Replacing $E_D(\text{var}_{\beta|D}(x_i^\top \beta))$ in equation (2) with the analytical form of $\text{var}(x_i^\top \beta | x_i, X_{\text{train}}, y_{\text{train}}, \sigma_\beta^2, \sigma_e^2)$, we get

$$r_i^2 = 1 - \frac{\text{var}(x_i^\top \beta | x_i, X_{\text{train}}, y_{\text{train}}, \sigma_\beta^2, \sigma_e^2)}{\text{var}(x_i^\top \beta)} = 1 - \frac{\sigma_e^2 \sum_{j=1}^J \left(\frac{\sigma_e^2}{\sigma_\beta^2} + \lambda_j\right)^{-1} x_i^\top v_j v_j^\top x_i}{\sigma_\beta^2 x_i^\top x_i}$$

As the eigenvalue of $X_{\text{train}}^\top X_{\text{train}}$ increases linearly with training sample size $N$ (ref. [68]), at the UKBB-level sample size (for example, $N = 371,018$ for our UKBB WB training data), the eigenvalues for the top PCs are usually larger than the ratio of environmental noise variance and genetic variance $\frac{\sigma_e^2}{\sigma_\beta^2}$. Thus, we can further approximate the analytical form with:

$$r_i^2 = 1 - \frac{\sigma_e^2 \sum_{j=1}^J \frac{1}{\lambda_j} x_i^\top v_j v_j^\top x_i}{\sigma_\beta^2 x_i^\top x_i} = 1 - \frac{\sigma_e^2}{\sigma_\beta^2} \frac{\sum_{j=1}^J \frac{1}{\lambda_j} x_i^\top v_j v_j^\top x_i}{x_i^\top x_i}$$

The term $\sum_{j=1}^J \frac{1}{\lambda_j} x_i^\top v_j v_j^\top x_i$ is the squared Mahalanobis distance of the testing individual $i$ from the centre of the training genotype data on its PC space and $x_i^\top x_i$ is the sum of squared genotype across all variants. Empirically, the ratio between the two is highly correlated with the Euclidean distance of the individual from the training data on that PC space ($R = 1$, $P$ value $< 2.2 \times 10^{-16}$ in the UKBB).

## Genetic distance (GD)
The GD is defined as the Euclidean distance between a target individual and the centre of training data on the PC space of training data.

$$d_i = \sqrt{\sum_{j=1}^J (x_i^\top v_j - \bar{x}_{\text{train}} v_j)^2} = \sqrt{\sum_{j=1}^J (x_i^\top v_j)^2}$$

in which $d_i$ is the GD of a testing individual $i$ from the training data, $x_i$ is an $M \times 1$ standardized genotype vector for testing individual $i$, $v_j$ is the $j$th eigenvector for the genotype matrix of training individuals, $\bar{x}_{\text{train}}$ is the average genotype in the training population ($\bar{x}_{\text{train}} v_j = 0$ given that the genotypes are centred with respect to the allele frequency in the training population), and $J$ is set to 20.

## Ancestry ascertainment in UKBB
The UKBB individuals are clustered into nine subcontinental GIA clusters—WB (white British), PL (Poland), IR (Iran), IT (Italy), AS (Ashkenazi), IN (India), CH (China), CB (Caribbean) and NG (Nigeria)—based on the top 16 precomputed PCs (Data-Field 22009) as described in ref. 39. First, UKBB participants are grouped by country of origin (Data-Field 20115) and the centre of each country on the PC space is computed as the geometric median for all countries, which serves as a proxy for the centre for each subcontinental ancestry. The centre of Ashkenazi GIA is determined using a dataset from ref. 69. Second, we reassign each individual to one of the nine GIA groups on the basis of their Euclidean distance to the centres on the PC space, as the self-reported country of origin does not necessarily match an individual's genetic ancestry. The genetic ancestry of an individual is labelled as unknown if its distance to any genetic ancestry centre is larger than one-eighth of the maximum distance between any pairs of subcontinental ancestry clusters. We are able to cluster 91% of the UKBB participants into 411,018 WB, 4,127 PL, 1,169 IR, 6,499 IT, 2,352 AS, 1,798 CH, 2,472 CB and 3,894 NG. GIAs are not necessarily reflective of the full genetic diversity of a particular region but reflect only the diversity present in the UKBB individuals.

## Ancestry ascertainment in ATLAS
The ATLAS individuals are clustered into five GIA clusters—European Americans (EA), Hispanic Latino Americans (HL), South Asian Americans (SAA), East Asian Americans (ESA) and African Americans (AA)—as described in ref. 5 on the basis of their proximity to 1000 Genome super populations on the PC space. First, we filter the ATLAS-typed genotypes with plink2 by Mendel error rate (plink --me 1 1 --set-me-missing), founders (--filter-founders), minor allele frequency (--maf 0.15), genotype missing call rate (--geno 0.05) and Hardy–Weinberg equilibrium test $P$ value (--hwe 0.001). Next, ATLAS genotypes were merged with the 1000 Genomes phase 3 dataset. Then, linkage disequilibrium (LD) pruning was carried out on the merged dataset (--indep 200 5 1.15 --indep-pairwise 100 5 0.1). The top 10 PCs were computed with the flashpca2 (ref. 70) software with all default parameters. Next, we use the super population label and PCs of the 1000 Genome individuals to train the $K$-nearest neighbours model to assign genetic ancestry labels to each ATLAS individual. For each ancestry cluster, we run the $K$-nearest neighbours model on the pair of PCs that capture the most variation for each genetic ancestry group: the European, East Asian and African ancestry groups use PCs 1 and 2, the Admixed American group uses PCs 2 and 3, and the South Asian group uses PCs 4 and 5. In each analysis, we use tenfold cross-validation to select the $k$ hyper-parameter from $k = 5, 10, 15, 20$. If an individual is assigned to multiple ancestries with probability larger than 0.5 or is not assigned to any clusters, their ancestry is labelled as unknown. We label the five 1000 Genome super population as EA for Europeans, HL for Admixed Americans, SAA for South Asians, AA for Africans and ESA for East Asians. We can cluster 95% of the ATLAS participants into 22,380 EA, 6,973 HL, 625 SAA, 3,331 EAA and 1,995 AA, and the ancestry of 2,332 individuals is labelled as unknown.

## Genotype data
In simulations, we use 1,054,151 UKBB HapMap 3 SNPs for simulating phenotypes, training PGS models and calculating PGS for testing individuals in UKBB. For real-data analysis, we use an intersection of UKBB HapMap 3 SNPs and ATLAS imputed SNPs for the training of PGS in UKBB and calculating PGS for remaining UKBB individuals and ATLAS individuals. We start from 1,054,151 UKBB HapMap 3 SNPs and 8,048,268 ATLAS imputed SNPs. As UKBB is on genome build hg37 and ATLAS is on hg38, we first lift all ATLAS SNPs from hg38 to hg37 with the snp_modifyBuild function in the bigsnpr R package. Next, we match UKBB SNPs and ATLAS SNPs by chromosome and position with the snp_match function in bigsnpr. Then, we recode ATLAS SNPs

using UKBB reference alleles with the plink2 --recode flag. In the end, 979,457 SNPs remain for training the LDpred2 models in real-data analysis.

## Simulated phenotypes

We use simulations on all UKBB individuals to investigate the impact of GD from training data on the various metrics of PGS. We fix the proportion of causal SNPs $p_{causal} = 0.01$ and heritability as $h_g^2 = 0.25$. The simulated genetic effects and phenotype are generated as follows. First, we randomly sample

$$\beta_m \sim \begin{cases} N\left(0, \dfrac{h_g^2}{\text{var}(x_m)Mp_{causal}}\right) & c_j = 1, \text{ with probability } p_{causal} \\ 0 & c_j = 0, \text{ with probability } 1 - p_{causal} \end{cases}$$

in which $\text{var}(x_m)$ is the variance of allele counts for SNP $m$ among all UKBB individuals. Second, we compute the genetic liability for each individual as $g_i = \sum_{m=1}^{M} x_{im}\beta_m$ and randomly sample environmental noise $\epsilon_i \sim N(0, 1 - h_g^2)$. Third, we generate phenotype as $y_i = g_i + \epsilon_i$. We repeat the process 100 times to generate 100 sets of genetic liability and phenotypes.

## GD from PGS training data

To compute the GD of testing individuals from the training population, we carry out PCA on the 371,018 UKBB WB training individuals and project the 48,586 UKBB testing individuals and 36,778 ATLAS testing individuals on the PC space. We start from the 979,457 SNPs that are overlapped in UKBB and ATLAS. First, we carry out LD pruning with plink2 (--indep-pairwise 1000 50 0.05) and exclude the long-range LD regions. Next, we carry out PCA analysis with flashpca2 (ref. 70) on the 371,018 UKBB WB training individuals to obtain the top 20 PCs. Then, we project the remaining 48,586 UKBB individuals that are not included in the training data and 36,778 ATLAS individuals onto the PC space of training data by using SNP loadings (--outload loadings.txt) and their means and standard deviations (--outmeansd meansd.txt) output from flashpca2. In the end, we compute the GD for each individual as the Euclidean distance of their PCs from the centre of training data with the equation $d_i = \sqrt{\sum_{j=1}^{20} (pc_{ij})^2}$, in which $pc_{ij}$ is the $j_{th}$ PC of individual $i$.

## LDpred2 PGS model training

The PGS models were trained on 371,018 UKBB individuals labelled as WB with the LDpred2 (ref. 53) method for both simulation and real-data analysis. For simulation analysis, we use 1,054,151 UKBB HapMap 3 variants. For real-data analysis, we use 979,457 SNPs that are overlapped in UKBB HapMap 3 variants and ATLAS imputed genotypes.

First, we obtain GWAS summary statistics by carrying out GWAS on the training individuals with plink2 using sex, age and precomputed PC1–16 as covariates. Second, we calculate the in-sample LD matrix with the function snp_cor from the R package bigsnpr[71]. Next, we use the GWAS summary statistics and LD matrix as input for the snp_ldpred2_auto function in bigsnpr to sample from the posterior distribution of genetic effect sizes. Instead of using a held-out validation dataset to select hyperparameters $p$ (proportion of causal variants) and $h2$ (heritability), snp_ldpred2_auto estimates the two parameters from data with the Markov chain Monte Carlo (MCMC) method directly. We run 10 chains with different initial sparsity $p$ from $10^{-4}$ to 1 equally spaced in log space. For all chains, we set the initial heritability as the LD score regression heritability[72] estimated by the built-in function snp_ldsc. We carry out quality control of the 10 chains by filtering out chains with estimated heritability that is smaller than 0.7 times the median heritability of the 10 chains or with estimated sparsity that is smaller than 0.5 times the median sparsity or larger than 2 times the median sparsity. For each chain that passes filtering, we remove the first 100 MCMC iterations as burn-in and thin the next 500 iterations by selecting every fifth iteration to reduce autocorrelation between MCMC samples. In the end, we obtain an $M \times B$ matrix $[\tilde{\beta}^{(1)}, \tilde{\beta}^{(2)}, ..., \tilde{\beta}^{(B)}]$, in which each column of the matrix $\tilde{\beta}^{(b)}$ is a sample of posterior causal effects of the $M$ SNPs. Owing to the quality control of MCMC chains, the total number of posterior samples $B$ ranges from 500 to 1,000.

## Calculation of PGS and accuracy

We use the score function in plink2 to compute the PGS for 48,586 and 36,778 testing individuals in UKBB and ATLAS, respectively. For each $\tilde{\beta}^{(b)}$, we compute the PGS for each individual $i$ as $x_i^\top \tilde{\beta}^{(b)}$ with plink2 (--score). For each individual with genotype $x_i$, we compute $x_i^\top \tilde{\beta}^{(1)}, x_i^\top \tilde{\beta}^{(2)}, ..., x_i^\top \tilde{\beta}^{(B)}$ to approximate its posterior distribution of genetic liability. The genotype $x_i^\top$ is centred to the average allele count (--read-freq) in training data to reduce the uncertainty from the unmodelled intercept. We estimate the PGS with the posterior mean of the genetic liability as $\hat{g}_i = E_{\beta|D}(x_i^\top \beta) = \frac{1}{B}\sum_{b=1}^{B} x_i^\top \tilde{\beta}^{(b)}$. We estimate the individual-level PGS uncertainty as $\text{var}(\hat{g}_i) = \text{var}_{\beta|D}(x_i^\top \beta) = \frac{1}{B}\sum_{b=1}^{B}(x_i^\top \tilde{\beta}^{(b)} - \hat{g}_i)^2$. The individual-level PGS accuracy is calculated as $\widehat{r_i^2} = 1 - \frac{\text{var}(\hat{g}_i)}{h_g^2}$ for simulation ($h_g^2$ is the heritability estimated by the LDpred2 model) and as $\widehat{r_i^2} = 1 - \frac{\text{var}(\hat{g}_i)}{h_g^2 \text{var}(y_{train} - \hat{y}_{train})}$ for real-data analysis, in which $\text{var}(y_{train} - \hat{y}_{train})$ is the variance of residual phenotype in training data after regressing out GWAS covariates.

## Calibration of credible interval in simulation

We run the LDpred2 model on 371,018 WB training individuals for the 100 simulation replicates. In each simulation, for individual with genotype $x_i$, we compute $x_i^\top \tilde{\beta}_r^{(1)}, x_i^\top \tilde{\beta}_r^{(2)}, ..., x_i^\top \tilde{\beta}_r^{(B)}$ to approximate their posterior distribution of genetic liability, generate a 90% credible interval CI-$g_{ir}$ (90% credible interval of genetic liability of $i_{th}$ individual in $r_{th}$ replication) with 5% and 95% quantile of the distribution and check whether their genetic liability is contained in the credible interval $I(g_{ir} \in \text{CI-}g_{ir})$. We compute the empirical coverage for each individual as the mean across the 100 simulation replicates $\text{coverage}_i = \frac{1}{100}\sum_{r=1}^{100} I(g_{ir} \in \text{CI-}g_{ir})$.

## Ethics declarations

All research carried out in this study conformed with the principles of the Helsinki Declaration. All individuals provided written informed consent to the original recruitment of the UCLA ATLAS Community Health Initiative. Patient Recruitment and Sample Collection for Precision Health Activities at UCLA is an approved study by the UCLA Institutional Review Board (Institutional Review Board number 17-001013). All analyses in this study use de-identified data (without any protected health information) with no possibility of re-identifying any of the participants.

## Reporting summary

Further information on research design is available in the Nature Portfolio Reporting Summary linked to this article.

## Data availability

The individual-level genotype and phenotype data of UKBB are available by application from http://www.ukbiobank.ac.uk/. Owing to privacy concerns, de-identified individual-level data for UCLA ATLAS are available only to UCLA researchers and can be accessed through the Discovery Data Repository Dashboard (https://it.uclahealth.org/about/ohia/ohia-products/discovery-data-repository-dashboard-0). Summary ATLAS association statistics are publicly available at https://atlas-phewas.mednet.ucla.edu/. PGSs for 84 traits investigated in this manuscript are available at the PGS Catalog (PGP000457, https://www.pgscatalog.org/publication/PGP000457). MCMC samplings of PGSs are available at https://doi.org/10.6084/m9.figshare.22413970.

## Code availability

Scripts for simulations and real-data analyses are available at https://github.com/yidingdd/individual-pgs-accuracy (ref. 72).

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

**Acknowledgements** We thank D. Geschwind, Y. Wu, R. Mester, R. Johnson and K. S. Burch for insightful comments. We gratefully acknowledge the resources provided by the Institute for Precision Health and participating patients from the UCLA ATLAS Community Health Initiative. The UCLA ATLAS Community Health Initiative in collaboration with UCLA ATLAS Precision Health Biobank is a programme of the Institute for Precision Health, which directs and supports the biobanking and genotyping of biospecimen samples from participating patients from UCLA in collaboration with the David Geffen School of Medicine, UCLA Clinical and Translational Science Institute and UCLA Health. The ATLAS Community Health Initiative is supported by UCLA Health, the David Geffen School of Medicine and a grant from the UCLA Clinical and Translational Science Institute (UL1TR001881). This research was conducted using the UKBB resource under application 33127. We thank the participants of UKBB for making this work possible. This work was financially supported in part by National Institutes of Health awards U01HG011715, R01HG009120 and R01MH115676. The content is solely the responsibility of the authors and does not necessarily represent the official views of the National Institutes of Health.

**Author contributions** Y.D. and B.P. conceived and designed the experiments. Y.D., K.H., Z.X., A.P., E.P. and K.B. carried out the experiments and statistical analyses. F.P., B.J.V. and L.M.O.L. provided statistical support. Y.D., K.H. and B.P. wrote the manuscript with the participation of all authors.

**Competing interests** The authors declare no competing interests.

### Additional information

**Correspondence and requests for materials** should be addressed to Yi Ding or Bogdan Pasaniuc.

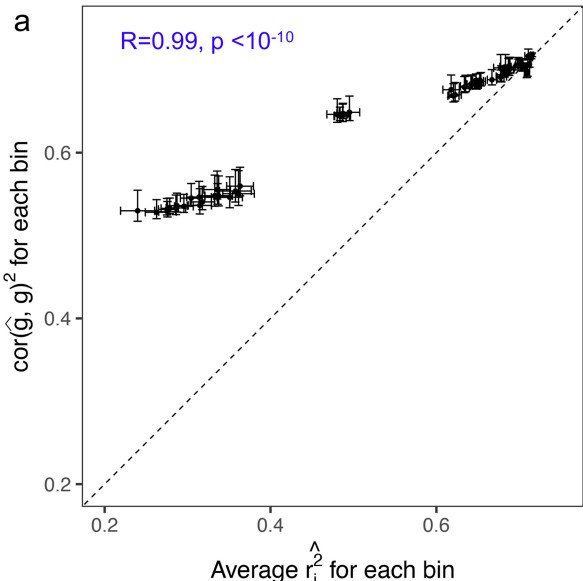

a  R=0.99, p <10⁻¹⁰

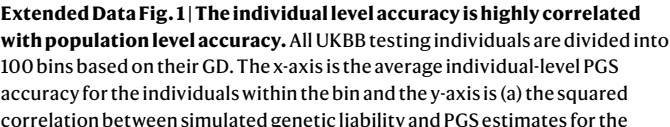

b  R=0.86, p <10⁻¹⁰

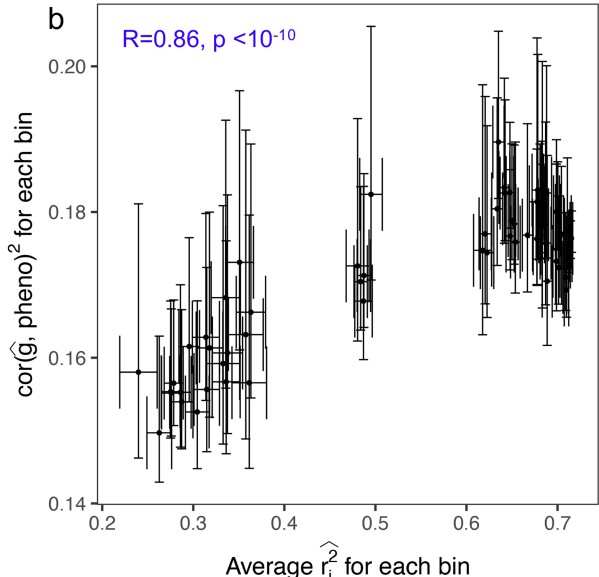

**Extended Data Fig. 1 | The individual level accuracy is highly correlated with population level accuracy.** All UKBB testing individuals are divided into 100 bins based on their GD. The x-axis is the average individual-level PGS accuracy for the individuals within the bin and the y-axis is (a) the squared correlation between simulated genetic liability and PGS estimates for the individuals within the bin (b) the squared correlation between simulated phenotype and PGS estimates. The dot and error bars represent the mean and ±1.96 s.e.m from 100 simulations. Both p-values were derived from two-sided Pearson correlation tests without adjustment for multiple hypothesis testing. Any p-value below $10^{-10}$ is annotated as $p < 10^{-10}$.

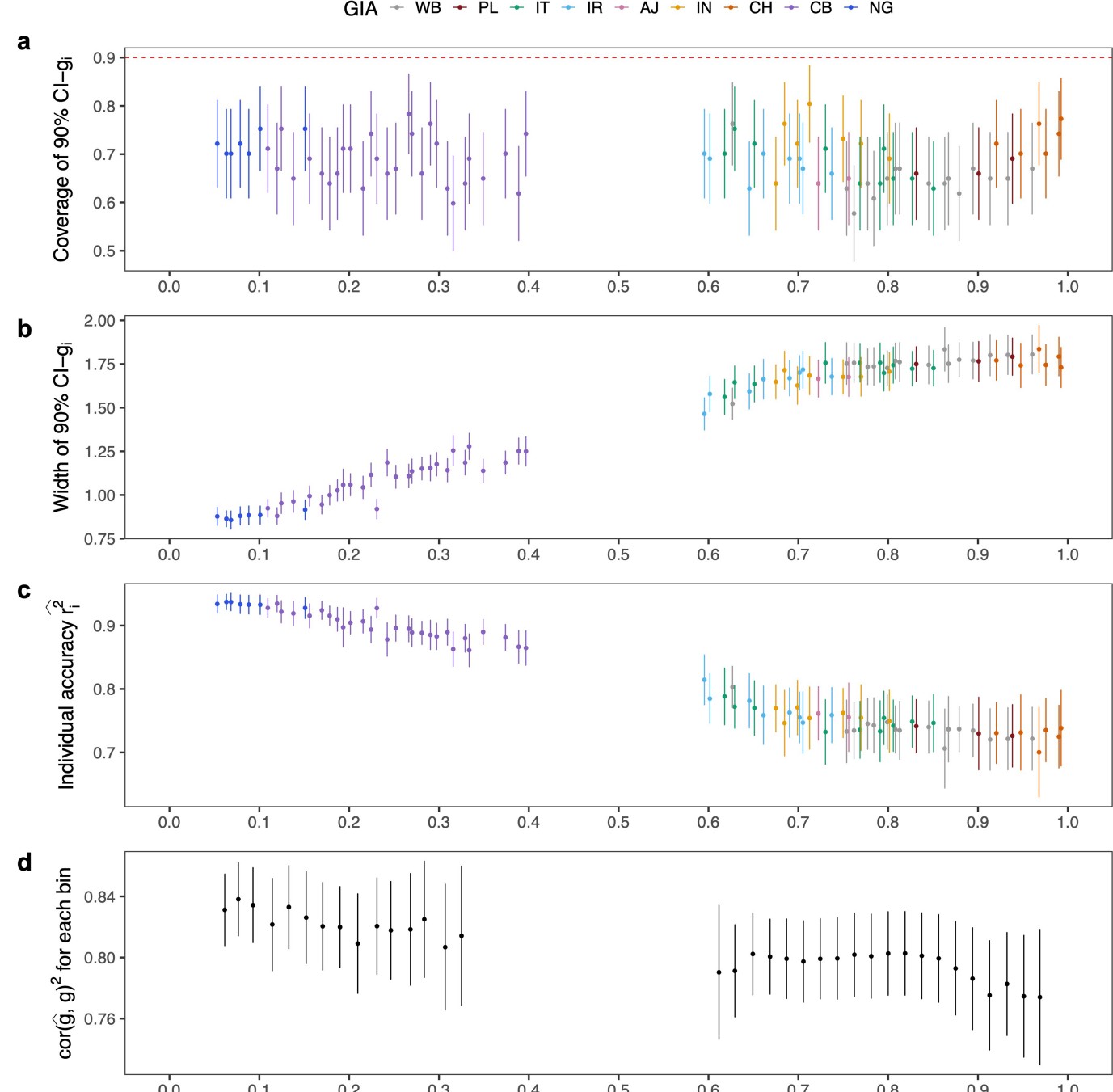

**Extended Data Fig. 2 | PGS performance varies across GD in simulations using CB and NG as training data** ($h_g^2 = 0.8$ **and** $p_{causal} = 0.1\%$**).** (a) The coverage of the 90% credible intervals of genetic liability (CI-$g_i$) is approximately uniform across testing individuals at all GDs. The red dotted line represents the expected coverage of 90% CI-$g_i$. Each dot represents a randomly selected UKBB testing individual. For each dot, the x-axis is its GD from African training data, the y-axis is the empirical coverage of 90% CI-$g_i$ calculated as the proportion of simulation replicates where the 90% credible intervals contain the individual's true genetic liability, and the error bars represent mean ±1.96 standard error of the mean (s.e.m) of the empirical coverage calculated from 100 simulations. (b) The width of 90% CI-$g_i$ increases with GD. For each dot, the y-axis is the average width of 90% CI-$g_i$ across 100 simulation replicates, and the error bars represent ±1.96 s.e.m. (c) Individual PGS accuracy decreases with GD. For each dot, the y-axis is the average individual level PGS accuracy across 100 simulation replicates, and the error bars represent ±1.96 s.e.m. (d) Population-level metrics of PGS accuracy recapitulates the decay in PGS accuracy across genetic continuum. All UKBB testing individuals are divided into 100 equal-interval bins based on their GD. The x-axis is the average GD for the bin and the y-axis is the squared correlation between genetic liability and PGS estimates for the individuals within the bin. The dot and error bars represent the mean and ±1.96 s.e.m from 100 simulations.

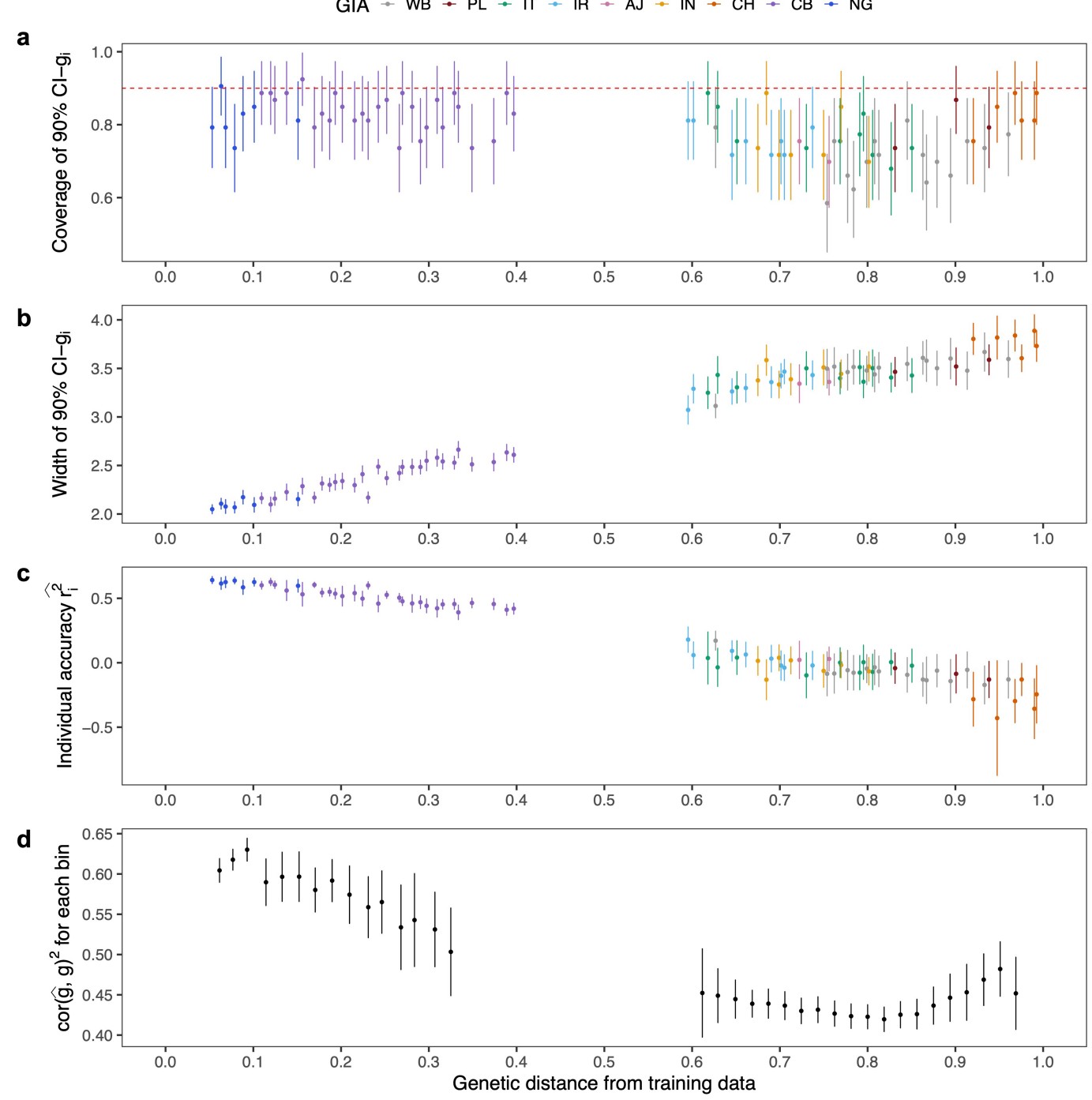

**Extended Data Fig. 3 | PGS performance varies across GD in simulations using CB and NG as training data** ($h^2_g = 0.8$ and $p_{causal} = 1\%$)**.** (a) The coverage of the 90% credible intervals of genetic liability (CI-$g_i$) is approximately uniform across testing individuals at all GDs. (b) The width of 90% CI-$g_i$ increases with GD. (c) Individual PGS accuracy decreases with GD. (d) Population-level metrics of PGS accuracy recapitulates the decay in PGS accuracy across genetic continuum. See Extended Data Fig. 2 for a detailed figure description.

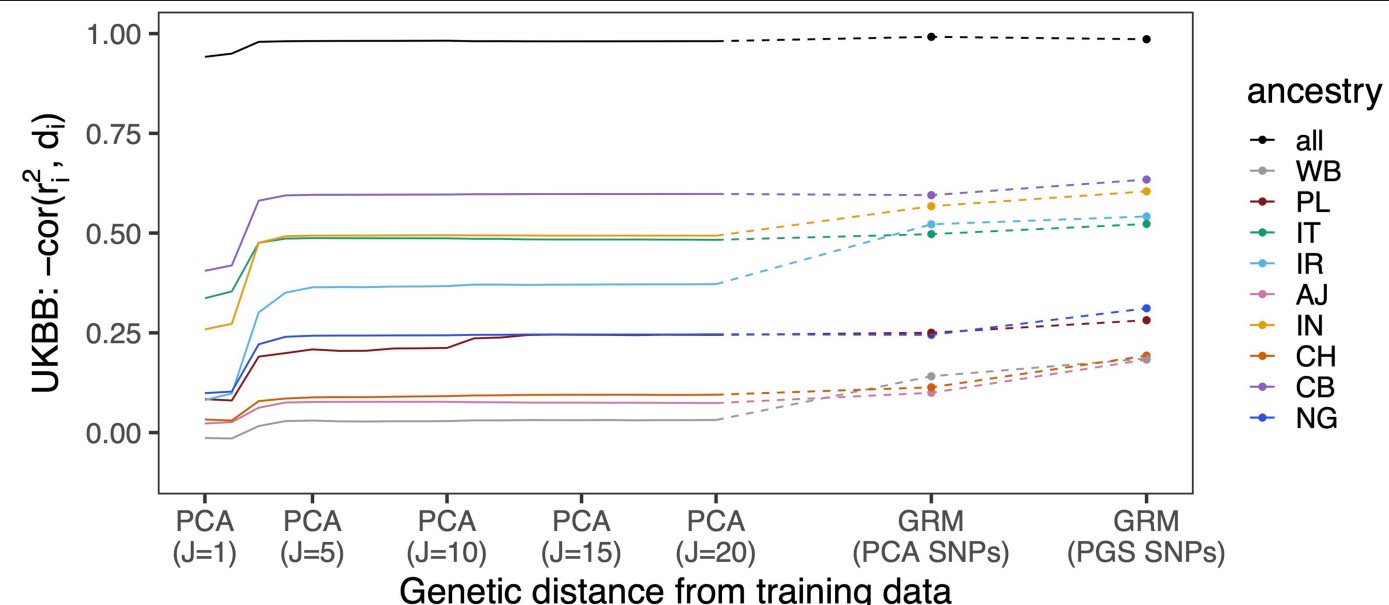

**Extended Data Fig. 4 | The effect of different metrics of GD on the correlation between GD and accuracy.** The y-axis $-cor(r_i^2, d_i)$ is the correlation between the GD and PGS accuracy; a larger correlation means GD has a better prediction of accuracy. The x-axis are different GD metrics: (1) GD based on PCA with varying number of PCs (from J = 1 to J = 20) and (2) GD based on GRM using pruned PCA SNPs only or all SNPs in PGS models. The GRM GD is computed as $d_i(\text{GRM}) = \sqrt{\frac{1}{K}\sum_{k=1}^{K}(x_i - x_k)^2}$, where $x_i$ is the standardized genotype of $i_{th}$ testing individual and $x_k$ is the standardized genotype of $k_{th}$ training individual.

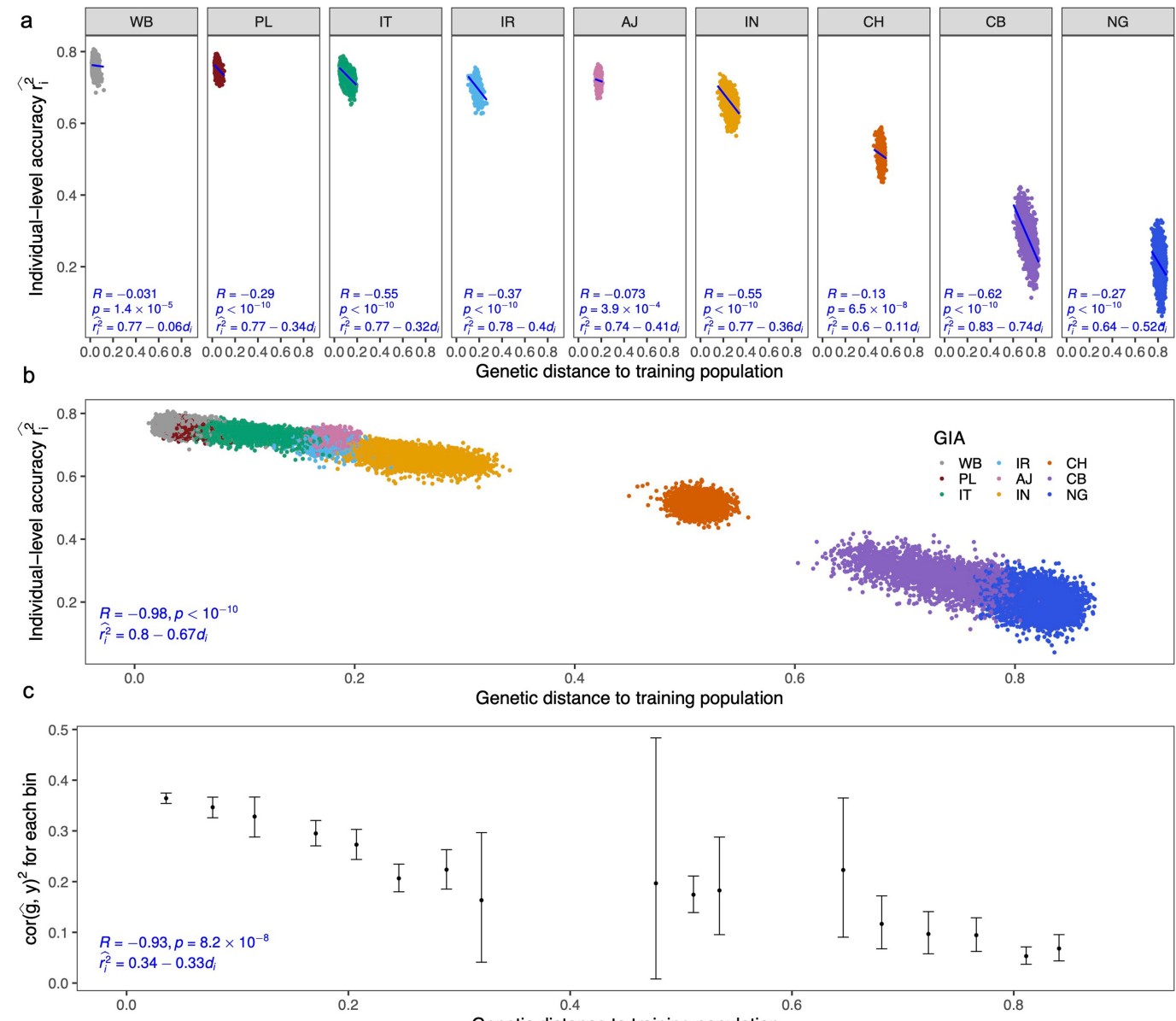

**Extended Data Fig. 5 | The individual-level accuracy for height PGS decreases across the genetic ancestry continuum in UKBB.** (a) Individual PGS accuracy decreases within subcontinental GIA clusters. Each dot represents a testing individual from UKBB. For each dot, the x-axis represents its distance from the training population on the genetic continuum; the y-axis represents its PGS accuracy. The color represents the GIA cluster. (b) Individual PGS accuracy decreases across the entire UKBB. (c) The population PGS accuracy decreases with the average GD in each bin. All UKBB individuals are divided into 20 equal-interval GD bins. The x-axis is the average GD within the bin; the y-axis is the squared correlation between PGS and phenotype for individuals in the bin. The dot and error bar show mean and 95% confidence interval from 1000 bootstrap samples. $R$ and $p$ refer to the correlation between GD and PGS accuracy and its significance from two-sided Pearson correlation tests without adjustment for multiple hypothesis testing. Any p-value below $10^{-10}$ is shown as $p < 10^{-10}$.

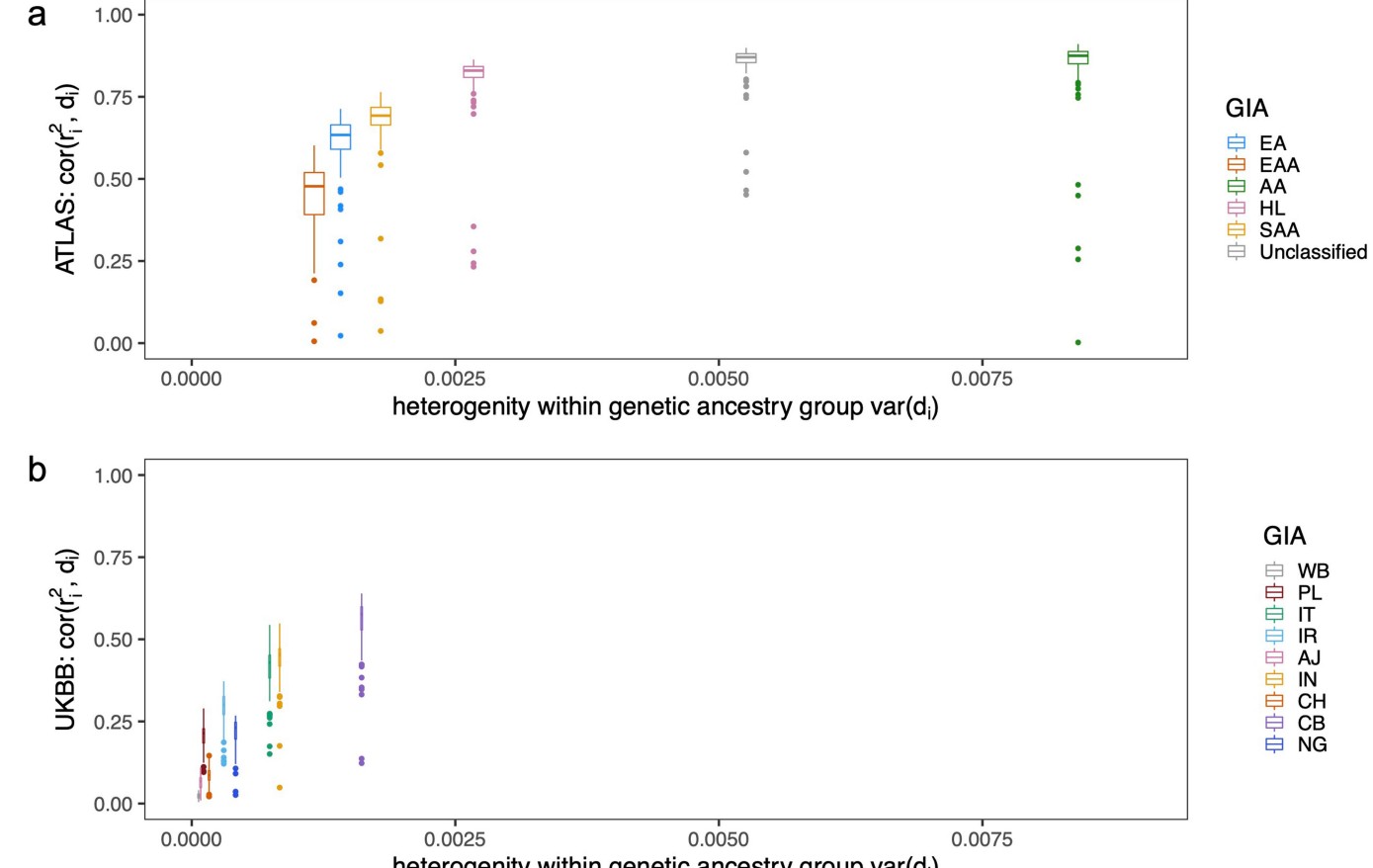

**Extended Data Fig. 6 | Lower heterogeneity within the genetic ancestry group corresponds to a lower correlation between genetic distance and individual PGS accuracy.** (a) The distribution of correlations between PGS accuracy and GD for 84 traits in ATLAS. (b) The distribution of correlations between PGS accuracy and GD for 84 traits in UKBB. The x-axis is the homogeneity of the genetic ancestry clusters measured as the variance of GD within a genetic ancestry cluster; a larger var($d_i$) indicates a larger variation of genetic background. Each boxplot contains 84 points corresponding to the correlation between PGS accuracy and GD within the group specified by x-axis for each of the 84 traits. The box shows the first, second and third quartile of the 84 correlations, and whiskers extend to the minimum and maximum estimates located within 1.5×IQR from the first and third quartiles, respectively.

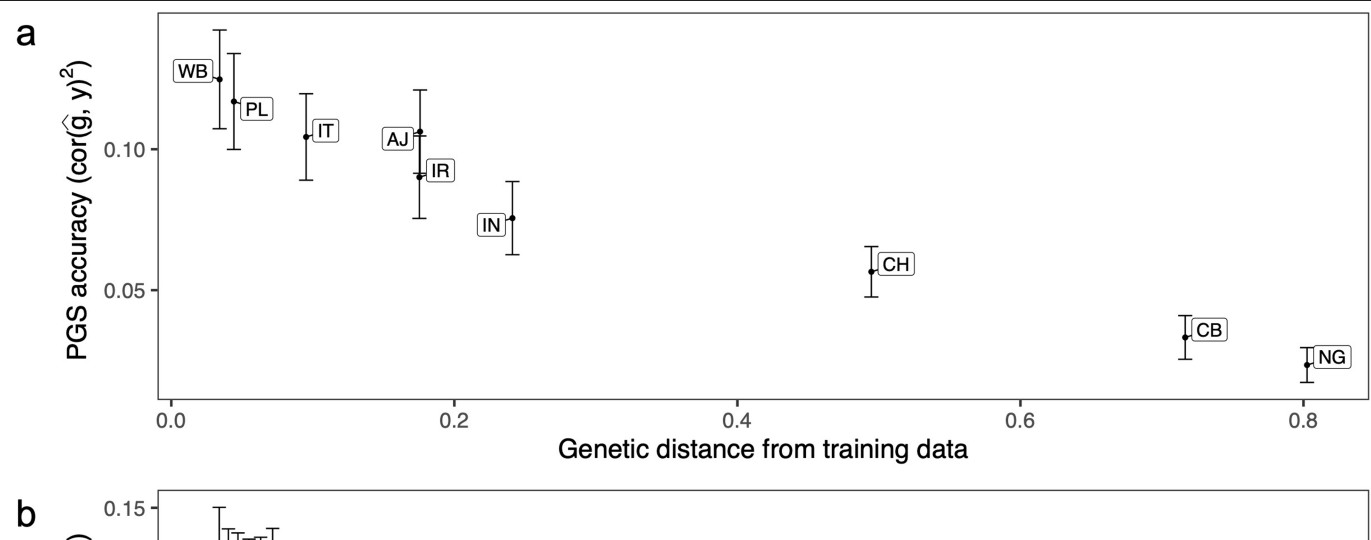

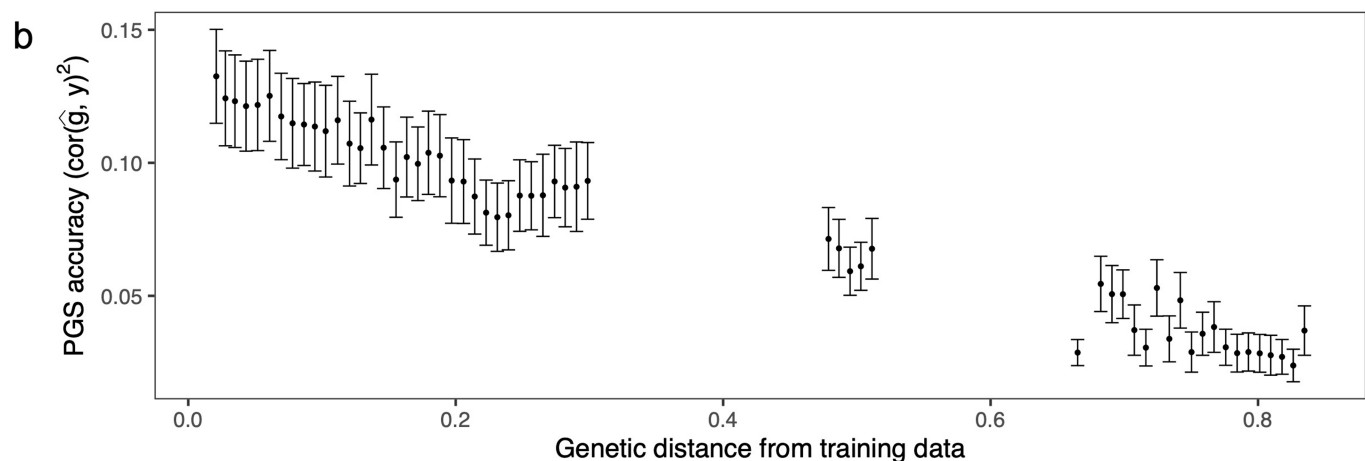

**Extended Data Fig. 7 | Empirical PGS accuracy decreases with genetic distance in UKBB averaged across 84 traits.** (a) Empirical PGS accuracy decreases across subcontinental GIA clusters. (b) Empirical PGS accuracy decreases across bins of GD. The x-axis is the average GD for all individuals within each GIA cluster or GD bin; the y-axis is the accuracy for each GIA cluster or GD bin. The dot and error bar show mean and ±1.96 standard error of the mean across 84 traits.

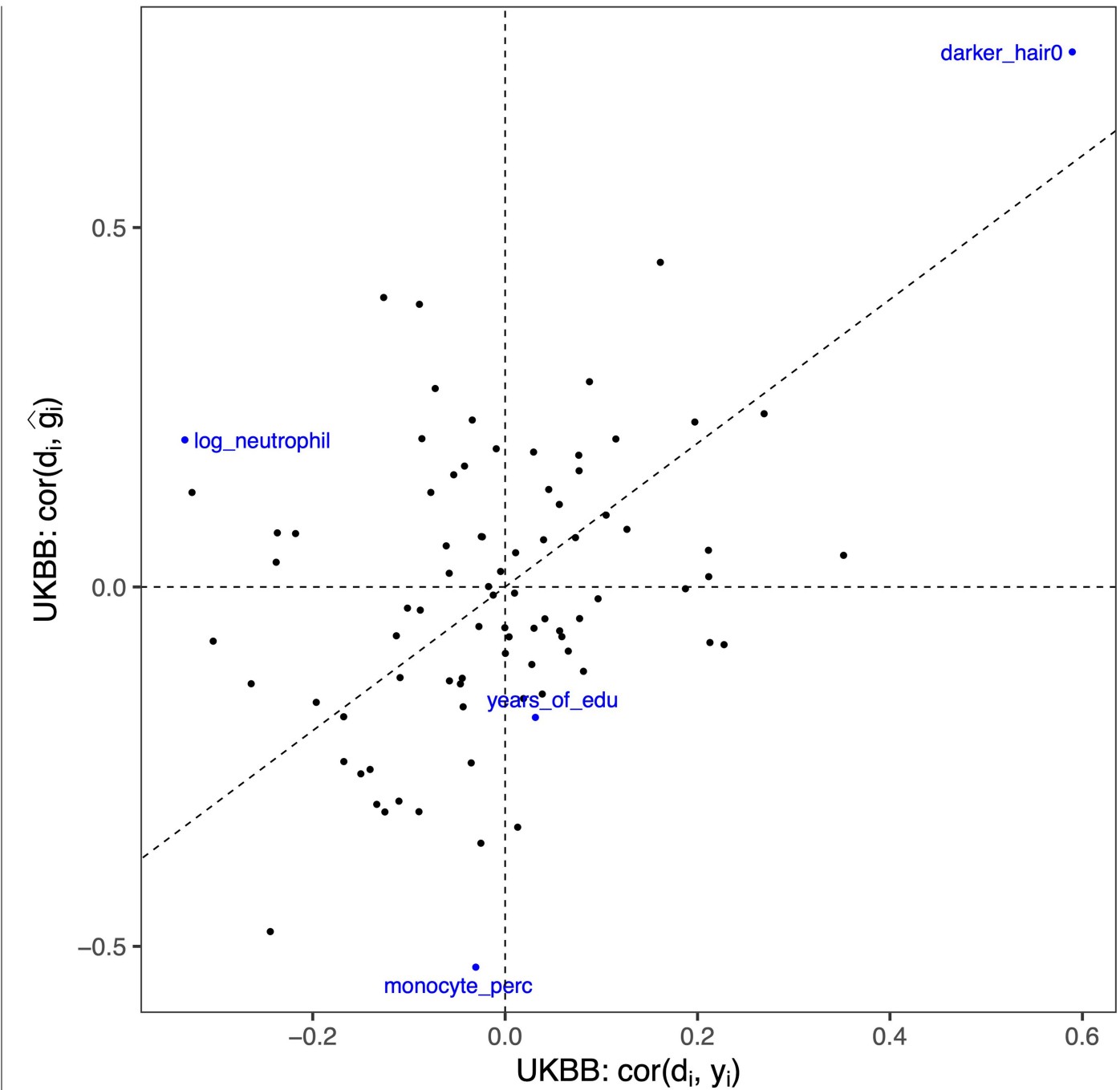

**Extended Data Fig. 8 | Discordant directions of phenotype/PGS-distance correlations in UKBB.** The x axis is the correlation between phenotype and GD and the y axis is the correlation between PGS estimates and GD for all 48,586 testing individuals in UKBB. Numerical results are reported in Supplementary Table 4.

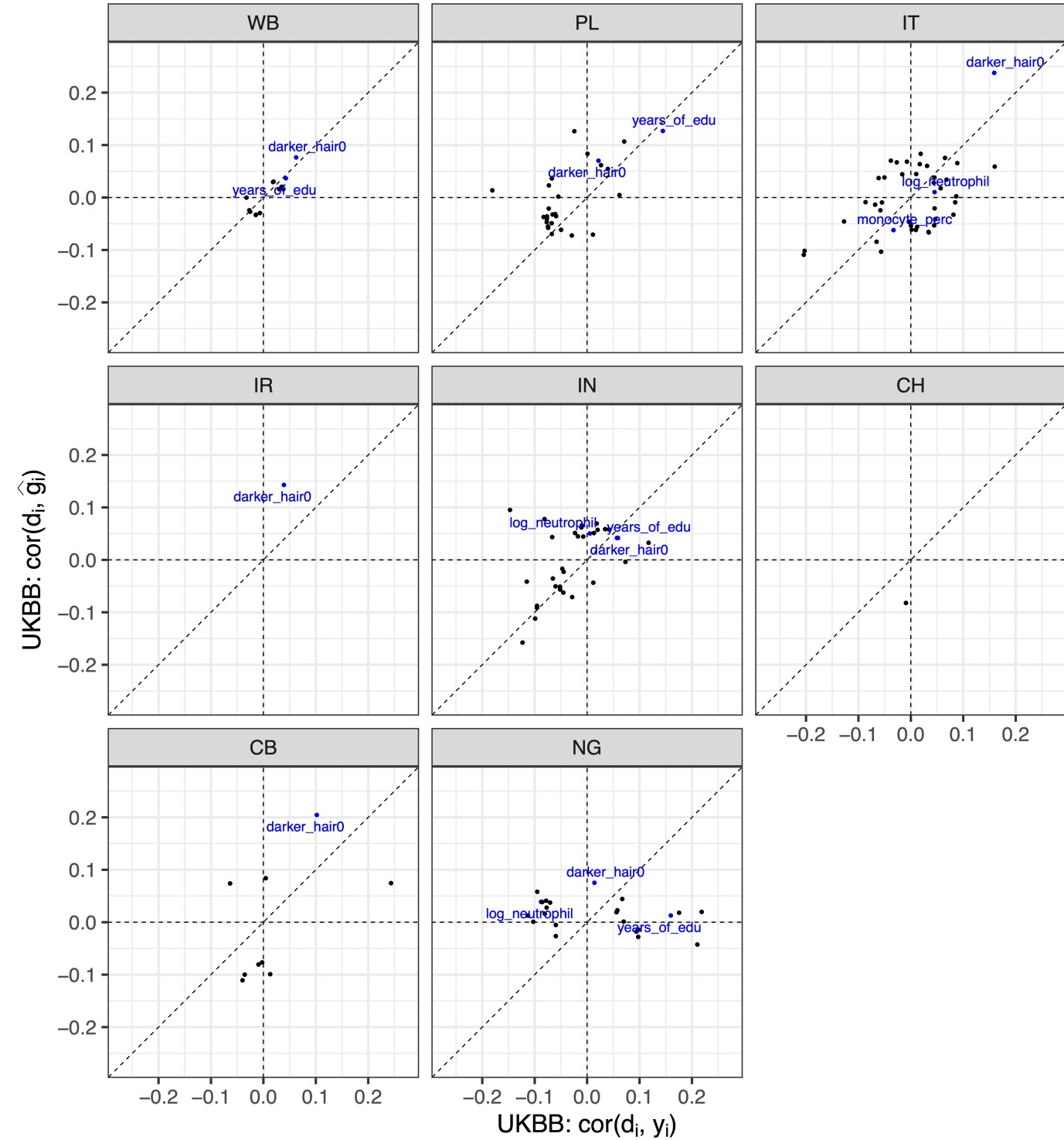

**Extended Data Fig. 9 | The correlation of PGS/phenotype with GD within each ancestry clusters in UKBB.** Only traits that exhibit significant correlation between GD and PGS/phenotype are shown as dots in the figure. The Ashkenazi cluster is not included because no significant correlations are observed.

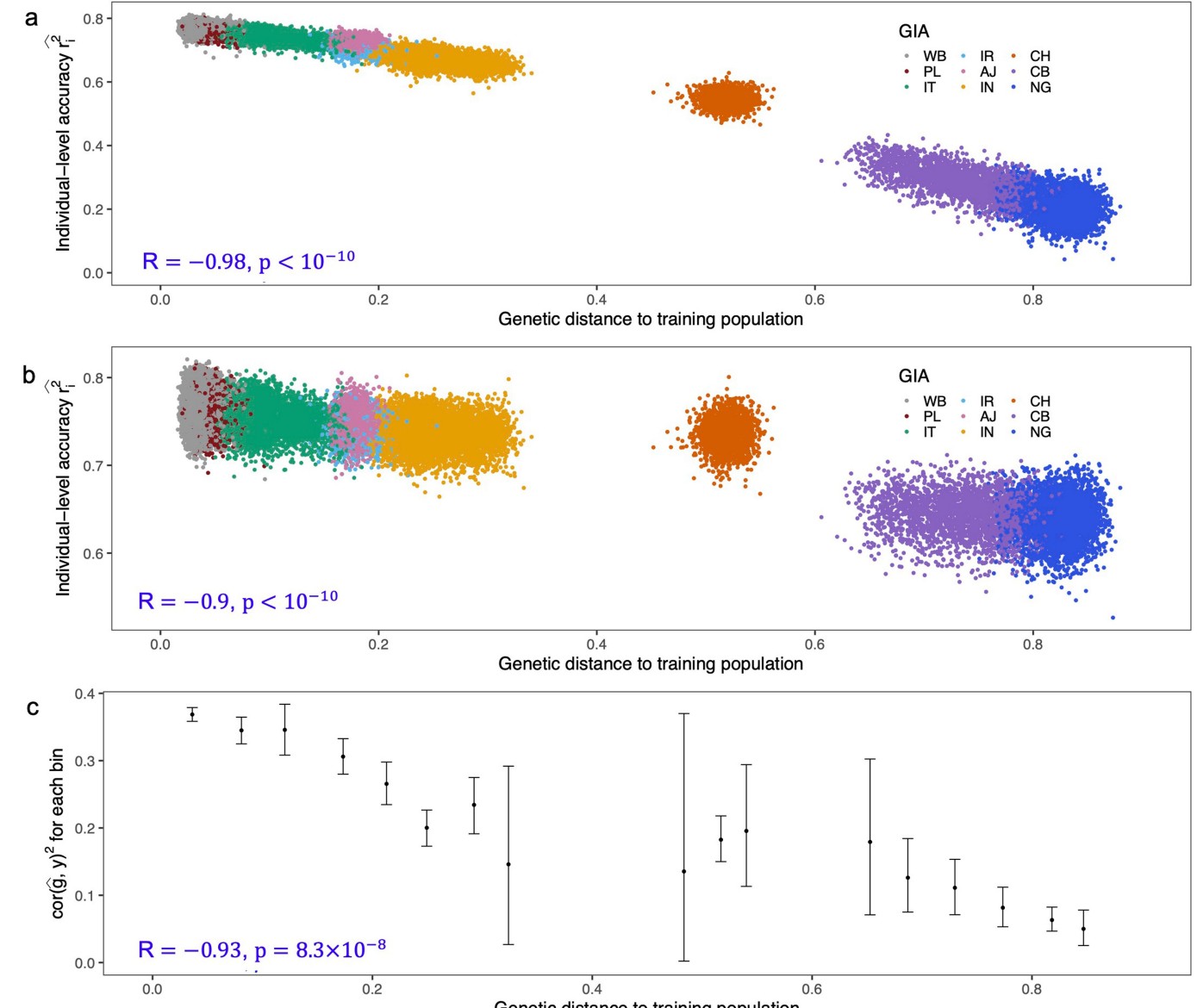

**Extended Data Fig. 10 | Comparison of individual accuracy for height in UKBB.** (a) Accuracy computed from equation (1), with $var_\beta(x_i^\top\beta)$ set as fixed heritability; (b) Accuracy computed from equation (1), with $var_\beta(x_i^\top\beta)$ estimated from Monte Carlo sampling from prior distribution of $\beta$. (c) Empirical accuracy estimated as the squared correlation between PGS and height for each genetic distance bin. All UKBB individuals are divided into 20 equal-interval GD bins.

The x-axis is the average GD within the bin; the y-axis is the squared correlation between PGS and phenotype for individuals in the bin. The dot and error bar show mean and 95% confidence interval from 1000 bootstrap samples. Both (a) and (b) reflect the decreasing trend of empirical accuracy in (c). All p-values were derived from two-sided Pearson correlation tests without adjustment for multiple hypothesis testing. Any p-value below $10^{-10}$ is shown as $p < 10^{-10}$.

# Reporting Summary

## Statistics

For all statistical analyses, confirm that the following items are present in the figure legend, table legend, main text, or Methods section.

| n/a | Confirmed | |
|---|---|---|
| ☐ | ☒ | The exact sample size (*n*) for each experimental group/condition, given as a discrete number and unit of measurement |
| ☒ | ☐ | A statement on whether measurements were taken from distinct samples or whether the same sample was measured repeatedly |
| ☐ | ☒ | The statistical test(s) used AND whether they are one- or two-sided *Only common tests should be described solely by name; describe more complex techniques in the Methods section.* |
| ☐ | ☒ | A description of all covariates tested |
| ☐ | ☒ | A description of any assumptions or corrections, such as tests of normality and adjustment for multiple comparisons |
| ☐ | ☒ | A full description of the statistical parameters including central tendency (e.g. means) or other basic estimates (e.g. regression coefficient) AND variation (e.g. standard deviation) or associated estimates of uncertainty (e.g. confidence intervals) |
| ☐ | ☒ | For null hypothesis testing, the test statistic (e.g. *F*, *t*, *r*) with confidence intervals, effect sizes, degrees of freedom and *P* value noted *Give P values as exact values whenever suitable.* |
| ☐ | ☒ | For Bayesian analysis, information on the choice of priors and Markov chain Monte Carlo settings |
| ☒ | ☐ | For hierarchical and complex designs, identification of the appropriate level for tests and full reporting of outcomes |
| ☐ | ☒ | Estimates of effect sizes (e.g. Cohen's *d*, Pearson's *r*), indicating how they were calculated |

*Our web collection on statistics for biologists contains articles on many of the points above.*

## Software and code

Policy information about availability of computer code

| Data collection | No software is used for data collection. |
|---|---|
| Data analysis | LDpred2 software implementing individual PRS uncertainty: https://privefl.github.io/bigsnpr/articles/LDpred2.html , https://github.com/yidingdd/individual-pgs-accuracy. Softwares used in this manuscript: bigsnpr(v1.8.1 for LDpred2),  flashpca(v2.0), plink(v2.0a3LM), ggplot2(v3.3.5). |

For manuscripts utilizing custom algorithms or software that are central to the research but not yet described in published literature, software must be made available to editors and reviewers. We strongly encourage code deposition in a community repository (e.g. GitHub). See the Nature Portfolio guidelines for submitting code & software for further information.

## Data

Policy information about availability of data

All manuscripts must include a data availability statement. This statement should provide the following information, where applicable:

- Accession codes, unique identifiers, or web links for publicly available datasets
- A description of any restrictions on data availability
- For clinical datasets or third party data, please ensure that the statement adheres to our policy

The individual-level genotype and phenotype data of UKBB are available by application from http://www.ukbiobank.ac.uk/. Due to privacy concerns, de-identified individual level data for UCLA ATLAS are available to UCLA researchers and can be accessed through the Discovery Data Repository Dashboard (https://

## Human research participants

Policy information about studies involving human research participants and Sex and Gender in Research.

**Reporting on sex and gender**
Biological sex are used in our study as covariates in training PGS models to avoid potential confounding effects. The finding in our study is applied to both sex.

**Population characteristics**
The average age of UCLA-ATLAS participants, defined as a participant's age recorded in the EHR as of September 2021, is 55.6 (SD: 17.2) years. Based on biological sex inferred from genotype data, 55% participants are Female and 45% participants are male. We perform PCA to cluster individuals into different genetic ancestry culsters; 61% are inferred to be European Americans, 17% Hispanic/Latino Americans, 2% South Asian Americans, 9% East Asian Americans, 5% African Americans and 6% are unclassified due to lack of reference panel.

**Recruitment**
Participants are recruited from 18 UCLA Health medical centers, laboratories, and clinics located throughout the greater Los Angeles area. Participants watch a short video outlining the goals of the initiative and document their choice of whether they wish to consent to participation. Our result is not impacted by selection bias if there's any. Full details on recruitment procedure are provided at https://www.uclahealth.org/precision-health/programs/ucla-atlas-community-health-initiative Patient Recruitment and Sample Collection for Precision Health Activities at UCLA is an approved study by the UCLA Institutional Review Board (UCLA IRB). IRB#17-001013

**Ethics oversight**
UCLA Institutional Review Board

Note that full information on the approval of the study protocol must also be provided in the manuscript.

# Field-specific reporting

Please select the one below that is the best fit for your research. If you are not sure, read the appropriate sections before making your selection.

☒ Life sciences ☐ Behavioural & social sciences ☐ Ecological, evolutionary & environmental sciences

For a reference copy of the document with all sections, see nature.com/documents/nr-reporting-summary-flat.pdf

# Life sciences study design

All studies must disclose on these points even when the disclosure is negative.

**Sample size**
We used two dataset in our study: the UK Biobank contains 487409 participants and UCLA-ATLAS contains 36778 participants. We did not collect new dataset for this manuscript. The UK Biobank dataset has a reasonable sample size for training PGS models as it has been widely used for the same purpose by previous publications. The UCLA-ATLAS dataset was solely used for evaluating the PGS performance in our study and its sample size is comparable to previous research studies for the same purpose.

**Data exclusions**
In UCLA-ATLAS cohort, individuals with >5% missingness in genotype are excluded due to low data quality.

**Replication**
We successfully replicated our finding that the individual PGS accuracy decays with increased genetic distance from the training data in two datasets: UK Biobank and UCLA-ATLAS.

**Randomization**
Randomization is not relevant for this study as we didn't assign individuals into different experimental groups.

**Blinding**
The investigators are blinded to group allocation.

# Reporting for specific materials, systems and methods

We require information from authors about some types of materials, experimental systems and methods used in many studies. Here, indicate whether each material, system or method listed is relevant to your study. If you are not sure if a list item applies to your research, read the appropriate section before selecting a response.

## Materials & experimental systems

| n/a | Involved in the study |
|-----|----------------------|
| ☒ | ☐ Antibodies |
| ☒ | ☐ Eukaryotic cell lines |
| ☒ | ☐ Palaeontology and archaeology |
| ☒ | ☐ Animals and other organisms |
| ☒ | ☐ Clinical data |
| ☒ | ☐ Dual use research of concern |

## Methods

| n/a | Involved in the study |
|-----|----------------------|
| ☒ | ☐ ChIP-seq |
| ☒ | ☐ Flow cytometry |
| ☒ | ☐ MRI-based neuroimaging |

