## [Peer Review File · Nature]

Manuscript Title: Polygenic scoring accuracy varies across the genetic ancestry continuum

Reviewer Comments & Author Rebuttals

Reviewer Reports on the Initial Version:

Referees' comments:

Referee #1 (Remarks to the Author):

The authors use the theoretical concept prediction error variance (PEV) of a predicted genetic value for an individual to show that PEV increases the less related an individual is to the individuals that have a phenotype. The authors do not phrase it that way, but in essence this is what their paper is about. The authors show empirically, using two large biobank datasets, that individual prediction accuracy declines with relatedness (genetic distance from the samples with phenotypes) both within and across traditional (but arbitrary) ancestry boundaries, and that the decline is approximately linear. The authors quantify the association between genetic distance and prediction accuracy across a number of traits. They thereby show that prediction error variance greatly increases for genomes that are not well represented in the discovery data, in line with published results (as the authors acknowledge). They do not provide a "solution", other than state (as many others have done) that more (biobank) data should be collected across the entire spectrum of human genome variation to increase prediction accuracy for individuals irrespective of their genomic ancestry.

Comments on the Introduction:

There are non-genetic prediction tools used in clinical practice that suffer from the same issues that are raised by the authors, yet it seems fashionable to pick out polygenic scores (PGS) as providing an equity barrier. The authors may wish to point out that PGS are not an exception in terms of equitable use of clinical predictors.

It is correct that in human genetics, population or sample based average measures of prediction accuracy have been used, for understandable reasons – it is hard enough to explain a point estimate of a PGS to a person, let alone the error variance around it. However, in agriculture, scientists have for decades not only used well-developed theory on prediction error variance, they also routinely provide individual-specific 'reliability' metrics for practical applications. My suggestion is that the authors acknowledge this more explicitly and provide more and appropriate references.

The authors hint at 'relatedness' (ln260; ln296) and 'animal breeding theory' (ln339) but don't acknowledge that there is a large body of literature out there on prediction error using either pedigree relationships or SNP-based relationships. 'Animal breeding theory' is a misnomer because the theory is (hard core) genetical and statistical, it just happens that it was first derived (& applied) in agriculture. All the theory presented in the present paper is a special case of more general results in the literature. Many relevant papers are not in highly specialized journals but in textbooks and mainstream genetics journals. Examples of relevant literature include:

Falconer (1960). An Introduction to Quantitative Genetics (text book). Has relevant derivations how the accuracy of prediction depends on the relationship with individuals with a phenotype. The simplest case is to predict a genetic value from a single (pedigree) relative with a phenotype. Then $r^2(\text{true and predicted gv}) = \theta^2 * h^2$, with θ = relationship and h^2 = heritability. An

extreme example of that was shown empirically by Truong et al. 2020 (Nature Communications), also using UK Biobank data. Those authors show that prediction accuracy from close relatives was equivalent to that from a much larger sample of distant relatives.

Lynch and Walsh (1998). Genetics and Analysis of Quantitative Traits (Chapter 26). The theory about PEV when using a pedigree relationship matrix is the same as when using a GRM (genomic relationship matrix) from SNPs, and the expression simplifies when there are no fixed effects. The relevant matrix that the authors approximate in the current paper is $C22$ (Eq 26.6), which simplifies to the inverse of $[I + G(\text{inverse})]$, with G the SNP-based GRM. G is dense hence approximations are necessary for large datasets.

Wientjes YC, Veerkamp RF, Bijma P, Bovenhuis H, Schrooten C, Calus MP. Empirical and deterministic accuracies of across-population genomic prediction. Genetics Selection Evolution. 2015 Feb 6;47(1):5. doi: 10.1186/s12711-014-0086-0. This paper contains derivations of the accuracy for across-population prediction accuracy, using genomic relationships. Eq[10] in this paper seems most relevant.

Habier D, Fernando RL, Dekkers JC. The impact of genetic relationship information on genome-assisted breeding values. Genetics. 2007 Dec;177(4):2389-97. doi: 10.1534/genetics.107.081190. These authors show (using simulations) that "the impact of relationships was greatest for RR-BLUP", with RR-BLUP their name for the PGS used by Ding and colleagues.

Lee et al. 2017 (Scientific Report); Truong et al. 2020 (Nature Communications). These authors report higher prediction accuracy for individuals who are more closely related to the individuals with a phenotype in the UK Biobank. Although the contrast in this paper is between close (pedigree) relatives and very distantly related individuals, the concept and results are as reported by Ding and colleagues.

Ben Zaabza, H., E. A. Mäntysaari, and I. Strandén. 2020a. Using Monte Carlo method to include polygenic effects in calculation of SNP-BLUP model reliability. Journal of Dairy Science 103: 5170-5182. This paper provides derivations and approximation to the PEV when using a SNP-BLUP model (as used by Ding and colleagues).

Technical comments:

It was not clear to me why the authors use 'gv' and not simply 'g' (as frequently done in the literature) to parameterise the genetic value.

The authors present their theory and results as if gv is the full genetic value and its variance the full heritability. However, it is only that which is captured by common SNPs and therefore some of the unmodelled error variance is actually genetic. A case in point is the example of neutrophil count. The distinction is important because for some trait the r^2 value reaches ~ 0.8 , which gives the impression that the total genetic value is predicted nearly perfectly, even if the majority of genetic variation might not be captured by SNPs.

Under some assumptions (SNPs effects are random and from a single normal distribution), a SNP-based model is equivalent to a relationship model (VanRaden 2008, J Dairy Science; Goddard 2008, Genetica; Yang et al. 2010, Nature Genetics). This model is also what the authors assume. Therefore, if the authors were to perform a GBLUP analysis then the inverse of the equations (as in Lynch & Walsh chapter 26) gives the PEV and hence the prediction accuracy. The authors approximate the reliability by reducing the dimensionality of the GRM (the eigenvalues of XX' are the same as those of $X'X$), using the top-20 eigenvectors only, and then further by ignoring a shrinkage term (treating the top-20 eigenvectors as fixed effects). Whilst this seems to work fine for individuals who are distantly related (which is picked up easily by the top-20 eigenvectors), it is likely to work less well for another sample from the same population (/ ancestry). This is

consistent with Fig4b. Therefore, my suggestion is that the authors explore using the entire GRM to quantify reliability ($J = N$ or M , whichever is smaller) and use shrinkage. My guess is that this will show better concordance between observed and predicted errors. If that is not feasible computationally then the authors could increase J and quantify if/how the results in Fig4b improve.

The strong inverse correlation between genetic distance and accuracy seems to imply that the reduction in accuracy in samples from ancestries that are different from the GWAS samples is fully accounted for by the distance metric. However, previous studies have shown that MAF and LD do not fully account for the loss in accuracy across ancestries. Can the authors elaborate/discuss how their results compare to those studies? In addition, the authors state repeatedly that more data should be collected/considered from other ancestries. However, there is emerging evidence (including from some of the authors) that causal effects are shared and similar in size across ancestries. Therefore, in principle at least, discovery of causal variants could be performed in a single ancestry and a PGS based on the causal variants should work across all ancestries (apart from those that are fixed). Can the authors comment on this thought experiment?

In309-313: I found this section very confusing. The only randomness that matters for this paper is in the genotypes (the x-variables)!

In315-324: It might be helpful to point out to the reader what this result means, i.e., the regression of true on predicted genetic values has a slope of 1.

Supp Fig3b: why is this relationship non-linear? Would it be more linear if the x-axis is in variance units?

Supp Fig4: the metric d_i measures genetic distance from a UK Biobank sample. Individuals with different ancestries (e.g., East Asian and African) could have the same d_i values (Fig3b). Yet the genetic (and, perhaps, phenotypic) mean values of East Asian and African populations could be quite different (given that they both have an F_{st} of $\sim 0.12-0.15$ with Europeans). Therefore, I'm not sure what Supp Fig4 shows without a (discrete) ancestry component.

Referee #2 (Remarks to the Author):

This paper presents a short, clear, and highly important analysis of the portability of polygenic scores as a function of genetic ancestry. The paper is innovative in its methods (showing methods for doing individual-level PGS accuracy prediction) and for its framing (conceptualizing the analysis in terms of continuous genetic distance versus discrete ancestries). The lessons it gives are important for the human genetics community (and allied communities doing trait prediction in other taxa).

The paper is overall very well executed – my main comments are with regards to aspects of the presentation and some comments regarding the approximations used.

Major/Technical:

1a. A stated caveat of the method is that it assumes effect sizes are constant across ancestry. It would be helpful to be more clear throughout that for that reason, the r_i^2 estimate is not capturing all sources of error and is likely an upper bound on the accuracy.

1b. Given the current formulation of r_i^2 varies across individuals because of how individuals project on the training set eigenvectors (line 366 and discussion on lines 373-375). The key aspect is the additive genetic distance from the center of the training data's PC space. It would be helpful in the main text to perhaps denote this more clearly. I would propose calling these "Panel-distance

r_i^2 values, such that later someone else might be able to compute "Effect-size-variance r_i^2 " values, and eventually we might hope for a "Multi-factor r_i^2 " value that accounts for both sources of error? This suggestion may not be the ones to use, but I urge some consideration of how to both communicate the scope of the current method as it will be helpful for the future in multiple ways (preventing over-interpretation and making clear where new work will benefit).

1c. Another stated caveat is that the variance of the PGS in the denominator is assumed to be equal to the heritability and constant across ancestry. It would seem straightforward to use an approximation whereby the heritability varies in proportion to an individual-level heterozygosity. While not necessary for publication – this could be a useful improvement in performance. It also raises interesting questions – if an individual is further from the panel but also highly variable – would that moderate the decay of r_i^2 ?

2. Line 372-373: A theoretical result is presented here that is underplayed in the manuscript. The theory predicts that the performance will decay with genetic distance. And the main paper's empirical analysis supports the theory. I think the authors should consider promoting the theoretical result to the main paper – as it currently reads, the r_i^2 and GD result is an empirical observation, but a careful read shows it is both a theoretical expectation and an empirical observation.

3. Supp Fig 1a : the results show a high correlation but the scaling of the two axes is very different because covariance is used on the y-axis as opposed to the correlation – It would be nice to have an additional panel where the y-axis is the empirical correlation and where any bias away from the $y=x$ for r_i^2 can be observed directly.

4. Figure 3A: It would be helpful to report the slopes in absolute units (decay in r_i^2 per unit of genetic distance). As it is, because R depends on the variances it's difficult to know how much of a decay in performance one gets in terms of accuracy.

Line 372: The use of J here is confusing given its use in the genetic distance calculation (where J was fixed to 20). Here do you mean a complete eigendecomposition over all eigenvectors? (i.e. no truncation is implied and any truncation for a fixed J would be an approximation).

Minor (Grammatical / Clarity):

Line 40: Strike "genetic value" – the social connotation of the phrase makes it worth diminishing / removing it from technical use in genetics

Line 84: Insert >the< before "continuum"

Line 85: "genetic ancestry cluster" -> "genetic ancestry cluster"

Line 105: Notation gv_i confusing – I was puzzling if it was a multiplication of g and v_i and then eventually reverse engineered that it is the underlying additive genetic liability for the trait. That term is being called genetic value here, but as mentioned above I think we should invent new terms to get around the "value" phrasing. My encouragement is to use "genetic liability" or "additive genetic liability" to be more specific. While liability is traditionally reserved for discrete trait models – it's only a mild abuse of the terminology to use it for both discrete and continuous traits (many readers will probably not even notice the distinction and you could precisely introduce the notation where it's essential). If that is not comfortable, another idea would be to exploit the relationship to generalized linear models – in that literature the term "linear predictor" and Greek letter eta are used for what stands in for "genetic value" here.

Line 112: Especially good to change notation from gv_i given the use of v_i here for eigenvectors.

Line 112: Choice of J would be useful to declare in the main text.

Line 118: Insert >a< in "as function"

Line 125: Fix "the of"

Line 133: "under >a< discrete"

Line 170: "in the >the< HL cluster"

Line 212: "different correlations >for< PGS vs trait"

Line 217: Fix within /in

Line 221: "along GD" -> "with respect to GD"

Line 221: ">the< European American cluster"

Referee #3 (Remarks to the Author):

Polygenic scores (PGS) have recently become a popular approach for assessing predicted risk of clinical outcomes based on genetic variants identified through GWAS, where the weighting of variants used in PGS are usually proportional to variant effect sizes from the GWAS. PGS have been shown to not be transferrable across ancestrally diverse populations, with a strong bias when applied to non-European ancestry populations when PGS weights are calculated from discoveries from GWAS that have predominantly been conducted in European ancestry populations.

In this study, the authors investigate how the accuracy of PGS changes based on a proposed genetic distance (GD) measure between an individual and the population from which the PGS weights were constructed, where the GD is essentially a measure of genetic ancestry differences between groups and is based on continuous measures of ancestry only axes of variation that are reflected from a principal components analysis. In this study, the authors construct PGS weights for a variety of phenotypes using individuals who self-reported to be "white British" in the UK Biobank (UKB). They then assess the predication accuracy of PGS using groups of individuals with difference ancestries/ethnicities in UKB as well as from the Los Angeles biobank at UCLA (ATLAS). The authors find a strong negative association between PGS performance and their novel GD measure. They also identify differences in PGS accuracy across countries within Europe that are well explained by GD between the "white British" UKB training samples and individuals who descend from different countries in Europe, which to my knowledge, has not been previously demonstrated.

This article provides useful insight into the impact of fine-scale ancestry differences on PGS prediction performance and will be a well received contribution to the growing literature on PGS performance in diverse populations. While I find the relationships between GD and PGS to be very interesting, the results are largely what is to be expected based on previous studies that have investigated the performance of PGS in diverse populations. For example, in 2019 Martin et al. (AJHG 2019; ref. 9 in this paper) conducted an in-depth investigation of the role of demographic history on genetic risk prediction in diverse populations and similarly found that "the utility of polygenic risk scores computed using GWAS summary statistics are dependent on genetic similarity to the discovery cohort." A more recent paper by Martin et al. (Nature Genetics 2019; ref 10 in this paper) assessed accuracy of PGS in diverse populations, and similarly to this paper, they also evaluated accuracy of PGS weights constructed from European ancestry training samples. Martin et al. similarly found the PGS performed the best for individuals of European ancestry, the

worst in African ancestry individuals, and the PGS performance for Hispanic/Latino Americans (who are admixed with European ancestry) was better than both South Asians and East Asians. Indeed Figure 1B in this manuscript is quite similar to Figure 3 in Martin et al. (2019). This submitted paper, however, provides additional insight into the relationships between PGS and ancestry through the novel GD measure, where the authors show how individuals in discrete ancestry groups can be represented by continuous measures of genetic ancestry differences from the training population that is strongly negatively correlated with PGS performance, as shown in Figure 1d. The result from this paper are aligned with a 2020 paper by Bitarello and Mathieson ("Polygenic Scores for Height in Admixed Populations", *G3 Genes|Genomes|Genetics*, Volume 10, Issue 11) who similarly used the height phenotype for assessment and showed that the accuracy of PGS accuracy in AA corresponds to an ancestry gradient. They showed that when PGS weights are constructed using European ancestry samples, the predictive accuracy of PGS increases linearly as the proportion of European ancestry in AA.

I have a few additional comments for the authors.

1. The authors should also evaluate and illustrate the relationship between GD and PGS across populations when using PGS weights constructed using non-European ancestry populations.

2. GD is an essential part of this paper but the authors do not provide sufficient information on how this measure is constructed from the principal components in the Methods section. The authors should provide appropriate material, including formulas, about this.

3. In practice the utility of PRS will depend on the correlation of the predicted PRS and the actual phenotypic value. The authors should provide results of predicted PRS and the actual phenotype values of the individuals across ancestry groups as well as by the GD values.

Author Rebuttals to Initial Comments:

Response to reviewers for 2022-09-15441

Dear Editors,

Please find our responses to the reviewers' comments below. We are excited that all three reviewers found our work important in showcasing empirically that PGS accuracy in humans declines individual to individual both within and across arbitrary ancestry boundaries. The reviewers also raised very insightful critique which allowed us to greatly improve the quality of our manuscript and highlight its novelty. We have addressed all comments and concerns through new analyses and text edits (highlighted in blue in revised manuscript), summarized as follows:

- We include a thorough discussion of our work within the wide body of theory of individual-level genomic prediction reliability and their applications.
- We present new analyses using a new derivation for individual PGS accuracy that considers individual heterozygosity showing similar coupling between accuracy and genetic distance.
- We perform new analyses to investigate PGS accuracy trained in African ancestry individuals and applied to individuals of other ancestries and observe similar coupling between distance and accuracy.

We believe our manuscript to be greatly improved as a result of these changes. For your convenience, we provide point-by-point responses to reviewers' comments below, in blue.

Sincerely,

Bogdan Pasaniuc (on behalf of all co-authors)

Referee #1 (Remarks to the Author)

The authors use the theoretical concept prediction error variance (PEV) of a predicted genetic value for an individual to show that PEV increases the less related an individual is to the individuals that have a phenotype. The authors do not phrase it that way, but in essence this is what their paper is about. The authors show empirically, using two large biobank datasets, that individual prediction accuracy declines with relatedness (genetic distance from the samples with phenotypes) both within and across traditional (but arbitrary) ancestry boundaries, and that the decline is approximately linear. The authors quantify the association between genetic distance and prediction accuracy across a number of traits. They thereby show that prediction error variance greatly increases for genomes that are not well represented in the discovery data, in line with published results (as the authors acknowledge). They do not provide a “solution”, other than state (as many others have done) that more (biobank) data should be collected across the entire spectrum of human genome variation to increase prediction accuracy for individuals irrespective of their genomic ancestry.

We thank the reviewer for an accurate high-level summary of our work. We agree that our approach builds on existing theory coupling genetic relatedness with prediction error with main contribution being empirical. We also thank the reviewer for noting the main contribution of our work: **PGS accuracy declines individual-to-individual within and across traditional genetic ancestry groupings in real-world human population data across US and UK**. Therefore, PGS accuracy can differ vastly for two people from the same ancestry group – and can be very similar for two people from different ancestry groups – depending on their genetic distance to training data. We believe this finding has paramount importance in how PGS should be used in a wide range of fields from social studies to human genetics to personalized medicine.

Please find our point-by-point responses below.

Comments on the Introduction:

1. There are non-genetic prediction tools used in clinical practice that suffer from the same issues that are raised by the authors, yet it seems fashionable to pick out polygenic scores (PGS) as providing an equity barrier. The authors may wish to point out that PGS are not an exception in terms of equitable use of clinical predictors.

We fully agree with the reviewer's suggestions that it is important to acknowledge the low portability issues of both genetic and non-genetic prediction tools. In this work we focused on PGS which have garnered tremendous attention in the past few years across a wide range of fields; examples include embryo selection, clinical decision-making, and socio-genomics studies.

Following reviewer suggestions, we have revised *Introduction and Discussion* to highlight that both genetic and non-genetic prediction tools suffer from portability issues.

Revised Introduction section:

“Such portability issues are also reported for non-genetic clinical models^{15–17}. Here we focus on PGS which have garnered tremendous attention in the past few years across a wide range of fields from embryo selection to clinical decision-making and resource allocation in socio-genomics^{18,19}. ”

Revised Discussion section:

“In the end, we caution that not only polygenic scores, but also traditional clinical risk assessment may suffer from limited portability across diverse populations¹⁷. For example, previous studies show that the Framingham risk score underestimate subclinical atherosclerosis risk for women with a family history of coronary heart disease¹⁵; a traditional clinical breast cancer risk model developed in European population in U.S. overestimated the breast cancer risk among older Korean women¹⁶. Here we focus on genetic prediction portability due to the wide interest and attention from both research community and society. We highlight that the portability of traditional clinical risk factor models in diverse populations is an essential component of health equity and requires thorough investigation.”

2. It is correct that in human genetics, population or sample based average measures of prediction accuracy have been used, for understandable reasons – it is hard enough to explain a point estimate of a PGS to a person, let alone the error variance around it. However, in agriculture, scientists have for decades not only used well-developed theory on prediction error variance, they also routinely provide individual-specific ‘reliability’ metrics for practical applications. My suggestion is that the authors acknowledge this more explicitly and provide more and appropriate references.

The authors hint at ‘relatedness’ (ln260; ln296) and ‘animal breeding theory’ (ln339) but don’t acknowledge that there is a large body of literature out there on prediction error using either pedigree relationships or SNP-based relationships. ‘Animal breeding theory’ is a misnomer because the theory is (hard core) genetical and statistical, it just happens that it was first derived (& applied) in agriculture. All the theory presented in the present paper is a special case of more general results in the literature. Many relevant papers are not in highly specialized journals but in textbooks and mainstream genetics journals. Examples of relevant literature include:

Falconer (1960). An Introduction to Quantitative Genetics (textbook). Has relevant derivations how the accuracy of prediction depends on the relationship with individuals with a phenotype. The simplest case is to predict a genetic value from a single (pedigree) relative with a phenotype. Then $r^2(\text{true and predicted gv}) = \theta^2 * h^2$, with θ = relationship and h^2 = heritability. An extreme example of that was shown empirically by Truong et al. 2020 (Nature Communications), also using UK Biobank data. Those authors show that prediction accuracy from close relatives was equivalent to that from a much larger sample of distant relatives.

Lynch and Walsh (1998). Genetics and Analysis of Quantitative Traits (Chapter 26). The theory about PEV when using a pedigree relationship matrix is the same as when using a GRM (genomic relationship matrix) from SNPs, and the expression simplifies when there are no fixed effects. The relevant matrix that the authors approximate in the current paper is C22 (Eq 26.6], which simplifies to the inverse of $[I + G(\text{inverse})]$, with G the SNP-based GRM. G is dense hence approximations are necessary for large datasets.

Wientjes YC, Veerkamp RF, Bijma P, Bovenhuis H, Schrooten C, Calus MP. Empirical and deterministic accuracies of across-population genomic prediction. *Genetics Selection Evolution*. 2015 Feb 6;47(1):5. doi: 10.1186/s12711-014-0086-0. This paper contains derivations of the accuracy for across-population prediction accuracy, using genomic relationships. Eq[10] in this paper seems most relevant.

Habier D, Fernando RL, Dekkers JC. The impact of genetic relationship information on genome-assisted breeding values. *Genetics*. 2007 Dec;177(4):2389-97. doi: 10.1534/genetics.107.081190. These authors show (using simulations) that “the impact of relationships was greatest for RR-BLUP”, with RR-BLUP their name for the PGS used by Ding and colleagues.

Lee et al. 2017 (Scientific Report); Truong et al. 2020 (Nature Communications). These authors report higher prediction accuracy for individuals who are more closely related to the individuals with a phenotype in the UK Biobank. Although the contrast in this paper is between close (pedigree) relatives and very distantly related individuals, the concept and results are as reported by Ding and colleagues.

Ben Zaabza, H., E. A. Mäntysaari, and I. Strandén. 2020a. Using Monte Carlo method to include polygenic effects in calculation of SNP-BLUP model reliability. *Journal of Dairy Science* 103: 5170-5182. This paper provides derivations and approximation to the PEV when using a SNP-BLUP model (as used by Ding and colleagues).

Indeed, our work builds on extensive theory applied mainly in agricultural domain, with only recent works porting some of this work to human genetics. We thank the reviewer for clearly articulating the difference between population-average vs individual-level metrics of accuracy for genomic prediction; unfortunately, individual-level reliability for genomic prediction (PGS) in humans remains vastly under-explored. Motivated by reviewer suggestions, we thoroughly revised text to properly contextualize our study within existing works.

Revised Introduction section:

“Genetic prediction and its accuracy (or reliability) have been extensively studied in agricultural settings with a focus on breeding programs²⁰⁻²³. At the population level, PGS accuracy can be expressed as a function of heritability, training sample size and the number of markers used in the predictor in single²⁴⁻²⁶ or multi-population settings with or without effect size heterogeneity²⁷. At the individual level, accuracy of genetic prediction from pedigree data²⁸⁻³⁰ can be derived as a function of the inverse of coefficient matrix of mixed models equations whereas accuracy of genetic prediction using whole-genome genetic data can be derived from the same equation, with pedigree matrix replaced with the genomic relationships matrix^{21-23,27,31,32} among training and testing individuals. Simulations guided by dairy breeding programs showcase that genomic prediction accuracy varies with genetic relatedness of the test individual to the training data^{33,34} as well as across generations due to the decay of genetic relationships³⁵.

PGS accuracy in humans has been traditionally assayed using population-level metrics of accuracy (e.g., R^2)^{2,11}. PGS accuracy decays as the target populations are more dissimilar from the training

data either using relatedness^{36,37} or continental/sub-continental groupings of ancestries^{9,10,38}. Notably, PGS reliability decays across groupings of European individuals according to their relatedness to training data³⁶ and across groupings of admixed individuals (e.g., African Americans) according to their proportion of European genetic ancestries³⁹. Recent works showcase that majority of decay in reliability across ancestries could be explained by differential linkage disequilibrium and frequency of causal variants with other sources being heterogeneity in genetic effects due to gene-gene and gene-environment interaction¹². In contrast to existing works in human genetics that focus on population-level metrics of accuracies using ancestry groupings of individuals here we leverage classic theory to investigate the reliability of PGS predictions at the level of single target individual showing that predictive power of PGS varies individual-to-individual within and across populations; for example, PGS accuracy can differ vastly for two people from the same ancestry group – and can be very similar for two people from different ancestry groups – depending on their genetic distance to training data.”

Technical comments:

3. It was not clear to me why the authors use ‘gv’ and not simply ‘g’ (as frequency done in the literature) to parameterise the genetic value.

We thank the reviewer for the suggestion. We have changed our notations to use ‘g’ to denote genetic value and \hat{g} to denote estimated genetic value to follow the previous literature.

4. The authors present their theory and results as if gv is the full genetic value and its variance the full heritability. However, it is only that which is captured by common SNPs and therefore some of the unmodelled error variance is actually genetic. A case in point is the example of neutrophil count. The distinction is important because for some trait the r^2 value reaches ~0.8, which gives the impression that the total genetic value is predicted nearly perfectly, even if the majority of genetic variation might not be captured by SNPs.

*We agree with the reviewer that it is important to distinguish between full genetic value and the genetic value that is captured by common SNPs. To address this, we revised both our *Results* and *Discussion* section to highlight that the error variance contains residual genetic components not captured by common SNPs, and R^2 should be interpreted in terms of the heritability captured by SNPs in the model.*

Revised Results section:

“We note two caveats of individual PGS accuracy: first, the genetic effects are assumed to be the same for all individuals regardless of its genetic ancestry background; second, in practice the SNPs used for PGS training may not capture full trait heritability. Therefore, the metric we proposed here is an upper bound of genetic prediction accuracy (Supplementary Note).”

Revised Discussion section:

“First, it’s important to note the individual PGS accuracy we proposed here is an upper bound of true accuracy and should only be interpreted in terms of the additive heritability captured by SNPs included

in the model. In our derivation, we make the assumption that causal variants and their effects are consistent across all genetic ancestries and that the SNPs in the model capture the full genetic component of a trait. However, substantial missing heritability is observed for complex traits due to the imperfect tagging of causal variants, non-additive genetics and exclusion of rare variants⁶², which decreases the genetic prediction accuracy even for individuals that are from the same genetic ancestry of training data. Population-specific causal variants and effect sizes may further decrease the real accuracy. For example, the prediction accuracy for neutrophil count is overestimated among African American individuals because Duffy-null SNP rs2814778⁵⁴ is not captured in white British training data. Future work could investigate the impact of the population-specific components of genetic architecture on the calibration of PGS accuracy.”

5. Under some assumptions (SNPs effects are random and from a single normal distribution), a SNP-based model is equivalent to a relationship model (VanRaden 2008, J Dairy Science; Goddard 2008, Genetica; Yang et al. 2010, Nature Genetics). This model is also what the authors assume. Therefore, if the authors were to perform a GBLUP analysis then the inverse of the equations (as in Lynch & Walsh chapter 26) gives the PEV and hence the prediction accuracy. The authors approximate the reliability by reducing the dimensionality of the GRM (the eigenvalues of XX' are the same as those of $X'X$), using the top-20 eigenvectors only, and then further by ignoring a shrinkage term (treating the top-20 eigenvectors as fixed effects). Whilst this seems to work fine for individuals who are distantly related (which is picked up easily by the top-20 eigenvectors), it is likely to work less well for another sample from the same population (/ ancestry). This is consistent with Fig4b. Therefore, my suggestion is that the authors explore using the entire GRM to quantify reliability ($J = N$ or M , whichever is smaller) and use shrinkage. My guess is that this will show better concordance between observed and predicted errors. If that is not feasible computationally then the authors could increase J and quantify if/how the results in Fig4b improve.

We thank the reviewer for the suggestions of using the entire GRM and shrinkage to quantify the reliability.

First, we clarify that, we only used GBLUP model to derive the mathematical relationship between prediction reliability and genetic distance rather than compute the reliability directly from the equation. Instead, we used the MCMC sampling implemented in LDpred2 to estimate PEV.

Second, we vary the number of eigenvectors (J) used in genetic distance and evaluate its effect on the correlation between genetic distance and individual PGS accuracy ($cor(d_i, r_i^2)$) in Figure 4b). We find that the correlation between individual PGS accuracy and genetic distance increases as more eigenvectors are included in the calculation of genetic distance when $J \leq 15$, but the improvement plateaus after $J > 15$, which means the top 15 eigenvectors already captures the decay of PGS accuracy well.

Third, due to the computation limitation, instead of computing the full GRM, we compute the average genomic relationship between a testing individual and all training individuals with the following equation:

$$d_i(\text{GRM}) = \sqrt{\frac{1}{K} \sum_{k=1}^K (x_i - x_k)^2},$$

where x_i is the standardized genotype of testing individual and x_k is the standardized genotype of training individual k . As the reviewer predicted, within each ancestry cluster, the GRM based distance predicts the accuracy much better, with the largest improvement observed for “White British” cluster. This holds even when computing GRM with a pruned set of PCA SNPs only. However, this metric relies on individual level training data that are usually not available, we keep using PCA based genetic distance for convenience.

Revised Results section:

“We further evaluated the impact of number of PCs used for calculating genetic distance on its ability to capture accuracy decay. We varied the number of PCs (J) from 1 to 20 and observed that the correlation between genetic distance and individual accuracy ($-\text{cor}(d_i, r_i^2(g_i, \hat{g}_i))$) increases when more PCs are used for computing genetic distance, but no further improvement is observed when $J > 15$ for any ancestry clusters or the whole biobank (Supplementary Figure 4). Therefore, we set $J = 20$ for simplicity. We also explored average squared genetic relationship from training data as an alternative metric of genetic distance and found it is a better prediction of accuracy decay within each genetic ancestry clusters (Supplementary Figure 4). However, this metric relies on individual level training data which are usually not available, we keep using PCA based genetic distance for convenience.”

Supplementary Figure 4. The effect of different metrics of genetic distance on the correlation between genetic distance and accuracy. The y-axis $-\text{cor}(r_i^2, d_i)$ is the correlation between the genetic distance and PGS accuracy; a larger correlation means genetic distance has a better prediction of accuracy. The x-axis are different genetic distance metrics: (1) genetic distance based on PCA with varying number of PCs (from $J=1$ to $J = 20$) and (2) genetic distance based on GRM

using pruned PCA SNPs only or all SNPs in PGS models. The GRM genetic distance is computed as $d_i(\text{GRM}) = \sqrt{\frac{1}{K} \sum_{k=1}^K (x_i - x_k)^2}$, where x_i is the standardized genotype of i_{th} testing individual and x_k is the standardized genotype of k_{th} training individual.

6. The strong inverse correlation between genetic distance and accuracy seems to imply that the reduction in accuracy in samples from ancestries that are different from the GWAS samples is fully accounted for by the distance metric. However, previous studies have shown that MAF and LD do not fully account for the loss in accuracy across ancestries. Can the authors elaborate/discuss how their results compare to those studies?

We thank the reviewer for the suggestion of further comparing our results to previous work. A recent work by Wang et al (Nat Comm, 2020 <https://www.nature.com/articles/s41467-020-17719-y>) quantifies that differences in LD and MAF explain between ~70% and 100% of the reduction of relative accuracy of PGS in AFR ancestry; the remaining unexplained accuracy reduction can be attributed to other factors such as gene-environment interaction, population specific causal variants and heritability differences. In our work, we investigated the correlation between genetic distance and estimated PGS accuracy, which is based on assumptions that the causal variants and effects are the same across ancestries. While there may be differences in the magnitude of the results, our conclusion is in line with Wang et al's work in that both studies agree similarity with the training data is a major contributor to the decline in PGS accuracy, as larger MAF/LD difference results in larger genetic distance (and lower similarity). We leave it as future work to develop more realistic PGS accuracy by modeling population-specific variants and effect sizes.

Revised Discussion section:

“Our finding aligns with previous research by demonstrating that a decrease in similarity (as measured with relatedness, LD/MAF difference, Fst, etc) between testing individuals and training data is a major contributor to the reduction of PGS accuracy. However, practical factors that may impact the transferability such as genotype-environment interaction, population-specific causal variants are not modeled in the calculation of individual PGS accuracy. We leave it as future work to develop more realistic PGS accuracy estimation and investigate its variation with genetic distance.”

7. In addition, the authors state repeatedly that more data should be collected/considered from other ancestries. However, there is emerging evidence (including from some of the authors) that causal effects are shared and similar in size across ancestries. Therefore, in principle at least, discovery of causal variants could be performed in a single ancestry and a PGS based on the causal variants should work across all ancestries (apart from those that are fixed). Can the authors comment on this thought experiment?

We thank the reviewer for suggesting this intriguing thought experiment. Assuming causal effects are shared and similar in size across ancestries, differences in frequency for causal genetic variants will impact the discovery of causal variants and consequently the accuracy of PGS. For example, in the limit, GWAS performed in European ancestries cannot identify any causal variants that have no variation in European but present at considerable frequencies in other ancestries. Second, and most

important, analyzing multi-ancestry data has vastly superior power for statistical fine mapping over single ancestry data (contrasting different LD patterns is extremely powerful towards fine mapping); therefore, even when causal variants are similar across ancestries, differential MAF and LD allow for multi-ancestry studies to outperform (in discovery power and thus prediction using PGS) single-ancestry studies per sample unit.

Revised Discussion section:

“Incorporating diverse populations in the training data can provide advantages for both European and non-European individuals. This is because when the SNPs associated with a trait are observed with different MAF and LD in non-European populations, PGS methods will provide more accurate genetic effects estimation. Furthermore, by broadening the diversity of the training data, PGS models will become more transferable to non-European populations as a result of reduced genetic distance from training data.”

8. In309-313: I found this section very confusing. The only randomness that matters for this paper is in the genotypes (the x-variables)!

We thank the reviewer for the question. To clarify, our paper focuses on the accuracy of PGS at the individual level, where the genotype (x-variable) for a given individual is fixed. It’s essential to assume a random effects model, which implies genetic value g is a random variable, otherwise $cov_{\beta,D}(g_i, \hat{g}_i)$ and $var_{\beta,D}(g_i)$ will be 0. We would like to note the same assumptions are adopted in Wientjes et al 2015 (equation 10) in the development of reliability at the individual.

We fully agree that an alternative and potentially more interesting approach is to treat genotype as a random variable and assume each individual’s genotype is sampled from a distribution. However, this approach would require a more complex modeling of an individual’s genotype as a function of genetic distance. Therefore, we consider it a topic for future work.

Revised methods section:

“We define individual PGS accuracy as the squared correlation between an individual’s genetic value g_i and its PGS estimate \hat{g}_i following the general form in ref²⁸:

$$r_i^2 = \frac{cov_{\beta,D}(g_i, \hat{g}_i)}{var_{\beta,D}(g_i)var_{\beta,D}(\hat{g}_i)} = \frac{var_D(x_i^T \hat{\beta})^2}{var_{\beta}(x_i^T \beta)var_D(x_i^T \hat{\beta})}$$

Here we are interested in the PGS accuracy of a given individual, therefore, the genotype is treated as a fixed variable and genetic effects are treated as a random variable. We note a random effects model is essential otherwise $cov_{\beta,D}(g_i, \hat{g}_i)$ and $var_{\beta,D}(g_i)$ is 0. Under a random effects model both the genetic liability and PGS estimate for individual i are random variables. The randomness of $g_i = x_i^T \beta$ comes from the randomness in β and the randomness of $\hat{g}_i = x_i^T \hat{\beta}$ comes from the randomness of both β and the training data D .”

9. In315-324: It might be helpful to point out to the reader what this result means, i.e., the regression of true on predicted genetic values has a slope of 1.

We thank the reviewer for the suggestion. We added an additional equation in Method section to further interpret the derivation.

Revised methods section:

“The equation 2 also implies the slope from regression of observed phenotypic values (or true genetic liability) on the estimated PGS equal to 1 (Supplementary Figure 11), which offers an alternative way to assess the calibration of PGS in ref^{64,66}.”

$$slope = \frac{cov(x_i^T \hat{\beta}, y_i)}{var(x_i^T \hat{\beta})} = \frac{cov(x_i^T \hat{\beta}, x_i^T \beta + \epsilon_i)}{var(x_i^T \hat{\beta})} = \frac{var(x_i^T \hat{\beta})}{var(x_i^T \hat{\beta})} = 1$$

”

Supplementary Figure 11. Slope of regressing phenotype on PGS is calibrated across genetic ancestry groups in simulation. The PGS model is trained in White British individuals and applied to testing individuals from a diverse genetic background in UKBB. Each boxplot contains 100 points corresponding to the estimated slope by regressing simulated ($h_g^2 = 0.25, p_{causal} = 0.01$) on PGS estimates for all individuals within the genetic ancestry cluster specified by x-axis. The box shows the first, second and third quartile of the 100 slopes, and whiskers extend to the minimum and maximum estimates located within $1.5 \times$ IQR from the first and third quartiles, respectively.

10. Supp Fig3b: why is this relationship non-linear? Would it be more linear if the x-axis is in variance units?

We thank the reviewer for the comment. We change the x-axis into variance unit, but the relationship is still non-linear. Since the y-axis $cor(r_i^2, d_i)$ is bounded at 1, the correlation between accuracy and distance stops increasing when there’s substantial genetic heterogeneity within the cluster.

Supplementary Figure 6. Lower heterogeneity within the genetic ancestry group corresponds to a lower correlation between genetic distance and individual PGS accuracy
 (a) The distribution of correlations between PGS accuracy and genetic distance for 84 traits in ATLAS. (b) The distribution of correlations between PGS accuracy and genetic distance for 84 traits in UKBB. The x-axis is the homogeneity of the genetic ancestry clusters measured as the variance of genetic distance within a genetic ancestry cluster; a larger $var(d_i)$ indicates a larger variation of genetic background. Each boxplot contains 84 points corresponding to the correlation between PGS accuracy and genetic distance within the group specified by x-axis for each of the 84 traits. The box shows the first, second and third quartile of the 84 correlations, and whiskers extend to the minimum and maximum estimates located within $1.5 \times IQR$ from the first and third quartiles, respectively.

11. Supp Fig4: the metric d_i measures genetic distance from a UK Biobank sample. Individuals with different ancestries (e.g., East Asian and African) could have the same d_i values (Fig3b). Yet the genetic (and, perhaps, phenotypic) mean values of East Asian and African populations could be quite different (given that they both have an F_{st} of $\sim 0.12-0.15$ with Europeans). Therefore, I'm not sure what Supp Fig4 shows without a (discrete) ancestry component.

We thank the reviewer for the comment and we fully agree with that the different ancestry clusters can have very different genetic values and phenotypes despite their similar distance to European clusters. We also agree that genetic distance we proposed here is a one-dimension metric that does not capture the complexity of genetic/phenotypic variation across the entire genetic ancestry

continuum. To address the reviewer’s concern, we have re-created the figure within each ancestry cluster and show that even within the same genetic ancestry cluster, PGS and phenotype can still vary with genetic distance thus further showcasing the challenges in interpreting current PGS.

Revised results section:

“We show that GD correlates with PGS and phenotype even within the same genetic ancestry cluster and the correlation patterns vary across clusters (Supplementary Figure 9).”

Supplementary Figure 9. The correlation of PGS/phenotype with genetic distance within each ancestry clusters in UKBB. Only traits that exhibit significant correlation between GD and PGS/phenotype are shown as dots in the figure. The Ashkenazi cluster is not included because no significant correlations are observed.

Referee #2 (Remarks to the Author):

This paper presents a short, clear, and highly important analysis of the portability of polygenic scores as a function of genetic ancestry. The paper is innovative in its methods (showing methods for doing individual-level PGS accuracy prediction) and for its framing (conceptualizing the analysis in terms of continuous genetic distance versus discrete ancestries). The lessons it gives are important for the human genetics community (and allied communities doing trait prediction in other taxa).

The paper is overall very well executed – my main comments are with regards to aspects of the presentation and some comments regarding the approximations used.

We thank the reviewer for highlighting the novelty and well execution of our manuscript and for providing detailed comments which have helped us to greatly improve the quality of our manuscript. Please find our point-by-point responses below.

Major/Technical:

1a. A stated caveat of the method is that it assumes effect sizes are constant across ancestry. It would be helpful to be more clear throughout that for that reason, the r_i^2 estimate is not capturing all sources of error and is likely an upper bound on the accuracy.

We thank the reviewer for raising the potential issue of our homogenous effect size assumption. To fully account for reviewer's concern we provide theoretical justifications for how our proposed metric is an upper bound on genetic prediction accuracy due (1) SNPs included in the model cannot fully explain the total heritability of the trait and (2) effect sizes being different in training vs testing data. We present these results in a Supplementary Note and added text in the Result section to highlight this caveat.

Revised results section:

"We note two caveats of individual PGS accuracy: first, the genetic effects are assumed to be the same for all individuals regardless of its genetic ancestry background; second, in practice the SNPs used for PGS training may not capture full trait heritability. Therefore, the metric we proposed here is an upper bound of genetic prediction accuracy (Supplementary Note)."

1b. Given the current formulation of r_i^2 varies across individuals because of how individuals project on the training set eigenvectors (line 366 and discussion on lines 373-375). The key aspect is the additive genetic distance from the center of the training data's PC space. It would be helpful in the main text to perhaps denote this more clearly. I would propose calling these "Panel-distance r_i^2 " values, such that later someone else might be able to compute "Effect-size-variance r_i^2 " values, and eventually we might hope for a "Multi-factor r_i^2 " value that accounts for both sources of error? This suggestion may not be the ones to use, but I urge some consideration of how to both communicate the scope of the current method as it will be helpful for the future in multiple ways (preventing over-interpretation and making clear where new work will benefit).

We thank the reviewer for proposing better terminology for accuracy. We change the name to panel-distance r_i^2 as the reviewer suggested. We updated the overview of the study section to reflect the change:

“Given that this metric of accuracy is highly dependent on the genetic distance from the training data, we term it as panel-distance r_i^2 .”

1c. Another stated caveat is that the variance of the PGS in the denominator is assumed to be equal to the heritability and constant across ancestry. It would seem straightforward to use an approximation whereby the heritability varies in proportion to an individual-level heterozygosity. While not necessary for publication – this could be a useful improvement in performance. It also raises interesting questions – if an individual is further from the panel but also highly variable – would that moderate the decay of r_i^2 ?

We thank the reviewer for the suggestion to change the denominator in equation 1:

$$1 - \frac{E_D(\text{var}_{\beta|D}(x_i^T \beta))}{\text{var}_{\beta}(x_i^T \beta)} \quad (\text{equation 1})$$

First, instead of using a constant heritability to approximate the denominator $\text{var}_{\beta}(x_i^T \beta)$ in equation 1, we take the following steps to compute the denominator $\text{var}_{\beta}(x_i^T \beta)$ for each individual via Monte-Carlo integration:

- We sample β from its prior distribution

$$\beta_m \sim \begin{cases} N\left(0, \frac{\widehat{h}_g^2}{q_m(1-q_m)M\widehat{p}}\right) & c_j = 1, \text{ with probability } \widehat{p} \\ 0 & c_j = 0, \text{ with probability } 1 - \widehat{p} \end{cases}$$

where \widehat{p} and \widehat{h}_g^2 are estimated polygenicity and heritability from LDpred2 and q_m is the minor allele frequency of SNP m in the training data.

- For each sample of β^s , we compute $x_i^T \beta^s$ and compute $\text{var}(x_i^T \beta^s)$ across 1000 samples as an estimate of $\text{var}_{\beta}(x_i^T \beta)$

The new denominator is proportional to the sum of squared genotype $x_i^T x_i$ of an individual.

We used real data from UKBB to explore this new approach using PGS for height as example (Supplementary Figure 10); we find that the new metric recapitulates the decay in estimated PGS accuracy albeit the correlation is slightly reduced. This could be an effect of the approximations used in genetic distance computation not matching the Monte-Carlo estimation of the prior variance for a given individual. Although this direction holds great promise in further improving our understanding between genetic distance/relatedness to PGS accuracy, we leave the alternative accuracy metric for future work.

We add the following to the Discussion:

“Second, we approximate the variance of genetic liability in the denominator of equation 1 with heritability and set the value fixed for all individuals. Preliminary results show that replacing the denominator with a Monte Carlo estimation of genetic liability variance that are specific to each individual still recapitulate the accuracy decay in estimated PGS accuracy albeit the correlation is slightly reduced (Supplementary Figure 10). We leave the new accuracy metric for future investigation.”

Supplementary Figure 10. Comparing two types of individual accuracy for height in UKBB. (a) Accuracy computed from equation (1), with $\text{var}_\beta(x_i\beta)$ set as fixed heritability; (b) Accuracy computed from equation (1), with $\text{var}_\beta(x_i\beta)$ estimated from Monte Carlo sampling from prior distribution of β . (c) Empirical accuracy estimated as the squared correlation between PGS and height for each genetic distance bin. Both (a) and (b) reflect the decreasing trend of empirical accuracy in (c).

2. Line 372-373: A theoretical result is presented here that is underplayed in the manuscript. The theory predicts that the performance will decay with genetic distance. And the main paper's empirical analysis supports the theory. I think the authors should consider promoting the theoretical result to the main paper – as it currently reads, the r_i^2 and GD result is an empirical observation, but a careful read shows it is both a theoretical expectation and an empirical observation.

We thank the reviewer for highlighting our contribution of connecting PGS performance with genetic distance. We moved the equation to the overview of the study to highlight the theory.

Revised results section:

“Under an infinitesimal assumption where all variants are causal and drawn from a normal distribution $N(0, \sigma_\beta^2)$, and analytical form of PGS accuracy can be derived as:

$$r_i^2(g_i, \hat{g}_i) = 1 - \frac{\sigma_e^2 \sum_{j=1}^J \frac{1}{\lambda_j} x_i^T v_j v_j^T x_i}{\sigma_\beta^2 x_i^T x_i} = 1 - \frac{\sigma_e^2 \sum_{j=1}^J \frac{1}{\lambda_j} x_i^T v_j v_j^T x_i}{x_i^T x_i}$$

where σ_β^2 is per SNP heritability, σ_e^2 is the variance of residual environmental noise, and v_j and λ_j are the j_{th} eigenvector of training genotype data. The term $\sum_{j=1}^J \frac{1}{\lambda_j} x_i^T v_j v_j^T x_i$ is the Mahalanobis distance of the testing individual i from the center of the training genotype data on its PC space and $x_i^T x_i$ is the sum of squared genotypes across all variants. Empirically, the ratio between the two is highly correlated with the Euclidean distance of the individual from the training data on that PC space ($R=1$, P -value $< 2.2e-16$ in UKBB). ”

3. Supp Fig 1a : the results show a high correlation but the scaling of the two axes is very different because covariance is used on the y-axis as opposed to the correlation – It would be nice to have an additional panel where the y-axis is the empirical correlation and where any bias away from the $y=x$ for r_i^2 can be observed directly.

We thank the reviewer for pointing out the scaling issue. We mislabeled the y-axis as $cor(\widehat{PGS}, gv)$, which should be $cor(\widehat{PGS}, gv)^2$ ($cor(\hat{g}, g)^2$ after updating the annotation). We corrected the mislabeling and we added the $y=x$ identity line to show the bias in individual PGS accuracy, which shows downward bias for individuals with low accuracy (those that are distant from training data).

Supplementary Figure 1. The individual level accuracy is highly correlated with population level accuracy. All UKBB testing individuals are divided into 100 bins based on their genetic distance. The x-axis is the average individual-level PGS accuracy for the individuals within the bin and the y-axis is (a) the squared correlation between simulated genetic liability and PGS estimates for the individuals within the bin (b) the squared correlation between simulated phenotype and PGS estimates. The dot and error bars represent the mean and ± 1.96 s.e.m from 100 simulations.

4. Figure 3A: It would be helpful to report the slopes in absolute units (decay in \hat{r}_i^2 per unit of genetic distance). As it is, because R depends on the variances it's difficult to know how much of a decay in performance one gets in terms of accuracy.

We thank the reviewer for the suggestion. We've updated Figure 3 to include the regression equation to predict \hat{r}_i^2 from d_i .

Revised Figure 3:

Figure 3. The individual-level accuracy for height PGS decreases across the genetic ancestry continuum in ATLAS. (a) Individual PGS accuracy decreases within both homogenous and admixed genetic ancestry clusters. Each dot represents a testing individual from ATLAS. For each dot, the x-axis represents its distance from the training population on the genetic continuum; the y-axis represents its PGS accuracy. The color represents the inferred genetic ancestry cluster. R and p refer to the correlation between genetic distance and individual-level PGS accuracy and its significance from two-sided t-tests. (b) Individual PGS accuracy decreases across the entire ATLAS. (c) Population-level PGS accuracy decreases with the average genetic distance in each genetic distance bin. All ATLAS individuals are divided into 20 equal-interval genetic distance bins. The x-axis is the average genetic distance within the bin, the y-axis is the squared correlation between PGS and phenotype for individuals in the bin; The dot and error bar show mean and 95% confidence interval from 1000 bootstrap samples. (EA, European American; HL, Hispanic/Latino American; SAA, South Asian American; EAA, East Asian American; AA, African American.)

5. Line 372: The use of J here is confusing given its use in the genetic distance calculation (where J was fixed to 20). Here do you mean a complete eigendecomposition over all eigenvectors? (i.e. no truncation is implied and any truncation for a fixed J would be an approximation).

We thank the reviewer for the question on the choice of eigenvectors. We clarify that we used truncated SVD implemented in flashpca2 and set the number of eigenvectors as 20. In the original manuscript we used $J=20$ as it provided good correlation with accuracy. In the revision, we thoroughly investigated the choice of J on the genetic distance's ability to capture accuracy decay. We vary J from 1 to 20 when computing genetic distance and evaluate its effect on the correlation between genetic distance and individual PGS accuracy within each genetic ancestry cluster and the whole biobank. We observed that the correlation between genetic distance and individual accuracy $cor(d_i, r_i^2)$ increases when more PCs are used for computing genetic distance, but no further improvement is observed when $J > 15$. This hold both for analyses performed within each genetic ancestry cluster as well as including the whole biobank (Supplementary Figure 4). For simplicity, we keep $J = 20$ in the manuscript. Please see answer to Reviewer 1 Comment 5.

Revised results section:

“We further evaluated the impact of number of PCs used for calculating genetic distance on its ability to capture accuracy decay. We varied the number of PCs (J) from 1 to 20 and observed that the correlation between genetic distance and individual accuracy ($-cor(d_i, r_i^2(g_i, \hat{g}_i))$) increases when more PCs are used for computing genetic distance, but no further improvement is observed when $J > 15$ for any ancestry clusters or the whole biobank (Supplementary Figure 4). Therefore, we set $J = 20$ for simplicity. We also explored average squared genetic relationship from training data as an alternative metric of genetic distance and found it is a better prediction of accuracy decay within each genetic ancestry clusters (Supplementary Figure 4). However, this metric relies on individual level training data which are usually not available, we keep using PCA based genetic distance for convenience.”

Supplementary Figure 4. The effect of different metrics of genetic distance on the correlation between genetic distance and accuracy. The y-axis $-cor(r_i^2, d_i)$ is the correlation between the genetic distance and PGS accuracy; a larger correlation means genetic distance has a better

prediction of accuracy. The x-axis are different genetic distance metrics: (1) genetic distance based on PCA with varying number of PCs (from J=1 to J = 20) and (2) genetic distance based on GRM using pruned PCA SNPs only or all SNPs in PGS models. The GRM genetic distance is computed as $d_i(\text{GRM}) = \sqrt{\frac{1}{K} \sum_{k=1}^K (x_i - x_k)^2}$, where x_i is the standardized genotype of i_{th} testing individual and x_k is the standardized genotype of k_{th} training individual.

Minor (Grammatical / Clarity):

6. Line 40: Strike “genetic value” – the social connotation of the phrase makes it worth diminishing / removing it from technical use in genetics

We thank the reviewer for raising this potential issue of terminology. We changed genetic value into genetic liability as suggested by the reviewer in the manuscript.

Line 84: Insert >the< before “continuum”

We’ve added proper articles for all occurrences of “continuum”.

Line 85: “genetic ancestry cluster” -> “genetic ancestry clusters”

We’ve changed “individuals unassigned to a given genetic ancestry clusters” to “individuals unassigned to any given genetic ancestry clusters”

Line 105: Notation gv_i confusing – I was puzzling if it was a multiplication of g and v_i and then eventually reverse engineered that it is the underlying additive genetic liability for the trait. That term is being called genetic value here, but as mentioned above I think we should invent new terms to get around the “value” phrasing. My encouragement is to use “genetic liability” or “additive genetic liability” to be more specific. While liability is traditionally reserved for discrete trait models – it’s only a mild abuse of the terminology to use it for both discrete and continuous traits (many readers will probably not even notice the distinction and you could precisely introduce the notation where it’s essential). If that is not comfortable, another idea would be to exploit the relationship to generalized linear models – in that literature the term “linear predictor” and Greek letter eta are used for what stands in for “genetic value” here.

We thank the reviewer for the suggested alternative terminologies. We’ve changed “genetic value” to “additive genetic liability” and “gv” to “g” to avoid confusion.

Line 112: Especially good to change notation from gv_i given the use of v_i here for eigenvectors.

We’ve changed “gv_i” to “g_i” to avoid confusion.

Line 112: Choice of J would be useful to declare in the main text.

We thank the reviewer for the comment. We’ve clarified the choice of J in the *Results* section:

“We further evaluated the impact of number of PCs used for calculating genetic distance on its ability to capture accuracy decay. We varied the number of PCs (J) from 1 to 20 and observed that the correlation between genetic distance and individual accuracy ($-\text{cor}(d_i, r_i^2(g_i, \hat{g}_i))$) increases when more PCs are used for computing genetic distance, but no further improvement is observed when $J > 15$ for any ancestry clusters or the whole biobank (Supplementary Figure 4). Therefore, we set $J = 20$ for simplicity. We also explored average squared genetic relationship from training data as an alternative metric of genetic distance and found it is a better prediction of accuracy decay within each genetic ancestry clusters (Supplementary Figure 4). However, this metric relies on individual level training data which are usually not available, we keep using PCA based genetic distance for convenience.”

Also see Reviewer 1 Comment 5 and Reviewer 2 Comment 5.

Line 118: Insert >a< in “as function”

Line 125: Fix “the of”

Line 133: “under >a< discrete”

Line 170: “in the >the< HL cluster”

Line 212: “different correlations >for< PGS vs trait”

Line 217: Fix within /in

Line 221: “along GD” -> “with respect to GD”

Line 221: “>the< European American cluster”

We thank the reviewer for pointing out these typos and we have fixed in the revised manuscript.

Referee #3 (Remarks to the Author)

Polygenic scores (PGS) have recently become a popular approach for assessing predicted risk of clinical outcomes based on genetic variants identified through GWAS, where the weighting of variants used in PGS are usually proportional to variant effect sizes from the GWAS. PGS have been shown to not be transferable across ancestrally diverse populations, with a strong bias when applied to non-European ancestry populations when PGS weights are calculated from discoveries from GWAS that have predominantly been conducted in European ancestry populations.

In this study, the authors investigate how the accuracy of PGS changes based on a proposed genetic distance (GD) measure between an individual and the population from which the PGS weights were constructed, where the GD is essentially a measure of genetic ancestry differences between groups and is based on continuous measures of ancestry only axes of variation that are reflected from a principal components analysis. In this study, the authors construct PGS weights for a variety of phenotypes using individuals who self-reported to be "white British" in the UK Biobank (UKB). They then assess the prediction accuracy of PGS using groups of individuals with difference ancestries/ethnicities in UKB as well as from the Los Angeles biobank at UCLA (ATLAS). The authors find a strong negative association between PGS performance and their novel GD measure. They also identify differences in PGS accuracy across countries within Europe that are well explained by GD between the "white British" UKB training samples and individuals who descend from different countries in Europe, which to my knowledge, has not been previously demonstrated.

This article provides useful insight into the impact of fine-scale ancestry differences on PGS prediction performance and will be a well-received contribution to the growing literature on PGS performance in diverse populations. While I find the relationships between GD and PGS to be very interesting, the results are largely what is to be expected based on previous studies that have investigated the performance of PGS in diverse populations. For example, in 2019 Martin et al. (AJHG 2019; ref. 9 in this paper) conducted an in-depth investigation of the role of demographic history on genetic risk prediction in diverse populations and similarly found that "the utility of polygenic risk scores computed using GWAS summary statistics are dependent on genetic similarity to the discovery cohort." A more recent paper by Martin et al. (Nature Genetics 2019; ref 10 in this paper) assessed accuracy of PGS in diverse populations, and similarly to this paper, they also evaluated accuracy of PGS weights constructed from European ancestry training samples. Martin et al. similarly found the PGS performed the best for individuals of European ancestry, the worst in African ancestry individuals, and the PGS performance for Hispanic/Latino Americans (who are admixed with European ancestry) was better than both South Asians and East Asians. Indeed Figure 1B in this manuscript is quite similar to Figure 3 in Martin et al. (2019). This submitted paper, however, provides additional insight into the relationships between PGS and ancestry through the novel GD measure, where the authors show how individuals in discrete ancestry groups can be represented by continuous measures of genetic ancestry differences from the training population that is strongly negatively correlated with PGS performance, as shown in Figure 1d. The result from this paper are aligned with a 2020 paper by Bitarello and Mathieson ("Polygenic Scores for Height in Admixed Populations", G3 Genes|Genomes|Genetics, Volume 10, Issue 11) who similarly used the height phenotype for assessment and showed that the accuracy of PGS accuracy in AA corresponds to an ancestry

gradient. They showed that when PGS weights are constructed using European ancestry samples, the predictive accuracy of PGS increases linearly as the proportion of European ancestry in AA.

We thank the reviewer for summarizing our work. Standard practice for assessing PGS accuracy in humans (including all works cited by reviewer) uses population-level metrics of accuracy (e.g., R^2) that provide an average accuracy across individuals within a genetic ancestry grouping. This is a major shortcoming as genetic ancestry groupings in humans are arbitrary. For example, Martin et al Nat Gen 2019 groups together individuals of various Native American, European and African ancestries into a single “Admixed American (AMR)” category yielding a single PGS performance metric for all individuals in that category. As reviewer keenly notes, Bitarello and colleagues focus on this shortcoming showing that in African Americans, where the genome-wide proportion of African/European genetic ancestries varies individual-to-individual, PGS error correlates with genetic ancestries across groupings of individuals with different proportions of African/European ancestries. Notably, Bitarello and colleagues also rely on population-level metrics of accuracy and have to compare different groupings of African American individuals (either by cohort, or by proportion of genome-wide ancestries).

Building on previous literature, we consider PGS performance at the level of a single target individual (rather than a population) as a function of genetic distance to training data. In doing so, we are able to overcome several major shortcomings of arbitrariness in genetic ancestry groupings in humans for PGS performance evaluation to show:

- PGS performance decays individual-to-individual with genetic distance from training data in all populations; this is a direct consequence of fine-scale structure of human populations.
- Decay of PGS performance across genetic ancestries is not only a feature of traditionally-labeled admixed populations (e.g., African American; LatinX, etc) but occurs in all individuals irrespective of genetic ancestries. For example, even in Europeans (the most studied genetic ancestry grouping) we observe a decay in PGS performance.
- We connect the statistical genetics theory (developed mostly in agriculture field) with empirical data in humans showing that the impact of genetic distance (relatedness) on PGS performance is extremely large and needs to be accounted for.

I have a few additional comments for the authors.

1. The authors should also evaluate and illustrate the relationship between GD and PGS across populations when using PGS weights constructed using non-European ancestry populations.

We thank the reviewer for suggesting training in non-Europeans to replicate our findings. We train PGS in African individuals (Nigeria and Caribbean, $N_{\text{train}} = 5000$) and apply the model in the remaining testing individuals. Limited by the sample size of non-European ancestry individuals in UKBB, we performed simulations with 56,539 SNPs on chromosome 10 and simulated high signal-to-noise traits with a high heritability ($h_g^2 = 0.8$) and low polygenicity ($p_{\text{causal}} = 0.01$ and 0.001). We also conduct PCA on chromosome 10 of the 5000 African training individuals and project each testing individuals on the PC space to compute its Euclidean genetic distance from African training data. The

conclusions we draw from using European individuals as training data still hold when we use Africans as training data:

- First, the coverage of 90% credible interval doesn't vary across genetic distance, despite slight miscalibration.
- Second, the credible interval width increases and individual PGS accuracy decreases when the testing individual is further away from the African training data, which reflect the trend in empirical accuracy computed as squared correlation between PGS and genetic value.

We updated the Result section to reflect the new results:

"We also replicated our findings by using the African continental ancestry cluster (Nigeria and Caribbean) as training data. Limited by the training sample size of the African cluster, we simulated a high signal-to-noise trait by set $h_g^2 = 0.8$ and proportion of causal variants $p_{causal} = 1\%$ and 0.1% with 56,539 SNPs on chromosome 10 only. We train PGS models on 5,000 African individuals and apply it to the remaining testing individuals. The coverage of 90% credible interval is invariant to genetic distance despite slight miscalibration. The credible interval width increases and individual PGS accuracy decreases when the testing individual is further away from the African training data. This trend is consistent with the observed decrease in empirical accuracy computed as squared correlation between PGS and genetic value as genetic distance increases (Supplementary Figures 2 and 3)."

Supplementary Figure 2. PGS performance varies across genetic distance in simulations using Africans as training data ($h_g^2 = 0.8$ and $p_{causal} = 0.1\%$). (a) The coverage of the 90% credible intervals is approximately uniform across testing individuals at all genetic distances. The red dotted line represents the expected coverage of 90% credible interval. Each dot represents a randomly selected UKBB testing individual. For each dot, the x-axis is its genetic distance from African training data, the y-axis is the empirical coverage of 90% credible interval calculated as the proportion of simulation replicates where the 90% credible intervals contain the individual's true genetic liability, and the error bars represent mean ± 1.96 standard error of the mean (s.e.m) of the empirical coverage calculated from 100 simulations. (b) The width of 90% credible interval increases with genetic distance. For each dot, the y-axis is the width of 90% credible interval across 100 simulation replicates, and the error bars represent ± 1.96 s.e.m. (c) Individual PGS accuracy decreases with genetic distance. For each dot, the y-axis is the average individual level PGS accuracy across 100 simulation replicates, and the error bars represent ± 1.96 s.e.m. (d) Population-level

metrics of PGS accuracy recapitulates the decay in PGS accuracy across genetic continuum. All UKBB testing individuals are divided into 100 equal-interval bins based on their genetic distance. The x-axis is the average genetic distance for the bin and the y-axis is the squared correlation between genetic liability and PGS estimates for the individuals within the bin. The dot and error bars represent the mean and ± 1.96 s.e.m from 100 simulations.

Supplementary Figure 3. PGS performance varies across genetic distance in simulations using Africans as training data ($h_g^2 = 0.8$ and $p_{causal} = 1\%$). (a) coverage of the 90% credible intervals is approximately uniform across testing individuals at all genetic distances. (b) The width of 90% credible interval increases with genetic distance. (c) Individual PGS accuracy decreases with genetic distance. (d) Population-level metrics of PGS accuracy recapitulates the decay in PGS accuracy across genetic continuum.

2. GD is an essential part of this paper but the authors do not provide sufficient information on how this is measure is constructed from the principal components in the Methods section. The authors should provide appropriate material, including formulas, about this.

We thank the reviewer for bringing this to our attention. We've updated the method section and provided scripts for computing genetic distance.

Revised methods section:

“Genetic Distance. *The genetic distance is defined as the Euclidean distance between a target individual and the center of training data on the PC space of training data.*

$$d_i = \sqrt{\sum_{j=1}^J (x_i^T v_j - \bar{x}_{train} v_j)^2} = \sqrt{\sum_{j=1}^J (x_i^T v_j)^2}$$

where d_i is the genetic distance of a testing individual i from the training data, x_i is an $M \times 1$ standardized genotype vector for testing individual i , v_j is the j_{th} eigenvector for the genotype matrix of training individuals, \bar{x}_{train} is the average genotype in training population ($\bar{x}_{train} v_j = 0$ given that the genotypes are centered with respect to the allele frequency in training population) and J is set to 20. ”

“Genetic distance from PGS training data. *To compute the genetic distance of testing individuals from the training population, we perform PCA on the 371,018 UKBB white British training individuals and project the 48,586 UKBB testing individuals and 36,778 ATLAS training individuals on the PC space. We start from the 979,457 SNPs that are overlapped in UKBB and ATLAS. First, we perform LD pruning with plink2 (--indep-pairwise 1000 50 0.05) and exclude the long-range LD regions. Next, we perform PCA analysis with flashpca2⁶⁹ on the 371,018 UKBB white British training individuals to obtain the top 20 PCs. Then, we project the remaining 48,586 UKBB individuals that are not included in the training data and 36,778 ATLAS individuals onto the PC space of training data by using SNP loadings (--outload loadings.txt) and their means and standard deviations (--outmeansd meansd.txt) output from flashpca2. In the end, we compute the genetic distance for each individual as the Euclidean distance of its PCs from the center of training data with the equation: $d_i = \sqrt{\sum_{j=1}^{20} (pc_{ij})^2}$, where pc_{ij} is the j_{th} principal component of individual i . (<https://github.com/yidingdd/individual-pgs-accuracy/tree/main/genetic-distance>)”*

3. In practice the utility of PRS will depend on the correlation of the predicted PRS and the actual phenotypic value. The authors should provide results of predicted PRS and the actual phenotype values of the individuals across ancestry groups as well as by the GD values.

We thank the reviewer for raising the concern that utility of PRS based on its correlation with phenotypic value. As the reviewer suggested, we compute the empirical accuracy ($cor(\hat{g}, y)^2$, squared correlation between PGS and phenotype) for each genetic ancestry/genetic distance bins for all 84 traits and show that the accuracy decreases across genetic distance for both cases, while the latter provide a more continuous decreasing trend.

We updated the Result section to include the new analyses:

“Empirical analyses of PGS accuracy also show the same trend. We group individuals by ancestry clusters and compute the empirical accuracy as the squared correlation between PGS and residual phenotype for 84 traits. Averaging across 84 traits, we find the empirical accuracy decreases with increases genetic distance as reported by previous studies³⁸. Further analyses based on genetic distance bins show the decreasing trend at a finer scale (Supplementary Figure 7).”

Supplementary Figure 7. Empirical PGS accuracy decreases with genetic distance in UKBB averaged across 84 traits. (a) Empirical PGS accuracy decreases across subcontinental ancestries. (b) Empirical PGS accuracy decreases across bins of genetic distance. The x-axis is the average genetic distance for all individuals within each genetic ancestry cluster/genetic distance bin; the y-axis is the relative accuracy compared to the White British cluster. The dot and error bar show mean and ± 1.96 standard error of the mean across 84 traits.

Reviewer Reports on the First Revision:

Referees' comments:

Referee #1 (Remarks to the Author):

The authors have done a nice & thorough job in the revision, addressing my technical questions and concerns.

Referee #2 (Remarks to the Author):

The detailed attention to my comments is appreciated -- Excellent work!

Referee #3 (Remarks to the Author):

The authors have been responsive to reviewers' comments and questions from the original submission. I find the revised manuscript to be improved and the additional experiments and material to be very insightful. There are still a few issues that need to be addressed. Below are some comments for the authors on the revised manuscript.

Comments:

1. I think the authors need to clearly define "ancestry" (continental, sub-continental, fine-scale, etc) in the manuscript. In addition, the authors should distinguish "ancestry" from "nationality" or "country of origin" for clarity in various places. For example, in Figure 1 the authors use ancestry labels based on continental-level ancestry groupings that are widely used and accepted (African, East Asian, European, Hispanic/Latino American, South Asian, et.). However, in Supplementary Figure 4, the authors define categories of "ancestry" based on nationality: "White British", "Poland", "Italy", "Iran", "India" etc. I don't think that "nationality" should be interchanges with "ancestry." This lack of clarity about how ancestry is defined is also evident in a response to a reviewer's comments. For example, in response to reviewer 1's comment on page 2 in the response to reviewers document, the authors state: "Therefore, PGS accuracy can differ vastly for two people from the same ancestry group – and can be very similar for two people from different ancestry groups – depending on their genetic distance to training data." I think it is would be more appropriate to say that individuals who have the same nationality (or country of origin) can have very different ancestries, and similarly, individuals from different countries or nationalities can have similar ancestries.

2. In response to my comment to the authors that previous studies have demonstrated that PRS accuracy depends on genetic similarity between the training sample and the target population, the authors state that the major advancement of their work is that they consider PGS performance at the level of a single target individual rather than a population as a function of genetic distance to training data. I think this is a valid point. However, the training data used for PGS are from a group of individuals who can have quite variable genetic ancestry ancestries. So this approach proposed by the authors still relies on clustering individuals into a single discrete genetic ancestry group (for training), which I think is a limitation that is not solved by looking at individual level data in the target sample. I think the authors should include some discussion about this.

3. Related to comment 2 above, how does variability in genetic ancestry in the training data impact prediction accuracy? It would be good for the authors to provide some insight or discussion

on this.

4. It is a bit perplexing to me to construct an African PGS by using a combined sample of individuals from Nigeria and from Caribbean, which are two very different populations, the latter of which are admixed. Additional insight on this would be useful. Also, would predication accuracy be different if individuals using for the PGS training were from a single country versus multiple nationalities? What are advantages/disadvantages of using individuals from different countries with ancestry derived from the same continent.

5. In order to assess the practical utility of PRS, I asked the reviewers to provide correlation values of the predicted PRS and actual phenotype values. In response to this request, the authors have provided Supplementary Figure 7 that includes figures with "Relative accuracy" as compared to the White British group. This is insufficient as relative accuracy does not allow for assessment of PRS practicality. The authors need to provide updated or additional figures with absolute correlation values of predicted PRS and phenotype values as the response variable, as opposed (or in addition to) relative accuracy values.

6. Supplementary Figures 2 and 3 need to be improved. The authors are using similar colors for different countries and it is very difficult to decipher what points correspond to which countries. For example, Italy and the Caribbean are both plotted in green, while Poland, Nigeria, and Iran are all plotted in blue. Different color schemes need to be used for improved clarity.

Author Rebuttals to First Revision:

Response to reviewers for 2022-09-15441

Dear Editors,

Please find our responses to the reviewers' comments below. We thank the reviewers and editors for their suggestions to improve the clarity of our manuscript. We have made three main improvements to the manuscript:

- We properly define the concept of genetically inferred ancestry (GIA) as we use it in our work. To further emphasize this we also replaced country names in all plots with GIA labels.
- We note that the African ancestry data sets used in our work are not representative of all the genetic diversity of Africa.
- We clarify that we lumped together two distinct GIA (Caribbean and Nigerian) only for the purposes of a simulation to attain sufficient sample size for PGS construction; we note in text the caveats of this simulation setup.

For your convenience, we provide point-by-point replies below, in blue font, to each comment raised by reviewers.

Sincerely,

Bogdan Pasaniuc (on behalf of all co-authors)

Editor

Wrt the comment made about construction of an African ancestry PGS with a Nigerian and Caribbean dataset we'd also ask that you add a comment on how these are representative (or not) of all African ancestries (this is not explicitly mentioned by the referee, but would also be important in our opinion)

We thank both Reviewer 3 and the editors for raising the important issue to allow us to clarify this point. We clarify in text that the combined data set is not representative of all African ancestries.

To expand, we had to combine two distinct genetically inferred ancestries (Caribbean and Nigerian) for the purposes of simulation of a PGS to attain sufficient training sample size. Our simulation setup does not include population stratification and thus is robust to genetic stratification; importantly, we do not analyze any real trait data using this combined data set.

We clarify these points in text as follows.

Updated Introduction section:

“The concept of genetic ancestry can have different meanings depending on the context and is often conflated with social constructs such as nationality, race, and ethnicity. In this work, we use ancestry as genetically inferred ancestry (GIA), which describes the genetic similarity between individuals and reference populations (e.g., 1000 Genomes) as inferred by methods such as PCA; GIA used in our work depend on the choice of reference data and inference methods and do not represent the full genetic diversity of human populations.”

Updated Results section:

“To demonstrate that the continuous accuracy decay is not specific to PGS models trained on European ancestries, we conducted further analyses using a non-European training dataset composed of individuals of NG and CB GIA (we grouped the two GIA to attain sufficient sample size for simulations).”

Referee #1 (Remarks to the Author):

The authors have done a nice & thorough job in the revision, addressing my technical questions and concerns.

We thank the reviewer for their insightful critiques.

Referee #2 (Remarks to the Author):

The detailed attention to my comments is appreciated -- Excellent work!

We thank the reviewer for commending our work.

Referee #3 (Remarks to the Author):

The authors have been responsive to reviewers' comments and questions from the original submission. I find the revised manuscript to be improved and the additional experiments and material to be very insightful. There are still a few issues that need to be addressed. Below are some comments for the authors on the revised manuscript.

We thank the reviewer for noting the improvement of our manuscript. We also appreciate the questions below that allow us to clarify and improve our manuscript.

Comments:

1. I think the authors need to clearly define "ancestry" (continental, sub-continental, fine-scale, etc) in the manuscript. In addition, the authors should distinguish "ancestry" from "nationality" or "country of origin" for clarity in various places. For example, in Figure 1 the authors use ancestry labels based on continental-level ancestry groupings that are widely used and accepted (African, East Asian, European, Hispanic/Latino American, South Asian, et.). However, in Supplementary Figure 4, the authors define categories of "ancestry" based on nationality: "White British", "Poland", "Italy", "Iran", "India" etc. I don't think that "nationality" should be interchanges with "ancestry." This lack of clarity about how ancestry is defined is also evident in a response to a reviewer's comments. For example, in response to reviewer 1's comment on page 2 in the response to reviewers document, the authors state: "Therefore, PGS accuracy can differ vastly for two people from the same ancestry group – and can be very similar for two people from different ancestry groups – depending on their genetic distance to training data." I think it is would be more appropriate to say that individuals who have the same nationality (or country of origin) can have very different ancestries, and similarly, individuals from different countries or nationalities can have similar ancestries.

We appreciate the reviewer's comment on the lack of clarity in our use of ancestry terminology. We fully agree with the reviewer that nationality and ancestry are not interchangeable terms. As the reviewer highlights, these concepts are deeply intertwined with the concepts of genetic ancestry, race, and ethnicity (e.g., Peterson et al. *Cell* 2019, <https://doi.org/10.1016/j.cell.2019.08.051>).

Our work investigates genetically inferred ancestries (GIAs) and uses the term "ancestry" interchangeably with GIA. GIA is defined as a group of individuals that are genetically similar with a set of reference genomes whose ancestries are known (e.g., 1000 Genomes). We infer GIA through principal component analysis followed by clustering joint with 1000 Genomes data; we use the Prive et al AJHG 2022 GIA definitions (<https://pubmed.ncbi.nlm.nih.gov/35120604/>)

We fully agree with reviewer that the GIA is distinct from nationality/race/ethnicity or other social constructs and should never be used interchangeably.

Motivated by the reviewer's suggestion, we have updated the manuscript to clearly distinguish between genetic ancestry (GIA) and social constructs:

- We explicitly introduce GIA as distinct from social constructs in the Introduction.
- We replace "ancestry" with "GIA" in most locations in our manuscript.
- We clarify that GIA is not meant to be representative of all genetic diversity within the respective country/region but serves as a sample that is genetically similar with the individuals from that region.
- We relabeled all GIA labels in UK Biobank data from country names to identifiers to clearly emphasize that they are genetically inferred ancestries not meant to be representative of the full genetic diversities within those regions; "White British" -> "WB", "Poland" -> "PL", "Italy" -> "IT", "Iran" -> "IR", "Ashkenazi" -> "AJ", "India" -> "IN", "China" -> "CH", "Caribbean" -> "CB", "Nigeria" -> "NG".
- We updated the methods section to provide detailed description of GIA inference procedure.

Revised Introduction section:

"The concept of genetic ancestry can have different meanings depending on the context and is often conflated with social constructs such as nationality, race, and ethnicity. In this work, we use ancestry as genetically inferred ancestry (GIA), which describes the genetic similarity between individuals and reference populations (e.g., 1000 Genomes) as inferred by methods such as PCA; GIA used in our work depend on the choice of reference data and inference methods and do not represent the full genetic diversity of human populations."

Revised methods section:

"Ancestry ascertainment in UKBB. The UKBB individuals are clustered into nine sub-continental ancestry clusters WB("White British"), PL("Poland"), IR("Iran"), IT("Italy"), AS("Ashkenazi"), IN("India"), CH("China"), CB("Caribbean") and NG("Nigeria") based on the top 16 precomputed PCs (Data-Field 22009) as described in ref³⁸. First, UKBB participants are grouped by country of origin (Data-Field 20115) and the center of each country on the PC space is computed as the geometric medians for all countries, which serves as a proxy of center for each sub-continental ancestry. The center of Ashkenazi GIA is determined using a dataset from ref⁶⁷. Second, we reassign each individual to one of the nine ancestral groups based on their Euclidean distance to the centers on the PC space, as the self-reported country of origin doesn't necessarily match an individual's genetic ancestry. The genetic ancestry of an individual is labeled as unknown if its distance to any genetic ancestry center is larger than one eighth of maximum distance between any pairs of sub-continental ancestry clusters. We are able to cluster 91% of the UKBB participants into 411,018 WB, 4127 PL, 1169 IR, 6499 IT, 2352 AS, 1798 CH, 2472 CB and 3894 NG. GIA are not necessarily reflective of the full genetic diversity of a particular region but only reflect the diversity present in the UKBB individuals."

2. In response to my comment to the authors that previous studies have demonstrated that PRS accuracy depends on genetic similarity between the training sample and the target population, the authors state that the major advancement of their work is that they consider PGS performance at the level of a single target individual rather than a population as a function of genetic distance to training

data. I think this is a valid point. However, the training data used for PGS are from a group of individuals who can have quite variable genetic ancestries. So this approach proposed by the authors still relies on clustering individuals into a single discrete genetic ancestry group (for training), which I think is a limitation that is not solved by looking at individual level data in the target sample. I think the authors should include some discussion about this.

We thank the reviewer for raising this important point. We fully agree with the reviewer that the novelty of our manuscript is in showing that PGS performance varies at level of a single target individual. We also agree with the main limitation raised by reviewer in that training of PGS still relies on a cluster of genetic ancestries (as this limitation comes directly from the standard approach for performing GWAS).

We acknowledge this limitation that comes mainly from the methodologies that are used to estimate PGS. All such methods train PGS in one or more GIA (with appropriate control for population stratification through PCA and/or mixed models) thus relying on a clustering of genetic ancestries. For example, LDpred2 investigates a single GIA whereas more recent methods such as PRS-CSx integrates multiple GIA albeit each GIA is still treated as a discrete clustering of individuals. We leave the integration of continuous genetic ancestries within PGS training as important future work.

Revised discussion section:

“Sixth, while we advocate for the use of continuous genetic ancestry, we trained our PGS models on a discrete GIA cluster of WB because current PGS methods relies on the discrete genetic ancestry grouping to control for confounding effect of population structure (although substantial heterogeneity still remains within the group). We leave the development of PGS training methods that are capable to model continuous human genetics as future work.”

3. Related to comment 2 above, how does variability in genetic ancestry in the training data impact prediction accuracy? It would be good for the authors to provide some insight or discussion on this.

We thank the reviewer for raising this important question. Indeed variability in genetic ancestries could improve prediction accuracy (assuming population stratification is correctly accounted for) as follows:

- Genetic diversity in the training data allows for the estimation of ancestry-specific genetic variants that would otherwise be missed in a less diverse data set. For example, genetic variants that cause disease and are common in Africans but rare/absent in Europeans will be missed when training PGS on European only data. This will yield poor PGS prediction accuracy for individuals of African ancestries.
- Even for variants that are common across ancestries, multi-ancestry training data can leverage differences in genetic structure across ancestries (e.g., LD) to better pinpoint the true causal biological genetic effect thus yielding superior PGS performance across individuals of all ancestries.
- Variability in genetic ancestries in the training data can be used to better control for over-fitting of PGS in the training thus improving the prediction accuracy in a new testing data set.

We include these items in the discussion section as follows:

Revised Discussion section:

“Incorporating diverse populations in the training data can provide advantages for both European and non-European individuals. This is because when the SNPs associated with a trait are observed with different MAF and LD in non-European populations, PGS methods will provide more accurate genetic effects estimation. Furthermore, by broadening the diversity of the training data, PGS models will become more transferable to non-European populations as a result of reduced genetic distance from training data. However, increased diversity may bring challenges to statistical modeling, for example, the genetic difference may tag environment variation and bias the genetic risk prediction. Therefore, more sophisticated statistical methods are needed for entangling the correlation between genetic and environmental factors to provide accurate genetic risk estimation.”

4. It is a bit perplexing to me to construct an African PGS by using a combined sample of individuals from Nigeria and from Caribbean, which are two very different populations, the latter of which are admixed. Additional insight on this would be useful. Also, would prediction accuracy be different if individuals used for the PGS training were from a single country versus multiple nationalities? What are advantages/disadvantages of using individuals from different countries with ancestry derived from the same continent.

We thank the reviewer for raising the concern about combining individuals from different GIA to form a sufficiently large training data for PGS simulation purposes. We agree with the reviewer’s comment, and we find it unfortunate that large single GIA African publicly available data sets do not currently exist for the purpose of PGS simulations.

That being said, we argue that the simulation-based results using this ad-hoc data set are informative for the point being made: that PGS accuracy decay is not specific to PGS models trained on European ancestries data but a general feature when there are mismatch between training and testing data. The main concern about merging two GIA is the unaccounted population stratification. We note that we do not analyze any real traits in this data but simulated phenotypes from the real genotype data. Since our simulations do not include population stratification (causal variants and effects are randomly sampled independently of structure) the impact of population stratification is minor in our simulations.

Finally, regarding considering the nationality in PGS training, advantages include (1) integrating individuals from different nationality may still increase the genetic diversity in training population, thereby improving PGS accuracy as discussed in our response to comment 3. (2) different nationalities may tag environmental variation such as climate, social economic status, and health care, etc. Proper modeling of nationality information can help to disentangle the genetic and environmental factors of diseases/traits, thus providing a more unbiased genetic risk estimation and improve prediction accuracy. However, confounding effect introduced by different nationality may bias genetic risk estimation and reduce prediction accuracy. Therefore, careful modeling of nationality is advantageous for improving the accuracy and portability of PGS.

Revised Discussion section:

“Fifth, limited by sample size, we combined two GIA groups NG and CB as a training set to replicate PGS accuracy decay when using non-European ancestries as training data in simulation experiments. We acknowledge that it is not an optimal strategy for real data analysis as the population structure in the training data may confound with the true genetic effects and reduce prediction accuracy. We leave a more comprehensive investigation of non-European PGS training data for future work.”

5. In order to assess the practical utility of PRS, I asked the reviewers to provide correlation values of the predicted PRS and actual phenotype values. In response to this request, the authors have provided Supplementary Figure 7 that includes figures with “Relative accuracy” as compared to the White British group. This is insufficient as relative accuracy does not allow for assessment of PRS practicality. The authors need to provide updated or additional figures with absolute correlation values of predicted PRS and phenotype values as the response variable, as opposed (or in addition to) relative accuracy values.

We thank the review for the comment. We have included an updated figure of absolute correlation of PRS and phenotype values to assess the practical utility of PGS.

Supplementary Figure 7. Empirical PGS accuracy decreases with genetic distance in UKBB averaged across 84 traits.

(a) Empirical PGS accuracy decreases across subcontinental ancestries. (b) Empirical PGS accuracy decreases across bins of genetic distance. The x-axis is the average genetic distance for all individuals within each genetic ancestry cluster/genetic distance bin; the y-axis is the accuracy for each subcontinental ancestry/genetic distance bin. The dot and error bar show mean and ± 1.96 standard error of the mean across 84 traits.

6. Supplementary Figures 2 and 3 need to be improved. The authors are using similar colors for different countries and it is very difficult to decipher what points correspond to which countries. For example, Italy and the Caribbean are both plotted in green, while Poland, Nigeria, and Iran are all plotted in blue. Different color schemes need to be used for improved clarity.

We thank the reviewer for proposing better color scheme. We have revised the color schemes for supplementary figure 2 and 3 and all relevant UKBB figures to ensure clarity.

Supplementary Figure 2. PGS performance varies across genetic distance in simulations using Africans as training data ($h_g^2 = 0.8$ and $p_{causal} = 0.1\%$). (a) The coverage of the 90% credible intervals is approximately uniform across testing individuals at all genetic distances. The red

dotted line represents the expected coverage of 90% credible interval. Each dot represents a randomly selected UKBB testing individual. For each dot, the x-axis is its genetic distance from African training data, the y-axis is the empirical coverage of 90% credible interval calculated as the proportion of simulation replicates where the 90% credible intervals contain the individual's true genetic liability, and the error bars represent mean ± 1.96 standard error of the mean (s.e.m) of the empirical coverage calculated from 100 simulations. (b) The width of 90% credible interval increases with genetic distance. For each dot, the y-axis is the width of 90% credible interval across 100 simulation replicates, and the error bars represent ± 1.96 s.e.m. (c) Individual PGS accuracy decreases with genetic distance. For each dot, the y-axis is the average individual level PGS accuracy across 100 simulation replicates, and the error bars represent ± 1.96 s.e.m. (d) Population-level metrics of PGS accuracy recapitulates the decay in PGS accuracy across genetic continuum. All UKBB testing individuals are divided into 100 equal-interval bins based on their genetic distance. The x-axis is the average genetic distance for the bin and the y-axis is the squared correlation between genetic liability and PGS estimates for the individuals within the bin. The dot and error bars represent the mean and ± 1.96 s.e.m from 100 simulations.

Supplementary Figure 3. PGS performance varies across genetic distance in simulations using Africans as training data ($h_g^2 = 0.8$ and $p_{\text{causal}} = 1\%$). (a) The coverage of the 90% credible intervals is approximately uniform across testing individuals at all genetic distances. (b) The width of 90% credible interval increases with genetic distance. (c) Individual PGS accuracy decreases with

genetic distance. (d) Population-level metrics of PGS accuracy recapitulates the decay in PGS accuracy across genetic continuum.

Reviewer Reports on the Second Revision:

Referees' comments:

Referee #3 (Remarks to the Author):

The authors have done an excellent job addressing my comments and concerns in this latest revision of the manuscript. In particular, the additional content added now clearly distinguishes genetic ancestry from race and nationality, and the use of the term "genetically inferred ancestry (GIA)" provides additional clarity. I have no further comments or suggestions.